**EMBO** *reports*

# Lack of *At*MC1 catalytic activity triggers autoimmunity dependent on NLR stability

Jose Salguero-Linares[1], Laia Armengot [1], Joel Ayet [1], Nerea Ruiz-Solaní [1], Svenja C Saile [2,12], Marta Salas-Gómez [1], Esperanza Fernandez[3,4], Lode Denolf [3,4], Fernando Navarrete [1], Jenna Krumbach[1,13], Markus Kaiser[5], Simon Stael [6,7,8], Frank Van Breusegem[7,8], Kris Gevaert [3,4], Farnusch Kaschani [5], Morten Petersen [9], Farid El Kasmi [2], Marc Valls[1,10] & Núria S Coll [1,11]✉

## Abstract

**Plants utilize cell surface-localized pattern recognition receptors (PRRs) and intracellular nucleotide-binding leucine-rich repeat (NLR) receptors to detect non-self and elicit robust immune responses. Fine-tuning the homeostasis of these receptors is critical to prevent their hyperactivation. Here, we show that Arabidopsis plants lacking metacaspase 1 (*At*MC1) display autoimmunity dependent on immune signalling components downstream of NLR and PRR activation. Overexpression of a catalytically inactive *At*MC1 in an *atmc1* background triggers severe autoimmunity partially dependent on the same immune signalling components. Overexpression of the E3 ligase SNIPER1, a master regulator of NLR homeostasis, fully reverts the AtMC1-dependent autoimmunity phenotype, inferring that a broad defect in NLR turnover may underlie the severe phenotype observed. Catalytically inactive *At*MC1 localizes to punctate structures that are degraded through autophagy. Considering also previous evidence on the proteostatic functions of *At*MC1, we speculate that Wt *At*MC1 may either directly or indirectly control NLR protein levels, thereby preventing autoimmunity.**

**Keywords** Autoimmunity; Autophagy; Condensates; Metacaspases; Proteostasis
**Subject Categories** Immunology; Post-translational Modifications & Proteolysis; Signal Transduction

## Introduction

Plants perceive pathogenic microbes by detecting conserved pathogen-associated molecular patterns (PAMPs) at the plasma membrane through pattern-recognition receptors (PRRs), triggering PAMP-triggered immunity (PTI) (Jones and Dangl, 2006). Successful pathogens secrete effector proteins that can be delivered into the plant cell to dampen PTI responses (Couto and Zipfel, 2016). Intracellular immune receptors of the nucleotide-binding leucine-rich repeat-type (NLRs) detect pathogen effectors either directly or indirectly, unleashing a robust immune response termed effector-triggered immunity (ETI) that culminates in disease resistance (Jones and Dangl, 2006). Disease resistance is often accompanied by a form of localized cell death at the pathogen ingress site termed hypersensitive response (HR) (Balint-Kurti, 2019). Accumulating evidence supports the notion that immune pathways activated by PRRs and NLRs mutually potentiate each other to activate strong defences against pathogens (Ngou et al, 2021; Tian et al, 2021; Yuan et al, 2021).

NLRs are functionally classified into sensor NLRs (sNLRs), involved in perceiving pathogen effectors or monitoring their activity, and helper NLRs (hNLRs), which amplify the immune signal downstream of effector recognition and are evolutionarily more conserved (Jubic et al, 2019). NLRs can be further classified based on their N-terminal domain. sNLRs can harbour either a coil-coiled domain (CNLs) or a Toll/Interleukin 1-receptor domain (TNLs), whereas a RPW8 (RESISTANCE TO POWDERY MILDEW 8)-like CC domain is characteristic of hNLRs of the conserved RNLs family. Within RNLs, two main gene families have been described in *Arabidopsis thaliana* (hereafter Arabidopsis) encoding *ADR1* (*ACTIVATED DISEASE RESISTANCE 1: ADR1, ADR1-L1* and *ADR1-L2*) and *NRG1* (*N-REQUIREMENT GENE 1: NRG1.1, NRG1.2* and *NRG1.3*) (Jubic et al, 2019). While certain CNLs function as singletons, TNL function is genetically dependent

[1]Centre for Research in Agricultural Genomics (CRAG), CSIC-IRTA-UAB-UB, Campus UAB, Bellaterra 08193, Spain. [2]Center for Plant Molecular Biology (ZMBP), Eberhard Karls University of Tübingen, Tübingen, Germany. [3]VIB Center for Medical Biotechnology, VIB, B9052 Ghent, Belgium. [4]Department of Biomolecular Medicine, Ghent University, B9052 Ghent, Belgium. [5]Center of Medical Biotechnology (ZMB) University of Duisburg-Essen, Universitätsstr. 2, 45141 Essen, Germany. [6]Department of Molecular Sciences, Uppsala BioCenter, Swedish University of Agricultural Sciences and Linnean Center for Plant Biology, Uppsala, Sweden. [7]Department of Plant Biotechnology and Bioinformatics, Ghent University, 9052 Ghent, Belgium. [8]Center for Plant Systems Biology, VIB, B9052 Ghent, Belgium. [9]Department of Biology, University of Copenhagen, 2200 Copenhagen N, Denmark. [10]Department of Genetics, Microbiology and Statistics, Universitat de Barcelona, 08028 Barcelona, Spain. [11]Consejo Superior de Investigaciones Científicas (CSIC), 08001 Barcelona, Spain. [12]Present address: Plant Health Institute of Montpellier (PHIM), Université de Montpellier, INRAE, CIRAD, Institut Agro, IRD, Montpellier, France. [13]Present address: Department of Root Biology and Symbiosis, Max Planck Institute of Molecular Plant Physiology, Potsdam Science Park, Potsdam-Golm 14476, Germany. ✉E-mail: nuria.sanchez-coll@cragenomica.es

on the two immune nodes ENHANCED DISEASE STIMULATING 1-PHYTOALEXIN DEFICIENT 4 (EDS1-PAD4) and EDS1-SENESCENCE ASSOCIATED GENE 101 (EDS1-SAG101) (Feys et al, 2005, 2001). EDS1-PAD4 and EDS1-SAG101 form mutually exclusive heterodimers that associate with members of the RNL family ADR1 and NRG, respectively (Huang et al, 2022; Jia et al, 2022; Bonardi et al, 2011; Castel et al, 2019; Wu et al, 2019; Saile et al, 2020; Lapin et al, 2019). ADR1 and NRG1 oligomerize into pentameric resistosomes and can act as calcium-permeable cation-selective channels on their own and are enriched in PM puncta (Jacob et al, 2021; Bi et al, 2021). Interestingly, NRG1 can target organellar membranes (Ibrahim et al, 2024).

A genetically parallel pathway involving the synthesis of the phytohormone salicylic acid (SA) is required for transcriptional changes in defence-related genes during plant immunity (Cui et al, 2017; Mine et al, 2018). The SA pathway is dependent on the ISOCHORISMATE SYNTHASE 1 (ICS1 also known as SID2) enzyme and is bolstered by the EDS1-PAD4-ADR1 immune node via a mutually reinforcing feedback loop (Cui et al, 2017; Sun et al, 2021). Certain PRRs, such as the receptor-like kinase SUPPRES-SOR OF BIR1-1 (SOBIR1), links the surface-localized RECEPTOR-LIKE PROTEIN 23 (RLP23), that recognizes PAMPs, to the EDS1-PAD4-ADR1 immune node. Hence, EDS-PAD4-ADR1 might serve as a convergence point for signalling cascades elicited by either NLRs or PRRs, in conferring plant immunity (Pruitt et al, 2021).

Compared to mammals, higher plants encode a larger number of NLRs and PRRs that upon pathogen recognition are transcriptionally upregulated to exert a robust immune response (Tian et al, 2021). At the post-translation level, the ubiquitin-proteasome system (UPS) has been shown to maintain NLR homeostasis. Plant genomes encode for an extensive number of E3 ubiquitin ligases (~1500 genes) mediating diverse biological functions, including PRR and NLR turnover (Cheng et al, 2011; Gou et al, 2012; Liao et al, 2017; Lu et al, 2011; Mazzucotelli et al, 2006). Recently, the master E3 ligases, SNIPER1 and SNIPER2, have been shown to suppress autoimmune phenotypes caused by hyperactive gain-of-function NLR mutants by broadly regulating sNLR protein levels (Wu et al, 2020a). Further, the involvement of autophagy in the process cannot be ruled out, considering the complex interplay between the two protein degradation pathways that is just beginning to be unveiled (Raffeiner et al, 2023). Since tight control of NLR and PRR homeostasis is of utter importance for plant fitness and for avoiding autoimmunity, parallel and possibly redundant mechanisms to regulate immune receptor homeostasis may exist.

Plant metacaspases are an ancient group of cysteine proteases found in plants, yeast and protozoa (Minina et al, 2017). They are structurally divided into Type Is, which harbour an N-terminal prodomain, and Type IIs, which lack the prodomain but instead have a long linker region in between the p10 and p20 catalytic subunits. The Arabidopsis genome encodes for nine metacaspases, three Type Is (AtMC1-3/AtMCAIa-c) and six Type IIs (AtMC4-9/AtMCAIIa-f) (Tsiatsiani et al, 2011). Most metacaspases described to date participate in responses to stress, both biotic and abiotic (Coll et al, 2010; Escamez et al, 2016; Hander et al, 2019; He et al, 2008; Lambert et al, 2023; Luo et al, 2023; Pitsili et al, 2023; Ruiz-Solaní et al, 2023; Wu et al, 2024; Zou et al, 2023). In the context of plant immunity, the two type I metacaspases, AtMC1 and AtMC2, have been shown to antagonistically regulate HR triggered by avirulent pathogens in

young plants (Coll et al, 2010). While AtMC1 positively regulates HR in a catalysis-dependent manner, AtMC2 exerts its negative HR regulation despite the presence or absence of its catalytic cysteine (Coll et al, 2010). Importantly, the attenuated HR observed in young atmc1 mutants challenged with avirulent pathogens does not translate in enhanced disease susceptibility, uncoupling HR from disease resistance in this case (Coll et al, 2010). In contrast, adult atmc1 plants display reduced disease susceptibility (Wang et al, 2021), consistent with the age-dependent role of this protein (Coll et al, 2014; Ruiz-Solaní et al, 2023).

In the context of proteostasis, our lab has recently demonstrated that AtMC1 acts as a disaggregase to mitigate proteotoxic stress (Ruiz-Solaní et al, 2023). Although proteotoxic stress has been mostly studied in the context of heat stress, it is plausible to hypothesize that upon pathogen-triggered immune receptor activation proteotoxicity also occurs. In line with this, AtMC1 has been shown to negatively regulate protein accumulation of the auto-active hNLR mutant ADR1-L2 (D484V) and consequently, ADR1-L2 (D484V) autoimmunity is exacerbated when the atmc1 mutant allele is introduced in ADR1-L2 (D484V) plants (Roberts et al, 2013). Similarly, the maize ZmMC1 was also shown to negatively regulate immunity outputs triggered by auto-active sensor CNLs, causing re-localization of the NLRs tested to punctate dots (Luan et al, 2021). Interestingly, overexpressing the prodomain alone of AtMC2 results in autoimmunity that requires receptor-like kinases (RLKs) BAK1/BKK1 and SOBIR1 (Wu et al, 2024). Together, these data provide strong evidence that the Type I metacaspases AtMC1 and AtMC2 have a major role in immunity, although the mechanistic basis of their function remain poorly understood.

Herein, we set up to understand how AtMC1 genetically contributes to the turnover of immune receptors. We report that absence of AtMC1 results in autoimmunity that is dependent on SA synthesis and immune signalling through the convergent EDS1-PAD4 node. This phenotype is dramatically exacerbated by constitutive expression of a catalytically inactive AtMC1 variant. The catalytically inactive variant localizes to cytoplasmic condensate-like punctuate structures and co-immunoprecipitates with sNLRs, PRRs and other immune-related components. Since this phenotype is rescued by overexpressing the master regulator of sNLRs protein levels, SNIPER1, but not by mutating individual sNLRs or PRRs, we hypothesise that catalytically inactive AtMC1 acts as a platform where immune components targeted for autophagic degradation are trapped, thus interfering with their timely turnover. Based on these data, we infer that upstream of EDS1-PAD4 and SA synthesis, Wt AtMC1 might participate in the proteostatic turnover of immune components via condensate formation and autophagic degradation, thus preventing immune hyperactivation as plants approach adulthood.

## Results

### Absence of AtMC1 results in autoimmunity dependent on SA synthesis and signalling through the EDS1-PAD4 immune node

We previously reported that the Arabidopsis transfer DNA (T-DNA) knockout mutant atmc1 displays an early senescence phenotype when transferred from short day to long day photoperiod (Coll et al, 2014).

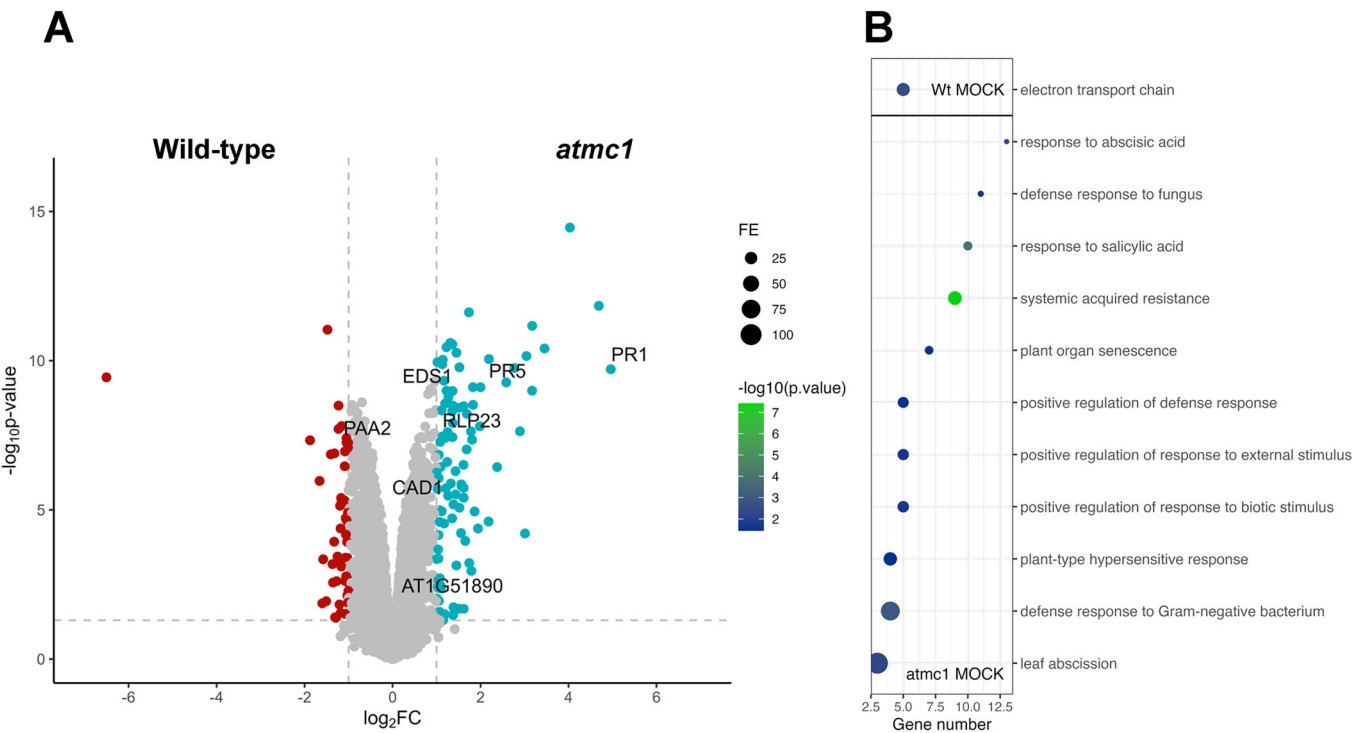

**Figure 1. Absence of *At*MC1 results in mild autoimmunity.**

(A) Volcano plot analysis of leaf proteome. Corrected *p*-value < 0.05 and log2 fold-change >1 for wild-type protein content (red dots) or <−1 for *atmc1* protein content (blue dots). Grey-coloured dots represent insignificant values (*p* ≥ 0.05) and/or ≥−1 log2 fold-change ≤1. Statistical analysis was done using a two-tailed unpaired Student's t test (*n* = 5 biological replicates per genotype per condition). Labelled dots correspond to proteins related to plant immunity. (B) Dot plot of gene ontology term showing the enriched pathways at the *p*-value < 0.05 significance level. Colours indicate the *p*-values from Fisher's exact test, Bonferroni corrected, and the dots' size is proportional to the number of differentially accumulated proteins in the given pathway. On top GO terms corresponding to *wild-type* significant proteins, and on the bottom, for *atmc1*. Source data are available online for this figure.

Label-free shotgun proteomics indicated hyperactivation of defence-related processes in *atmc1 vs* Wt plants (Fig. 1A). As shown in Fig. 1B, knock-out *AtMC1* plants featured an enrichment in proteins belonging gene ontology (GO) categories related to defence responses against pathogens, defence hormone signalling, systemic acquired resistance, hypersensitive response or senescence. The fact that *atmc1* plants may have defences on under basal conditions may underscore autoimmunity. Indeed, when continuously grown under short day conditions, *atmc1* plants exhibited hallmarks of autoimmunity: age-dependent growth restriction, spontaneous cell death, protein accumulation of the defence marker PATHOGENESIS-RELATED 1a (PR1a) and enhanced disease resistance to virulent *Pseudomonas syringae pv. tomato (Pto)* (Figs. 2 and EV1F,G). The same phenotypic features were observed for the T-DNA mutant (Fig. 2; Coll et al, 2010) and a full deletion CRISPR mutant of *AtMC1* (*atmc1-CR #1*, Fig. EV1), although PR1a accumulated slightly less in the T-DNA mutant compared to the CRISPR mutant (Fig. EV1E). This could be due to the fact that the T-DNA insertion is located after the first exon of the AtMC1 gene in *atmc1* plants, compared to the full-deleted gene in the *atmc1-CR #1 plants*. Interestingly, only *atmc1* mutants but no other type I metacaspase mutants (*atmc2* and *atmc3*) or a type II metacaspase mutant (*atmc4*) displayed autoimmunity (Appendix Fig. S1).

To explore the genetic contribution of core immune signalling components and SA synthesis to the autoimmune phenotype of *atmc1*

plants, we individually introduced mutant alleles impaired in basal immunity and ETI signalling (*eds1-12, pad4-1* and *nrg1.1 nrg1.2*) and SA synthesis (*sid2-1*) into the *atmc1* mutant background. Suppression of SA synthesis (*atmc1 sid2-1*) and EDS1-PAD4-dependent immune signalling (*atmc1 eds1-12, atmc1 pad4-1*) restored Wt-like plant growth (Fig. EV2A,C), prevented spontaneous cell death (Fig. EV2B), and suppressed PR1a protein accumulation (Fig. EV2D) observed in *atmc1* mutant plants. By contrast, introgression of the mutant alleles *nrg1.1 nrg1.2* which impair immunity through the hNLR gene family NRG1 did not prevent spontaneous cell death and PR1a protein accumulation in *atmc1 nrg1.1 nrg1.2* plants, although restoration of growth could be observed (Fig. EV2). Altogether, we conclude that autoimmunity in *atmc1* plants is dependent on SA synthesis and signalling through the EDS1-PAD4 immune node.

## Overexpression of a catalytically inactive variant of *At*MC1 (*At*MC1^C220A^) in an *atmc1* background triggers severe autoimmunity

To test whether the loss of *At*MC1 catalytic activity is sufficient to cause the autoimmune phenotype similar to a loss of *At*MC1, we created stable transgenic plants overexpressing either Wt *At*MC1 fused to a C-terminal GFP tag (*At*MC1-GFP) or *At*MC1-GFP with a Cys to Ala mutation in the p20 domain that renders the protease catalytically inactive (*At*MC1^C220A^) (Fig. 2A) (Coll et al, 2010).

**A**

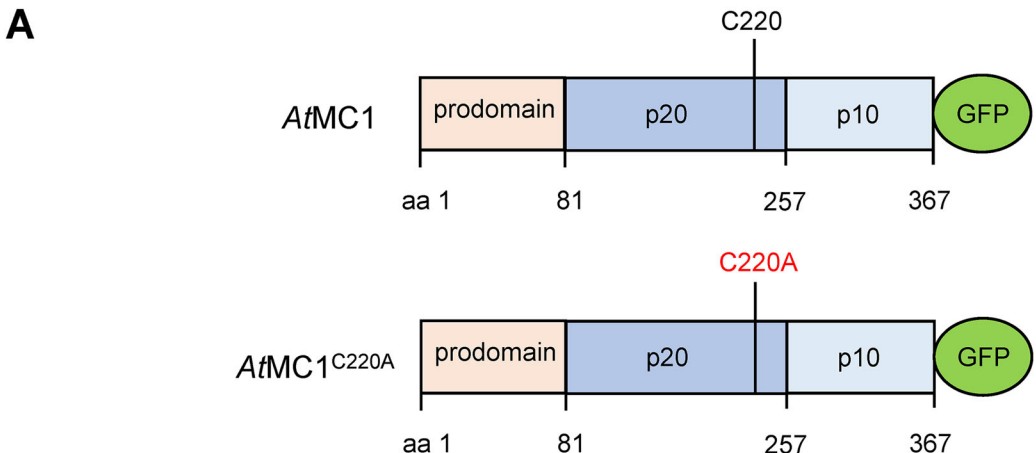

**B**

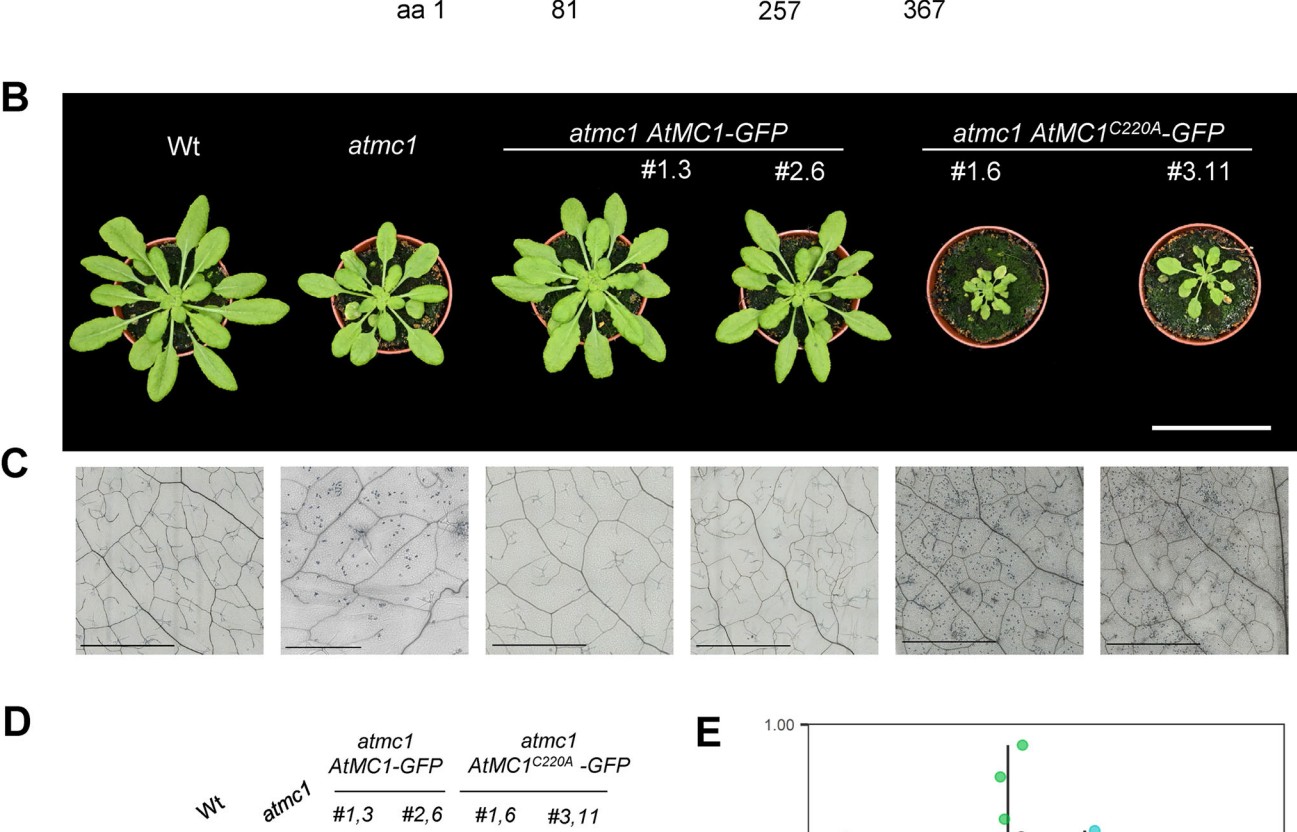

**C**

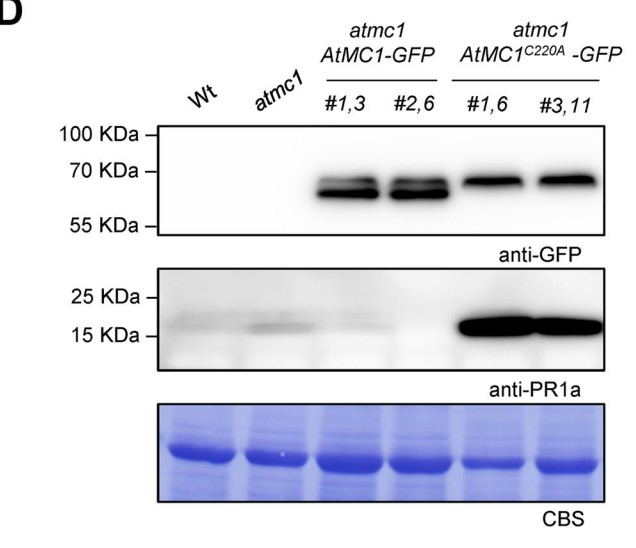

**D**

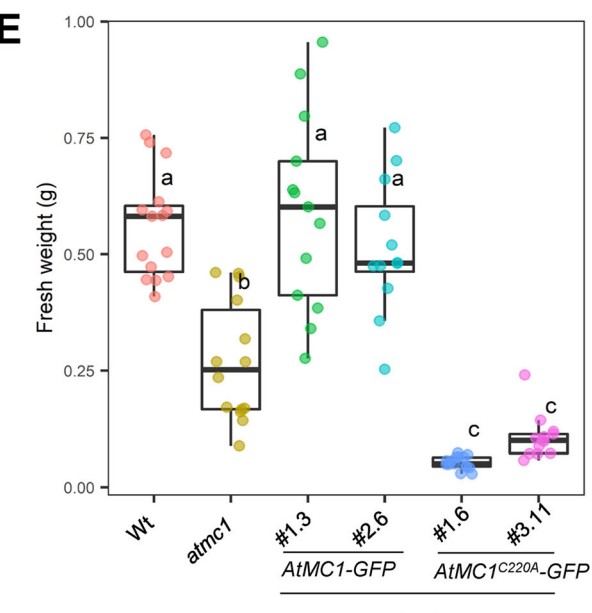

**Figure 2.  Overexpression of catalytically inactive *At*MC1 in an *atmc1* background leads to severe autoimmunity.**

(A) Scheme of *At*MC1 and catalytically inactive *At*MC1 (*At*MC1$^{C220A}$) proteins fused to GFP. The prodomain, p20 and p10 domains are indicated. The catalytic cysteine (C220) is also indicated. (B) Representative images of 40-day-old plants with the indicated genotypes grown under short day conditions. Two independent homozygous stable transgenics expressing either *AtMC1-GFP* (#1.3 and #2,6) or *AtMC1$^{C220A}$-GFP* (#1.6 and #3.11) under the control of a 35S constitutive promoter in the *atmc1* mutant background are shown. Scale bar = 5.5 cm. (C) Trypan blue staining of an area belonging to the 6th true leaf of the plants shown in (B). Scale bar = 0.5 mm. (D) Total protein extracts from the plants shown in (B) were run on an SDS-PAGE gel and immuno-blotted against the indicated antisera. Coomassie Blue Staining (CBS) of the immunoblotted membranes shows protein levels of Rubisco as a loading control. (E) Plant fresh weight of genotypes shown in (A) ($n = 12$). Different letters indicate statistical difference in fresh weight between genotypes (one-way ANOVA followed by post hoc Tukey, $p$ value < 0.05). In box plot, the centre line indicates the median, the bounds of the box show the 25th and 75th percentiles, the whiskers indicate minimum to maximum values. Source data are available online for this figure.

While overexpression of *AtMC1-GFP* fully complemented the low fresh weight (Fig. 2B,E), ectopic cell death (Fig. 2C) and PR1a protein accumulation of adult *atmc1* plants (Fig. 2D), over-expression of the catalytically inactive variant (*atmc1 AtMC1$^{C220A}$-GFP*) not only failed to complement the *atmc1* phenotype but displayed more exacerbated hallmarks of autoimmunity compared to *atmc1* mutant plants in many (at least 5) independent transgenic plants (Fig. 2B–E; Appendix Fig. S2A). Transgenic plants expressing *AtMC1-GFP* driven by its native promoter visually rescued the autoimmune phenotype of *atmc1* mutant plants, whereas expression of catalytically inactive *At*MC1$^{C220A}$ driven by its native promoter exhibited the same phenotype as *atmc1* mutant plants (Appendix Fig. S2B). These results suggest that a certain threshold of *At*MC1$^{C220A}$ protein accumulation is necessary to induce the observed severe autoimmunity and that an intact catalytic site is required to complement this phenotype. As expected, transgenic lines overexpressing catalytically inactive *At*MC2 in an *atmc2* mutant background (*atmc2 AtMC2$^{C256A}$*) did not display auto-immunity and grew as Wt and as *atmc2 AtMC2-GFP* plants (Fig. EV3A,B), suggesting that this phenomenon is exclusive to overexpression of catalytically inactive *At*MC1. Further, a potential competitive inhibition by AtMC2—the closest relative of AtMC1 and also previously involved in immunity (Coll et al, 2010)—was ruled out, as the phenotype of stable lines expressing *AtMC1$^{C220A}$-GFP* in an *atmc1 atmc2* double mutant background, was indistinguishable from *atmc1 AtMC1$^{C220A}$-GFP* (Fig. EV3C).

The N-terminal prodomain of *At*MC1 negatively regulates its function (Coll et al, 2010; Asqui Lema et al, 2018). To test whether the N-terminal prodomain was required for rescuing the auto-immune phenotype of *atmc1* plants or dispensable for the severe autoimmune phenotype in *atmc1 AtMC1$^{C220A}$-GFP* plants, we complemented *atmc1* plants with N-terminally truncated versions of *At*MC1 and *At*MC1$^{C220A}$ lacking the first 81 amino acids (*atmc1 ΔNAtMC1-GFP* or *atmc1 ΔNAtMC1$^{C220A}$-GFP*, Appendix Fig. S3A). As evidenced by overall phenotypes and fresh weight quantifications, *ΔNAtMC1-GFP* failed to rescue the *atmc1* phenotype to Wt levels (Appendix Fig. S3B–D). Interestingly, the N-terminal prodomain was required for the exacerbated autoimmune pheno-type observed in *atmc1 AtMC1$^{C220A}$* plants (Appendix Fig. S3).

*At*MC1$^{C220A}$ is an inactive protease as evidenced by the lack of self-processing (single protein band) when detected in western blots compared to Wt AtMC1 (two protein bands) (Appendix Fig. S4B; Fig. 2D). Accordingly, we asked whether the inability to be auto-processed at the junction between the N-terminal prodomain and p20 domain could explain the phenotype of plants expressing catalytically inactive *At*MC1. Given that most plant metacaspases (except *At*MC9) require Ca$^{2+}$ binding to become

active (Zhu et al, 2020), we generated transgenic plants over-expressing *At*MC1 with alanine substitutions within a conserved region of negatively charged residues in the p20 domain where Ca$^{2+}$ binds and activates *At*MC1 (D173A, E174A and D176A: *At*MC1$^{DED}$) (Zhu et al, 2020). Interestingly, although no auto-processing is observed in *At*MC1$^{DED}$-GFP extracts, *atmc1 AtMC1$^{DED}$-GFP* plants did not exhibit signs of severe autoimmu-nity and only partially restored the fresh weight defects of *atmc1* plants (Appendix Fig. S4A–C). Similarly, overexpression of an *At*MC1 variant carrying a point mutation at the predicted arginine autoprocessing site (*At*MC1$^{R49A}$) did not result in severe auto-immunity despite no autoprocessing being observed (Appendix Fig. S4D,E). Altogether, we conclude that catalytically inactive *At*MC1 triggers severe autoimmunity in a prodomain-dependent manner and that full-length catalytically active variants that are unable to be autoprocessed (*At*MC1$^{DED}$-GFP or *At*MC1$^{R49A}$-GFP) do not trigger severe autoimmunity.

## The autoimmune phenotype caused by catalytically inactive *At*MC1 is partially dependent on SA synthesis and the EDS1-PAD4-ADR1 immune node

We interrogated which components downstream of sNLRs or PRRs could be implicated in the autoimmune phenotype of *atmc1 AtMC1$^{C220A}$-GFP* plants. A deletion of EDS1 (*eds1-12*) partially rescued the fresh weight defects of *atmc1 AtMC1$^{C220A}$-GFP* plants (Fig. 3A,D), though spontaneous cell death (Fig. 3B) and PR1a accumulation still occurred (Fig. 3C). Introducing a mutation in *ICS1* (*sid2-1*) rescued fresh weight defects to the levels of *atmc1* mutant plants (Fig. 3D), partially prevented spontaneous cell death (Fig. 3B) and fully abolished PR1a protein accumulation (Fig. 3C). Mutating *SAG101* (*sag101-1*) neither rescued the fresh weight defects nor prevented PR1a protein accumulation or spontaneous cell death (Appendix Fig. S5). By contrast, mutating *PAD4* (*pad4-1*) partially rescued the fresh weight defects phenocopying *eds1-12 atmc1 AtMC1$^{C220A}$-GFP* plants (Appendix Fig. S5; Fig. 3). Mutations in the *NRG1* hNLR family (*nrg1.2 nrg1.2*) phenocopied *sag101-1 atmc1 AtMC1$^{C220A}$-GFP* plants (Appendix Fig. S5; Fig. EV4), whereas introgression of the *helperless* genetic background (all helper *NLRs* mutated: *nrg1.1, nrg1.2, adr1, adr1-l1, adr1l-2*; see Methods and Fig. EV1) also partially rescued the fresh weight defects and slightly prevented PR1a protein accumulation pheno-copying *pad4-1 atmc1 AtMC1$^{C220A}$-GFP* and *eds1-12 atmc1 AtMC1$^{C220A}$-GFP* plants (Appendix Fig. S5; Figs. EV4 and 3). We conclude that the autoimmune phenotype caused by *At*MC1$^{C220A}$-GFP is partially dependent on SA synthesis and signalling through the EDS1-PAD4-ADR1 immune node.

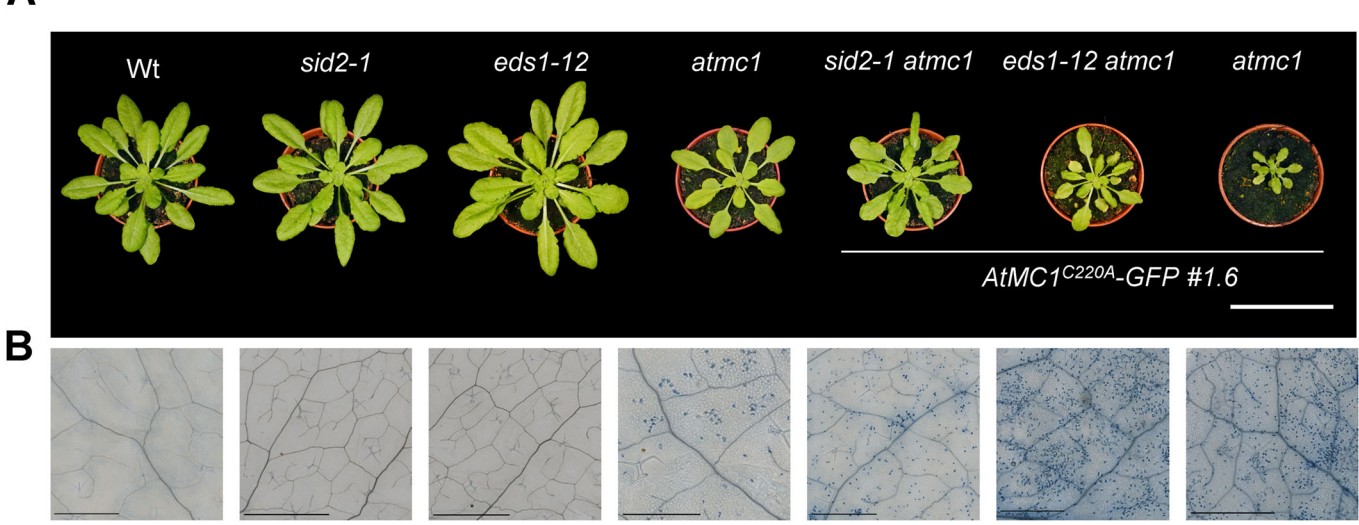

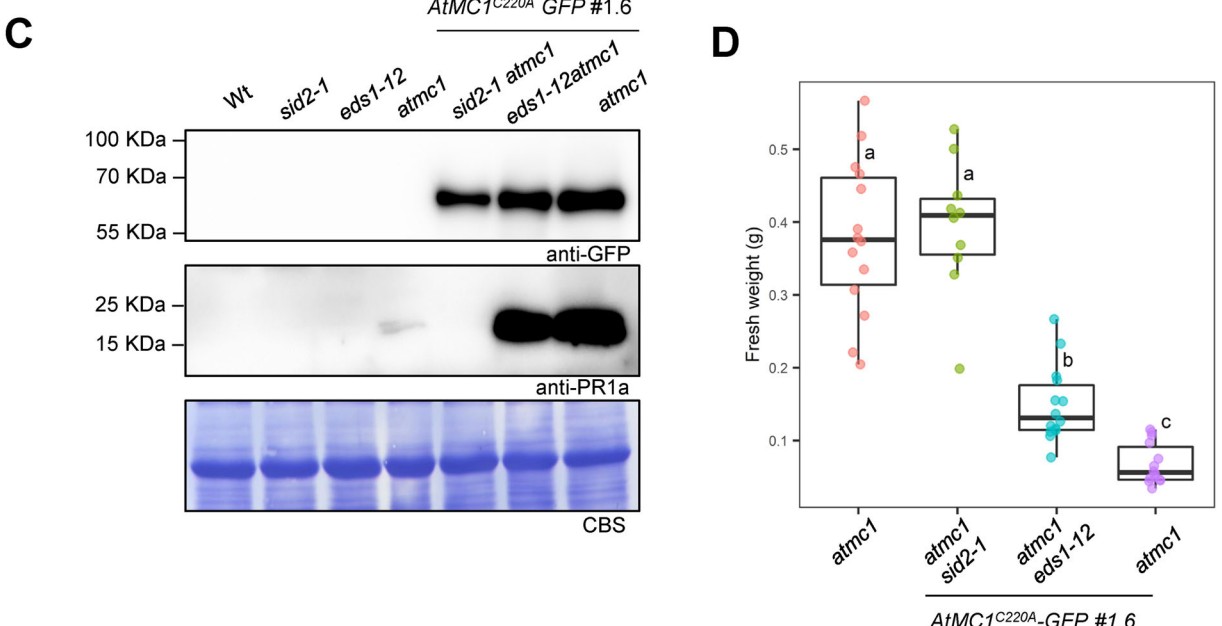

**Figure 3. Autoimmunity caused by catalytically inactive AtMC1 is partially dependent on SA synthesis and EDS1.**

(A) Representative images of 40-day-old plants with the indicated genotypes grown under short day conditions. Scale bar = 5.5 cm. (B) Trypan blue staining of an area belonging to the 6th true leaf of the plants shown in (A). Scale bar = 0.5 mm. (C) Total protein extracts from the plants shown in (A) were run on an SDS-PAGE gel and immuno-blotted against the indicated antisera. Coomassie Blue Staining (CBS) of the immunoblotted membranes shows protein levels of Rubisco as a loading control. (D) Plant fresh weight of genotypes shown in (A) (n = 12). Different letters indicate statistical difference in fresh weight between genotypes (one-way ANOVA followed by post hoc Tukey, p value < 0.05). In box plot, the centre line indicates the median, the bounds of the box show the 25th and 75th percentiles, the whiskers indicate minimum to maximum values. Quantification of fresh weight from Wt (Col-0), sid2-1 and eds1-12 were excluded from the fresh weight graph to better appreciate statistical differences between genotypes of interest. Source data are available online for this figure.

## Wt AtMC1 alleles suppress the autoimmune phenotype caused by catalytically inactive AtMC1

To test whether overexpression of catalytically inactive AtMC1 has a dominant effect over endogenous Wt AtMC1 alleles, we crossed a

Wt plant with an atmc1 AtMC1$^{C220A}$-GFP autoimmune plant and looked at the phenotype of Wt AtMC1$^{C220A}$-GFP in an F3 offspring. Interestingly, independent Wt AtMC1$^{C220A}$-GFP lines (#1,6 and #10,3) did not display autoimmunity features such as growth inhibition (Fig. 4A,B) and PR1a accumulation (Fig. 4C). To further

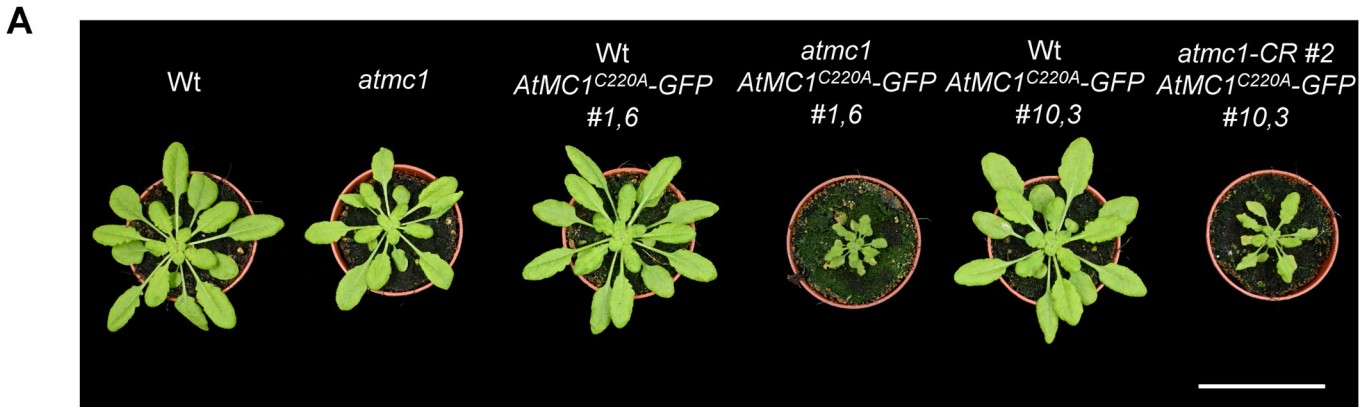

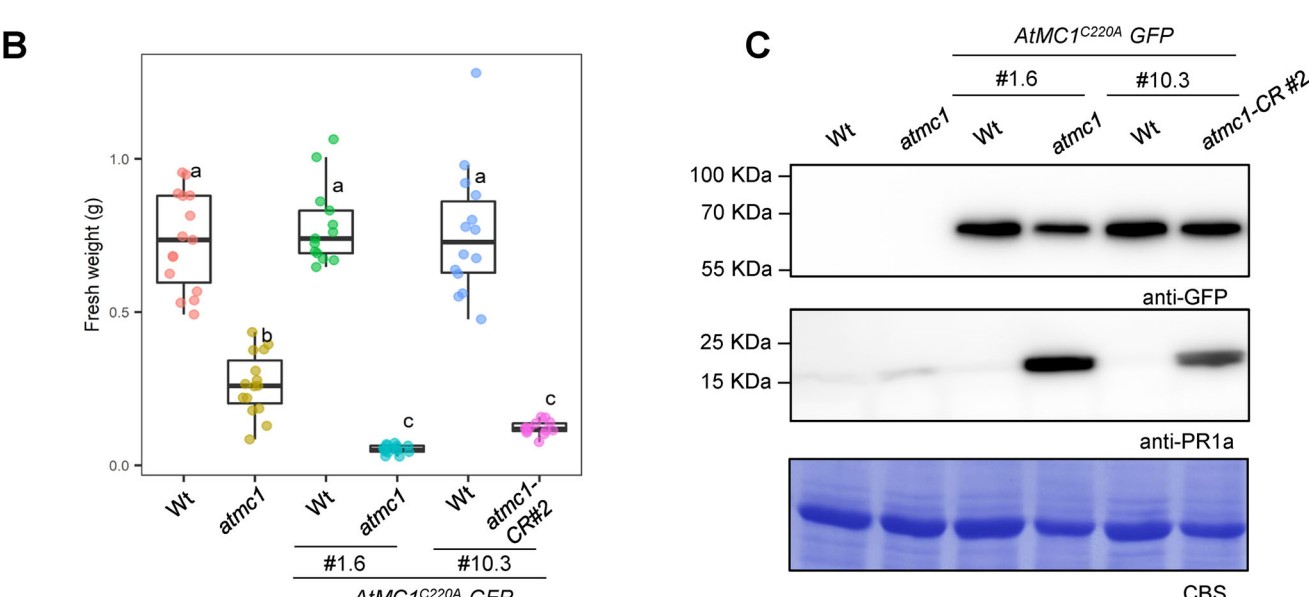

**Figure 4. Endogenous *Wt AtMC1* alleles suppress the autoimmune phenotype caused by catalytically inactive *At*MC1.**

(A) Representative images of 40-day-old plants with the indicated genotypes grown under short day conditions. Scale bar = 5.5 cm. (B) Plant fresh weight of genotypes shown in (A) ($n = 12$). Different letters indicate statistical difference in fresh weight between genotypes (one-way ANOVA followed by post hoc Tukey, $p$ value $< 0.05$). In box plot, the centre line indicates the median, the bounds of the box show the 25th and 75th percentiles, the whiskers indicate minimum to maximum values. (C) Total protein extracts from the plant genotypes shown in (A) were run on an SDS-PAGE gel and immuno-blotted against the indicated antisera. CBS of the immunoblotted membranes shows protein levels of Rubisco as a loading control. Source data are available online for this figure.

substantiate our result, we generated a CRISPR *AtMC1* deletion mutant (*atmc1-CR#2*) in line Wt *AtMC1^C220A^-GFP #10,3*, with single guide RNAs targeting the 5′ and 3′ untranslated region (UTRs) of the Wt *AtMC1* alleles (Fig. EV1A), thus not affecting the transgene which is in a coding sequence format. As expected, *atmc1-CR#2 AtMC1^C220A^-GFP* plants displayed a similar auto-immune phenotype as *atmc1* (T-DNA) *AtMC1^C220A^-GFP* plants (Fig. 4). Altogether, our data argues on the importance of gene dosage of Wt *AtMC1* alleles in suppressing the phenotype caused by catalytically inactive *At*MC1. Importantly, the levels of Wt *At*MC1 were comparable to those of *AtMC1^C220A^* in the transgenic line Wt *AtMC1^C220A^ #1,6* (Appendix Fig. S2B). Considering that the autoimmune phenotype does not occur when catalytically inactive *At*MC1 is overexpressed in a Wt background and partial rescues are achieved when mutating the same signalling components, we suspect that overexpression of catalytically inactive *At*MC1 may

represent an additive phenotype to the autoimmunity observed in *atmc1* plants.

## Catalytically inactive *At*MC1 associates with immune-related components involved in PTI and ETI

To better understand the mechanism by which catalytically inactive *At*MC1 enhances *atmc1* autoimmunity, we performed immuno-precipitation followed by mass spectrometry (IP-MS). Since catalytically inactive AtMC1 localized to microsomal fractions (total membranes) and Wt AtMC1 was mainly localized in soluble fractions (cytosol) (Fig. 5A), we pulled down *At*MC1^C220A^-GFP from extracts of Wt *AtMC1^C220A^-GFP* plants (in which no autoimmunity is visible) *vs atmc1 AtMC1^C220A^-GFP* plants (in which plants display severe autoimmunity) (Fig. 4). We used *At*MC1^C220A^-*GFP* in both backgrounds (and the same transgenic

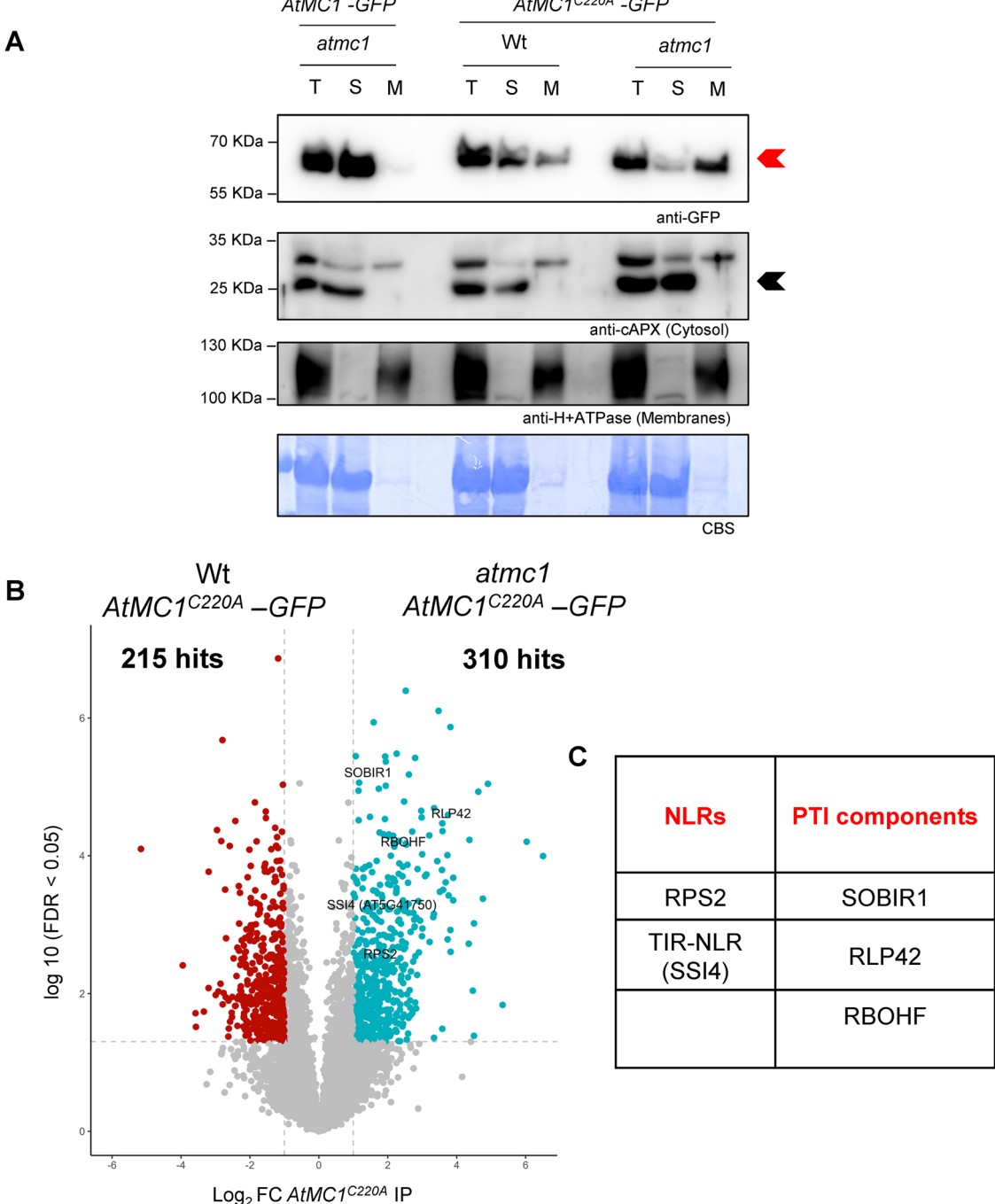

**Figure 5.  Inactive *At*MC1 is enriched in microsomes.**

(A) Fractionation assays from 40-day-old plant extracts with the indicated plant genotypes. Total (T), Soluble (S, cytoplasmic proteins) and Microsomal (M, total membranes) fractions were run on an SDS-PAGE gel and immunoblotted against the indicated antisera. Anti-cAPX and anti-H+ATPase were used as cytosol and membrane markers, respectively, to evaluate the success of fractionation. Red arrow shows levels of *At*MC1. Black arrow shows cAPX levels. Coomassie Blue Staining (CBS) of the immunoblotted membranes shows protein levels of Rubisco as a loading control. This experiment was repeated twice with similar results. (B) Volcano plot of normalized abundances (label-free quantification (LFQ), log2 scale) for proteins that immunoprecipitated with *At*MC1$^{C220A}$–GFP when expressed in either an *atmc1* mutant background (red) or a Wt background (blue) (Student´s t-test *p*-value < 0.05 and Log$_2$FC >1). The IPMS analysis was performed on samples collected in $n = 4$ independent biological replicates. (C) NLRs, and immune components involved in PTI that immunoprecipitated with *At*MC1$^{C220A}$–GFP in *atmc1* *At*MC1$^{C220A}$–GFP autoimmune plants and that were selected for further studies. Source data are available online for this figure.

**A**

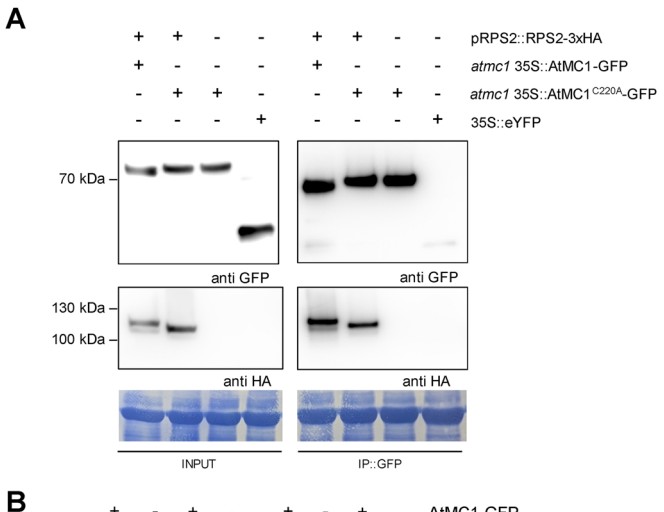

**B**

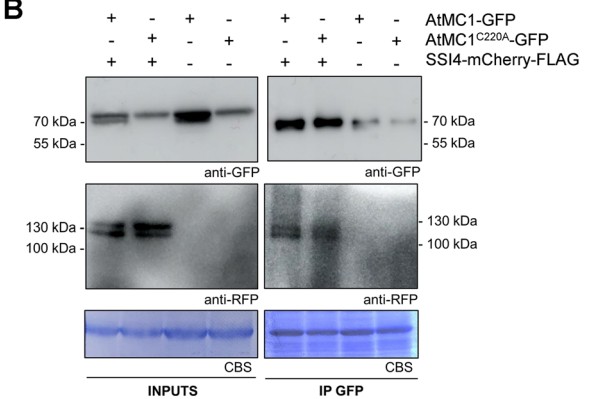

**C**

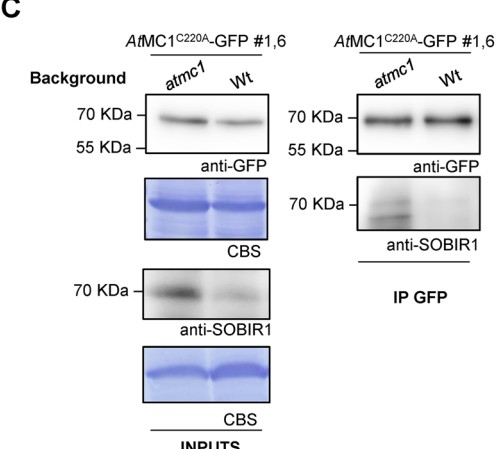

Figure 6. Catalytically inactive *At*MC1 interacts in planta with NLRs, and immune components involved in PTI.

(A, B) IP of Arabidopsis transgenics overexpressing *atmc1 35S::AtMC1-GFP*, *atmc1 35S::AtMC1^{C220A}-GFP* and proRPS2::RPS2-HA (F2 generation) (A), proS-SI4::SSI4-mCherry-FLAG (T1 generation) (B). The indicated constructs were immunoprecipitated with anti-GFP magnetic beads (IP GFP). Protein inputs from protein extracts before IP (INPUTS) and eluates from IPs were run on an SDS-PAGE and immunoblotted with the indicated antibodies. (C) *At*MC1^{C220A}-GFP was immunoprecipitated (IP: GFP) from extracts of transgenic Arabidopsis overexpressing *At*MC1^{C220A}-GFP in either an *atmc1* mutant or wild-type (Wt) background. Inputs and eluates were analysed by SDS-PAGE and immunoblotted with anti-SOBIR1 antibody. Coomassie Blue Staining (CBS) of the immunoblotted membranes shows protein levels of Rubisco as a loading control in the inputs. Source data are available online for this figure.

*At*MC1^{C220A}-GFP) *vs* Wt-looking plants (215 peptides) (Wt *At*MC1^{C220A}-GFP) (Fig. 5B). Gene Ontology (GO) searches revealed that interactors of *At*MC1^{C220A}-GFP in autoimmune plants are mainly involved in biological processes related to plant defence (Appendix Fig. S6). GO terms such as "defence-response to bacterium", "regulation of defence response", "response to wounding" and "response to SA" exhibit the greatest statistical confidence among the GOs found (Appendix Table S1).

Since it is estimated that a great proportion if not all, autoimmune phenotypes are either directly or indirectly NLR-dependent (Freh et al, 2022), we hypothesised that their aberrant hyperaccumulation through binding to catalytically inactive *At*MC1 could be the cause of the autoimmune phenotype. We found one CNL and one TNL, RPS2 and SSI4 (AT5G41750), respectively, as interactors of inactive *At*MC1 specifically in the autoimmune plant *atmc1 AtMC1^{C220A}-GFP* (Fig. 5B,C). Besides these NLRs, we also found interactors involved in PTI such as the PRR RECEPTOR-LIKE PROTEIN 42 (RLP42), and the receptor-like kinase, SOBIR1, which is required for the function of different PRRs of the RLP family (Liebrand et al, 2014). We also found the PM-localized NADPH oxidase, RBOHF, involved in active ROS production during HR and PTI, as a protein co-immunoprecipitating with *At*MC1 (Fig. 5B,C) (Torres et al, 2002). We tested association of these proteins with either Wt or catalytically inactive versions of *At*MC1 by co-immunoprecipitations (co-IPs) in *Nicotiana benthamiana* (Fig. EV5A–E). Both NLRs, RPS2-HA and SSI4-HA, co-immunoprecipitated with *At*MC1^{C220A}-GFP and to a lesser extent with Wt *At*MC1-GFP (Fig. EV5A,B). Similarly, 10xMyc-SOBIR1, 10xMyc-RLP42, and FLAG-RBOHF co-immunoprecipitated strongly with *At*MC1^{C220A}-GFP and to a lesser extent with Wt *At*MC1-GFP (Fig. EV5C–E). Importantly, the co-immunoprecipitation between *At*MC1 variants with RPS2 and SSI4 was confirmed in Arabidopsis double transgenic plants co-expressing *AtMC1-GFP* or *AtMC1^{C220A}-GFP* and either *RPS2-HA* (Fig. 6A) or *SSI4-mCherry-FLAG* (Fig. 6B) under the control of their native promoters or by using SOBIR1 antisera in *atmc1 AtMC1^{C220A}-GFP* plants using Wt *AtMC1^{C220A}-GFP* as a control (Fig. 6C).

Based on these results, we formulated two different hypotheses that could explain the phenotypes observed in *atmc1* and *atmc1 AtMC1^{C220A}-GFP* plants: (1) *At*MC1 is guarded by an NLR. (2) *At*MC1 participates in the proteostasis of immune components and catalytically inactive *At*MC1 traps immune components, thus preventing their turnover.

event #1,6) to be able to rule out the possibility of different protein levels affecting the nature of potential interactors captured in both genotypes and run a better controlled experiment. IP from plant extracts expressing free GFP (Wt *35S::GFP*) were used as a negative control. We reasoned that identifying interactors in microsomal fractions could give us a better understanding of the underlying causes of autoimmunity.

Overall, a higher number of statistically significant ($\log_2$FC > 2 FDR < 0.05) proteins were identified when *At*MC1^{C220A}-GFP was pulled down from autoimmune plants (310 proteins) (*atmc1*

## Individual mutations in NLRs and PTI-related components do not rescue the severe autoimmune phenotype of catalytically inactive AtMC1

To test our first hypothesis, we carried out an NLR-targeted forward genetic screen to find suppressors of the severe auto-immune phenotype of atmc1 AtMC1$^{C220A}$-GFP plants. We independently transformed a previously described collection of 139 dominant-negative (DN)-NLRs in atmc1 AtMC1$^{C220A}$-GFP plants (Lolle et al, 2017). These DN-NLRs carry a mutation in a conserved P-loop region within the ATPase domain of the NLR which can disrupt the function of Wt NLR alleles if both variants are co-expressed (Freh et al, 2022). This approach proved successful for the identification of two unrelated NLRs, DSC1 and DSC2, responsible for the autoimmune phenotype observed in camta3 (calmodulin-binding transcription activator 3) mutants (Freh et al, 2022; Lolle et al, 2017). Out of the 166 NLRs that should be present in Arabidopsis Col-0 accession (Lee and Chae, 2020), we individually transformed 139 DN-NLRs into the autoimmune atmc1 AtMC1$^{C220A}$-GFP plants and screened for rescued plants in the T$_1$ generation (Appendix Table S2). Neither of these DN-NLR transformations yielded a rescued plant in T$_1$, including independent T$_2$ DN-RPS2 and DN-SSI4 (AT5G41750) transgenics (Appendix Fig. S7A). Moreover, a null mutation in RPS2 (rps2-201c) and the knockout mutations in RLP42 (rlp42-2) or RBOHF (rbohf) did not suppress the autoimmune phenotype (Appendix Fig. S7B). SOBIR1 complexes were shown to recruit the co-receptor BAK1 and connect RLP23 to PAD4-EDS1-ADR1 upon ligand (PAMP) binding to RLP23 (Pruitt et al, 2021). As shown in Appendix Fig. S7C, introducing mutations in RLP23 (rlp23-1), SOBIR1 (sobir1-12) or the co-receptor BAK1 (bak1-4) did also not result in rescues of the autoimmune phenotype. Altogether, our data indicate that the severe autoimmune phenotype of atmc1 AtMC1$^{C220A}$-GFP plants seems to be mediated by more than one individual immune receptor. In light of this evidence, we set out to test our second hypothesis, i.e. that AtMC1 participates in the proteostasis of immune components and catalytically inactive AtMC1 may bind and stabilize immune components, preventing their normal turnover.

## Catalytically inactive AtMC1 forms stable condensate-like puncta and may stabilize NLRs and other immune regulators

AtMC1 has a crucial role in proteostasis during proteotoxic stress, being rapidly recruited into condensates and contributing to their timely clearance (Ruiz-Solaní et al, 2023). This function seems to rely on the condensation-prone physico-chemical properties of AtMC1 and on its disaggregase activity, for which an intact catalytic site is essential (Ruiz-Solaní et al, 2023). Considering this, one could speculate that AtMC1 might contribute to the proteostasis of NLRs and other immune regulators in situations where the levels of these proteins increase, such as during an acute immune response. Supporting this hypothesis, we observed that the levels of BIK1, SOBIR1 and RPS2 increased drastically in atmc1 AtMC1$^{C220A}$-GFP plants (Figs. 7A,B and S8A). Further, adding an extra copy of the AtMC1 interacting TNL SSI4 to atmc1 AtMC1$^{C220A}$-RFP plants (proSSI4::SSI4-mCitrine) (Appendix Fig. S8B) resulted in individuals displaying extremely strong

autoimmunity (Fig. 7C). However, the fact that mutating the catalytic site of AtMC1 does not particularly seem to strengthen the co-immunoprecipitation of this protein with its potential clients/substrates (Figs. 6 and EV5) could indicate that indeed AtMC1$^{C220A}$ acts as a trap for these proteins, since this variant of the protein substrate loading/unloading processes are hindered. Together, these data indirectly support a role of AtMC1 in preventing immune receptors hyperaccumulation dependent on its catalytic activity.

Interestingly, confocal microscopy analysis revealed that while AtMC1-GFP displayed a diffuse nucleocytoplasmic pattern (Fig. 7D and (Ruiz-Solaní et al, 2023)), AtMC1$^{C220A}$ in addition localized to puncta-like structures reminiscent of plant condensates when expressed in atmc1 plants (Fig. 7D). AtMC1$^{C220A}$ localization to condensates rarely occurred in Wt AtMC1$^{C220A}$-GFP, further supporting the observation that the endogenous Wt AtMC1 alleles suppress the phenotype caused by the catalytically inactive variant (Fig. 4). Moreover, the N-terminal prodomain is required for the localization of the catalytically inactive AtMC1 variant to the microsomal fraction and condensates (Appendix Fig. S9). Stable condensation of AtMC1$^{C220A}$ and potential stabilization of interactors within these condensates may at least partly account for the severe autoimmunity displayed by these plants.

## AtMC1$^{C220A}$ condensate-like puncta are partly degraded through autophagy

AtMC1 condensates have been previously identified as stress granules (SGs) that rapidly form and dissolve during heat stress (Ruiz-Solaní et al, 2023). Proper dynamics of SGs is essential for stress responses and their stabilization leads to proteotoxicity and chronic stress, contributing to various pathologies and accelerated aging (Ruiz-Solaní et al, 2023). Since we failed to obtain atmc1 AtMC1$^{C220A}$-GFP plants co-expressing any of the SG markers tested (Appendix Table S3), we could not definitively determine whether the observed AtMC1$^{C220A}$-containing condensates correspond to SGs. However, since one of the major mechanisms of excess SG disposal is granulophagy, a form of autophagy (Buchan et al, 2013), we tested whether AtMC1$^{C220A}$-containing condensates co-localize with autophagy markers and are destined to the vacuole for degradation. Interestingly, double transgenics expressing AtMC1$^{C220A}$-GFP along with the core autophagy receptor ATG8a, (atmc1 AtMC1$^{C220A}$-GFP x mCherry-ATG8a) exhibited partial colocalization upon treatment with the vacuolar ATPase inhibitor Concanamycin A (Conc A), which allows visualization of fluorescently labelled proteins in the vacuole (Fig. 8A). This indicates that at least part of the condensates containing AtMC1$^{C220A}$ may be targeted to vacuolar degradation via autophagy as an additional proteostatic mechanism activated to counterbalance proteotoxic stress in these autoimmune plants.

To further substantiate this result, we independently introduced mutations in ATG2 (atg2-1) and ATG5 (atg5-1), which are core autophagy machinery proteins required for the biogenesis of autophagosomes (Leary et al, 2018), in the autoimmune genotype atmc1 AtMC1$^{C220A}$-GFP. Accordingly, atg2-1 atmc1 AtMC1$^{C220A}$-GFP and atg5-1 atmc1 AtMC1$^{C220A}$-GFP plants exhibited a more severe autoimmune phenotype compared to atmc1 AtMC1$^{C220A}$-GFP (Fig. 8B). Moreover, these plants accumulated a higher number of larger AtMC1$^{C220A}$-GFP puncta compared to atmc1

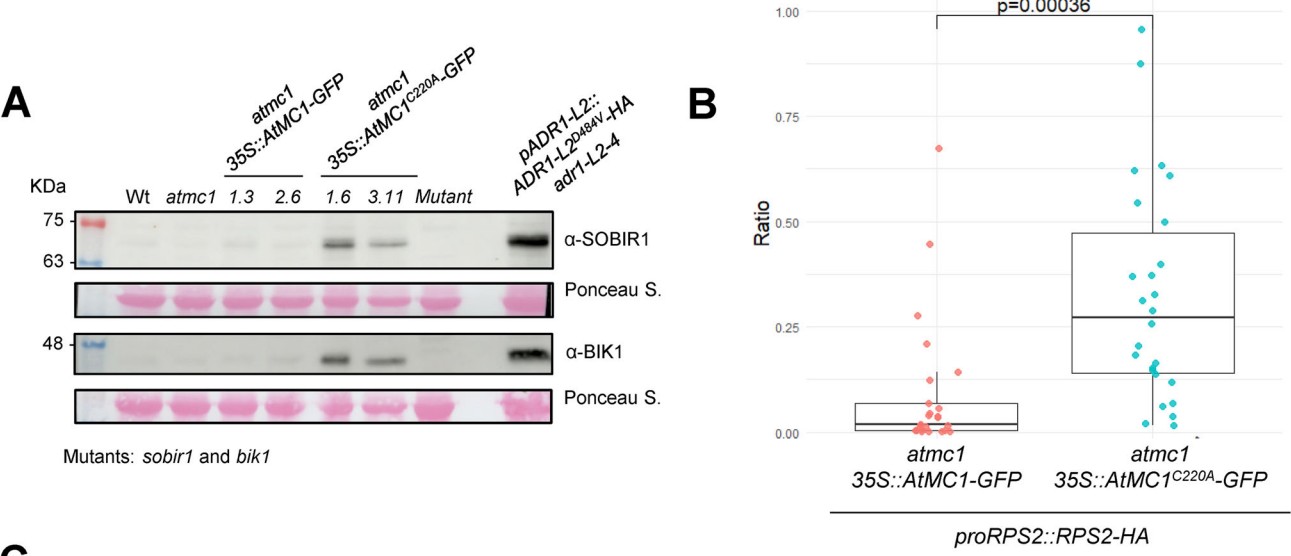

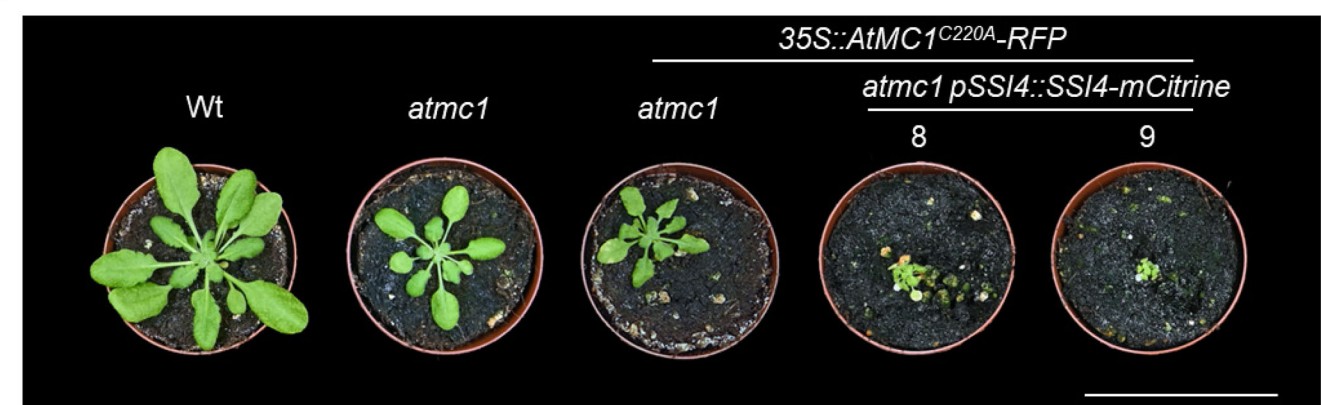

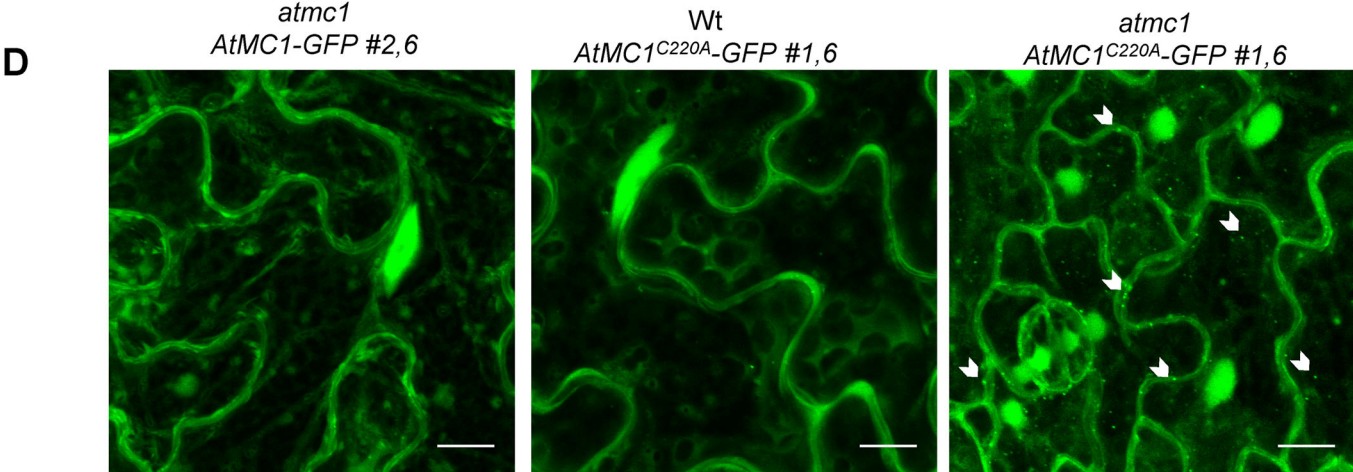

*At*MC1^C220A^-GFP plants (Fig. 8C). Altogether our results suggest that *At*MC1^C220A^-GFP complexes containing immune components are being degraded through autophagy, and impairment of this proteostatic recycling process further exacerbates the autoimmune phenotype displayed in *atmc1 At*MC1^C220A^-GFP plants.

## Overexpression of SNIPER1 rescues the autoimmune phenotype caused by catalytically inactive *At*MC1

In light of these results and considering that no genetic rescues were achieved when individually introducing mutations in sNLRs

◀ **Figure 7.  Immune components homeostasis is altered in *atmc1 AtMC1*^C220A^*-GFP*.**

(A) Total protein extracts from the indicated genotypes were run on an SDS-PAGE gel and immune-blotted with the indicated antisera. On the upper blot the mutant corresponds to *sobir1*, whilst on the lower blot to *bik1*. Ponceau staining of the immunoblotted membranes shows protein levels of Rubisco as a loading control. (B) Box plot representing the ratio of RPS2-HA protein accumulation in the indicated genotypes. The ratio RPS2::loading control was calculated using ImageJ by analysing the band intensity of anti-HA accumulation in relation to the loading control of each sample. Protein western-blots are found in Appendix Fig. S8 ($n = 25$). *P*-value was calculated using a two-tailed unpaired Student's t test. In box plot, the centre line indicates the median, the bounds of the box show the 25th and 75th percentiles, the whiskers indicate minimum to maximum values. (C) Representative images of 40-day-old plants with the indicated genotypes grown under short day conditions. Scale bar = 5.5 cm. (D) Representative confocal microscopy images from the leaf epidermis of 40-day-old plants grown under short day conditions with the indicated genotypes. Images represent a Z-stack of 18 images taken every 1 μm. Arrows indicate some of the puncta structures formed when *At*MC1^C220A^ is overexpressed in an *atmc1* mutant background. Scale bar = 10 μm. Source data are available online for this figure.

(Appendix Fig. S7 and Appendix Table S1), we explored whether a broad defect in sNLR homeostasis in plants expressing catalytically inactive *AtMC1* could account for the severe autoimmune phenotype observed. The E3 ubiquitin-ligase SNIPER1, is a master regulator that broadly controls sNLR levels (Wu et al, 2020b). SNIPER1 specifically binds to the nucleotide-binding domain (NBD) of sensor TNLs and CNLs to mediate their turnover through the 26S proteasome. Accordingly, autoimmune mutants that are sNLR-dependent such as *snc1, chs1-2, chs2-1,* and *chs3-2D* are fully rescued by over-expression of SNIPER1 (Wu et al, 2020b). Interestingly, when SNIPER1 was overexpressed in the autoimmune background *atmc1 AtMC1*^C220A^*-GFP*, independent transgenics (*atmc1 AtMC1*^C220A^*-GFP HA-SNIPER1*) exhibited an almost complete rescue in all phenotypic outputs tested: growth defects, suppression of spontaneous cell death and low accumulation of PR1a that inversely correlated with expression of SNIPER1 (Fig. 9A–C). Accordingly, overexpression of HA-SNIPER1 in the autoimmune background drastically reduced the number of *AtMC1*^C220A^*-GFP* puncta (Fig. 9D,E). Based on these data, we conclude that *atmc1 AtMC1*^C220A^*-GFP* plants might suffer from defects in overall sNLR homeostasis and consequently the phenotype is attenuated when a master regulator of sNLR levels is overexpressed, restoring proteostasis and preventing puncta stabilization. In contrast, decreasing the levels of autophagy (Fig. 8) or increasing the levels of an AtMC1-interacting NLR (Fig. 7) dramatically exacerbates autoimmunity, highlighting the importance of proteostasis in immune responses to turn down the levels of immune regulators to prevent cellular damage and reduce immunological trade-offs in plants.

## Discussion

Fine-tuning immune responses is of paramount importance for plant growth and fitness. Consequently, misregulation of immune receptor activation in the absence of pathogen attack leads to inappropriate and deleterious immune outputs, resulting in plant autoimmunity: a phenomenon in which spontaneous cell death, stunted growth, and sometimes plant lethality poses a serious disadvantage for plants (Freh et al, 2022). Hyperactivation of immune receptors during autoimmunity (particularly NLRs) may be caused by (i) gain-of-function mutations in NLRs (Roberts et al, 2013; Zhang et al, 2003), (ii) modifications or absence of NLR-monitored guardees including PTI components (Schulze et al, 2022; Wu et al, 2020a; Yang et al, 2022) or (iii) aberrant regulation of NLRs at the transcriptional and translational level (Freh et al, 2022; Van Wersch et al, 2016; Wu et al, 2020b). Alternatively, unsuited interactions between NLR loci in heterozygous progeny derived

from within-species ecotypes can lead to a class of autoimmunity known as hybrid incompatibility or hybrid necrosis (Bomblies and Weigel, 2007; Wan et al, 2021).

*At*MC1 and its homologue in maize *Zm*MC1 were previously shown to participate either in the regulation or subcellular re-localization of certain auto active NLRs, respectively (Luan et al, 2021; Roberts et al, 2013). In addition, our lab has recently demonstrated the dynamic recruitment of *At*MC1 to stress granules (SGs) in proteotoxic stress conditions, inferring a proteostatic function of *At*MC1 in clearance of aberrant aggregates that are formed under these circumstances (Ruiz-Solaní et al, 2023).

Herein, we observed that mutant plants lacking *At*MC1 display various hallmarks of autoimmunity as plants approach adulthood (Fig. 1). This includes enhanced resistance to various pathogens (Fig. 1; Coll et al, 2014; Wang et al, 2021), indicating that AtMC1-mediated autoimmunity may be mediated by multiple immune receptors. Although a role for AtMC1 in plant immunity has been previously reported (Coll et al, 2010; Wang et al, 2021) it remained unclear what is its mode of action in this context. Wang et al (2021) reported that AtMC1 may be involved in regulating the splicing of many pre-mRNAs, including regulators of plant immunity. However, the exact mechanism orchestrating this process remains unknown. Here, we explored the link between the previously reported homeostatic function of *At*MC1 and plant immunity. Interestingly, second-site mutations in key genes downstream of sNLR activation such as ICS1, EDS1 and PAD4, into the *atmc1* mutant background rescued the autoimmune phenotype, pointing towards a contribution of sNLRs to the phenotype (Figs. 3 and EV2) (Cui et al, 2017). Whilst complementation with Wt *At*MC1 rescues the phenotype, overexpression of a catalytically inactive *At*MC1 variant (*At*MC1^C220A^) in the *atmc1* mutant background results in severe autoimmunity (Fig. 2). We made use of this C-terminally GFP-tagged knock-in variant as a tool to explore mechanisms that could infer the function of Wt *At*MC1 in plant immunity, and that would otherwise remain obscured when investigating the mild autoimmune phenotype of *atmc1* mutant plants.

### AtMC1- and caspase 8-dependent autoimmunity feature striking similarities at the molecular level

Plant metacaspases are biochemically quite distinct to animal caspases, owing to their lack of aspartate specificity in their substrates and their preference for cleavage after arginine or lysine residues (Minina et al, 2017; Vercammen et al, 2004, 2007). However, metacaspases and caspases are often referred to as structural homologues as they share a common caspase-hemoglobinase fold at their catalytic domains (Minina et al,

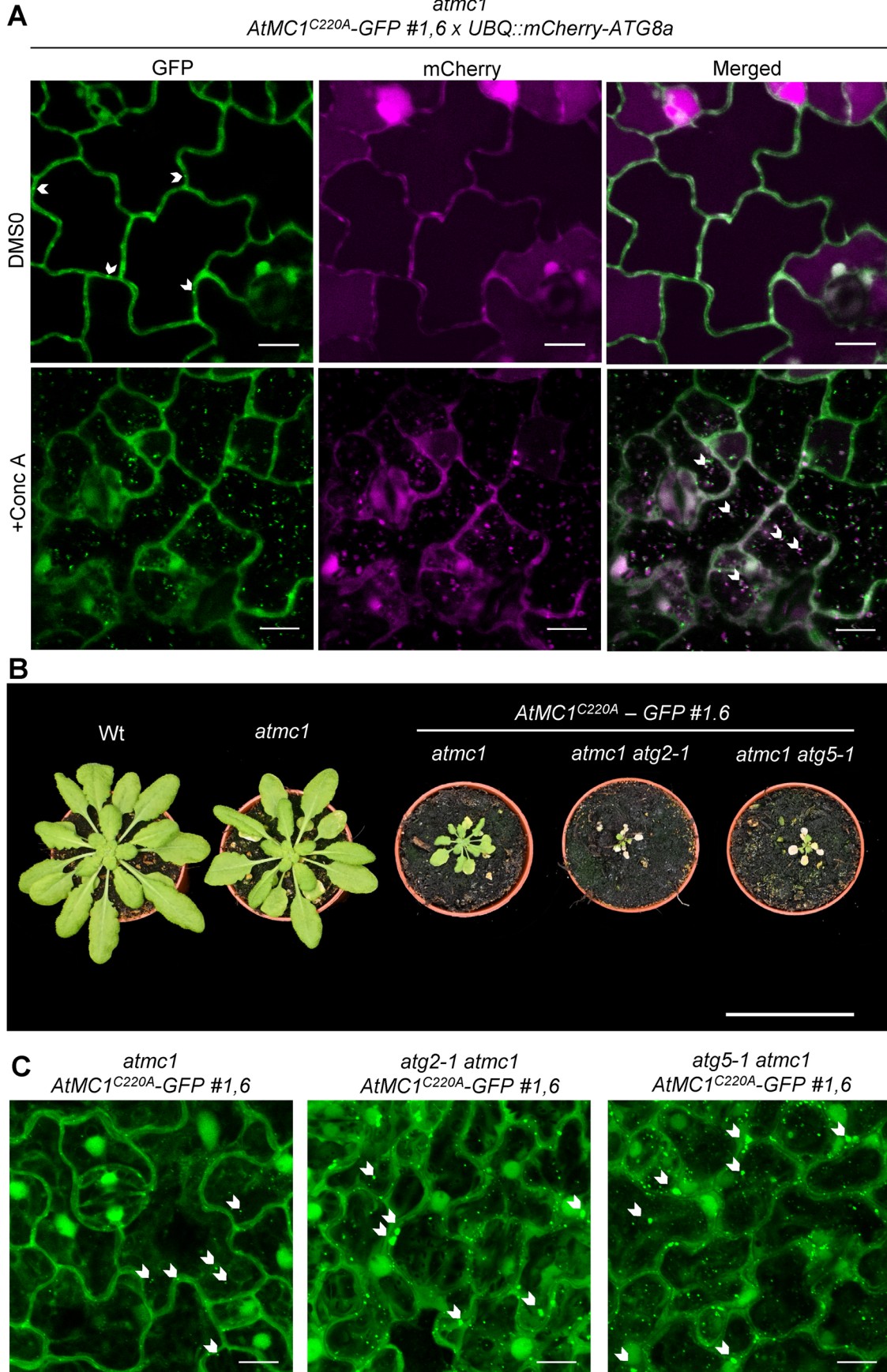

**A** *atmc1*
AtMC1$^{C220A}$-GFP #1,6 x UBQ::mCherry-ATG8a

GFP | mCherry | Merged

DMSO

+Conc A

**B** Wt | *atmc1* | AtMC1$^{C220A}$ – GFP #1.6

*atmc1* | *atmc1 atg2-1* | *atmc1 atg5-1*

**C** *atmc1*
AtMC1$^{C220A}$-GFP #1,6 | *atg2-1 atmc1*
AtMC1$^{C220A}$-GFP #1,6 | *atg5-1 atmc1*
AtMC1$^{C220A}$-GFP #1,6

**Figure 8. Catalytically inactive AtMC1 puncta are degraded through autophagy.**

(A) Representative single-plane confocal microscopy images from the leaf epidermis of 40-day-old plants grown under short day conditions with the indicated genotypes. Double transgenics expressing *UBQ::mCherry-ATG8a* (T$_2$ generation) in the *atmc1 AtMC1$^{C220A}$-GFP* background were treated with either DMSO as control (upper panels) or 1 µM Concanamycin A (Conc A) to be able to visualize fluorescently labelled proteins inside the vacuole (bottom panels). Arrows in the merged image (GFP and RFP channel) indicate colocalization of ATG8a-labelled autophagosomes along with *AtMC1$^{C220A}$* puncta structures. Scale bar = 10 µm. (B) Representative images of 40-day-old plants with the indicated genotypes grown under short day conditions. Scale bar = 5.5 cm. (C) Representative confocal microscopy images from the leaf epidermis of 40-day-old plants grown under short day conditions with the indicated genotypes. Images represent a Z-stack of 18 images taken every 1 µm. Arrows indicate some of the puncta structures formed when *AtMC1$^{C220A}$* is overexpressed in an *atmc1* mutant background. Scale bar = 10 µm. Source data are available online for this figure.

2017). Strikingly, we observed remarkable similarities in the phenotypes derived from expression of catalytically inactive caspase 8 (CASP8 CA) in mammals (Fritsch et al, 2019; Newton et al, 2019) and overexpression of catalytically inactive *At*MC1 in plants, both leading to some form of autoimmunity. Wt CASP8 participates in apoptotic and necroptotic cell death (Orning and Lien, 2021). Absence of CASP8 or loss of CASP8 catalytic activity results in embryonic lethality in mice (Fritsch et al, 2019; Newton et al, 2019). However, specific loss of CASP8 activity in mice epithelial cells induces intestinal inflammation as a result of aberrant activation of pyroptotic cell death (Fritsch et al, 2019). The authors showed a gene-dosage dependency in the phenotypes caused by inactive CASP8 and proposed that a distinct conformation in the protease compared to an active CASP8 may unmask the prodomain for interactions with components of the inflammasome (Fritsch et al, 2019; Newton et al, 2019). In our study, we find remarkable similarities in the structural requirements for the phenotype caused by catalytically inactive *At*MC1 compared to inactive CASP8 in mice. Overexpression of catalytically inactive *At*MC1 in a Wt background does not lead to autoimmunity (Fig. 4) in a similar way as Wt CASP8 alleles can suppress the inactive CASP8-dependent inflammatory phenotypes in mice (Fritsch et al, 2019). Besides, the N-terminal prodomain of CASP8 is required to engage cells into pyroptosis through binding to ASC specks (Fritsch et al, 2019; Newton et al, 2019). Similarly, Arabidopsis transgenics overexpressing a prodomainless catalytically inactive *At*MC1 variant do not display the autoimmune phenotype observed in plants overexpressing full-length catalytically inactive *At*MC1 (Appendix Fig. S3; Fig. 4). Accordingly, this prodomainless variant is neither enriched in microsomal fractions nor localizes to puncta structures observed for full-length catalytically inactive *At*MC1 (Figs. 5 and 7; Appendix Fig. S8). We also showed that overexpression of non-cleavable *At*MC1 variants that carry point mutations either at the putative prodomain cleavage site (R49) or at the Ca$^{2+}$ binding site does not result in severe autoimmunity (Appendix Fig. S4) in a similar manner as non-cleavable mice CASP8 does not lead to inflammation (Tummers et al, 2020). Based on these similarities, it is tempting to speculate that although immune components and cell death pathways are not strictly conserved between plants and animals, structural conservation in the way these proteases fold may trigger similar phenotypic outputs. Therefore, inactive *At*MC1 might also favour a conformation in which the prodomain may serve as a docking site for protein-protein interactions that would otherwise not occur in an active *At*MC1 under basal conditions.

Interestingly, the prodomain of *At*MC2 also participates in immune regulation since its overexpression leads to autoimmune phenotypes (Wu et al, 2024). *At*MC2 prodomain-triggered autoimmunity is also dependent on the presence of SA, but in contrast to

*At*MC1 autoimmunity, it can be rescued by mutating BAK1/BKK1 or SOBIR1. Overexpression of *At*MC1 prodomain does not lead to autoimmunity, nor does expression of a catalytically inactive version of AtMC2 (Fig. EV3 and (Wu et al, 2024)). This could indicate that while the prodomain of both Type I metacaspases *At*MC1 and *At*MC2 might serve as a docking site for immune regulators, although they might target different signalling nodes and their functions and mode of action may have diverged over time.

## Catalytically inactive AtMC1 as a docking site of immune-related protein complexes

Our proteomic analyses comparing interactors of catalytically inactive *At*MC1 when expressed in either an *atmc1* background (autoimmunity) or a Wt (no autoimmunity) background suggested that *At*MC1$^{C220A}$ interacts promiscuously with proteins involved in plant defence exclusively in plants exhibiting autoimmunity (Fig. 5). In planta co-IPs corroborated the ability of inactive *At*MC1 to co-immunoprecipitate with sNLRs (RPS2 and SSI4 (AT5G41750), PRRs (RLP42 and SOBIR1) or other immune-related components (RBOHF) (Figs. 6 and EV1). Absence of these interactors in the IP-MS experiment when plants express catalytically inactive *At*MC1 in a Wt background (Wt *AtMC1$^{C220A}$-GFP*) (Fig. 6), may imply that Wt *At*MC1 can compete for binding with defence-related interactors in these plants through more transient interactions, possibly participating in their homeostatic regulation (cleavage/disaggregation/clearance), thus preventing inactive *At*MC1 from stabilizing NLRs, PRRs or other defence-related interactors. This function would be in line with the previously reported pro-survival role of AtMC1 as regulator of protein clearance during proteotoxic stress situations (heat, senescence) (Ruiz-Solaní et al, 2023), which has also been reported for yeast MC1 (Mca1) (Eisele-Bürger et al, 2023; Hill et al, 2014; Lee et al, 2010). Interestingly, Mca1 can act as a molecular co-chaperone within protein aggregates, aiding in their clearance and promoting pro-survival/anti-aging mechanisms (Eisele-Bürger et al, 2023). Whether AtMC1 features a co-chaperone function in addition to its reported disaggregase activity remains to be clarified. Considering that our interpretation is based on co-immunoprecipitation data and we have not performed assays to test direct binding, we cannot rule out the possibility that the interaction of *AtMC1$^{C220A}$-GFP* or *AtMC1-GFP* with immune-related components is indirect.

## The autoimmune phenotype caused by catalytically inactive AtMC1 is dependent on a salicylic acid-mediated feedback loop and hyperaccumulation of NLRs

Introducing individual second-site mutations on *At*MC1$^{C220A}$ interactors or transgenesis of an almost-complete catalogue of

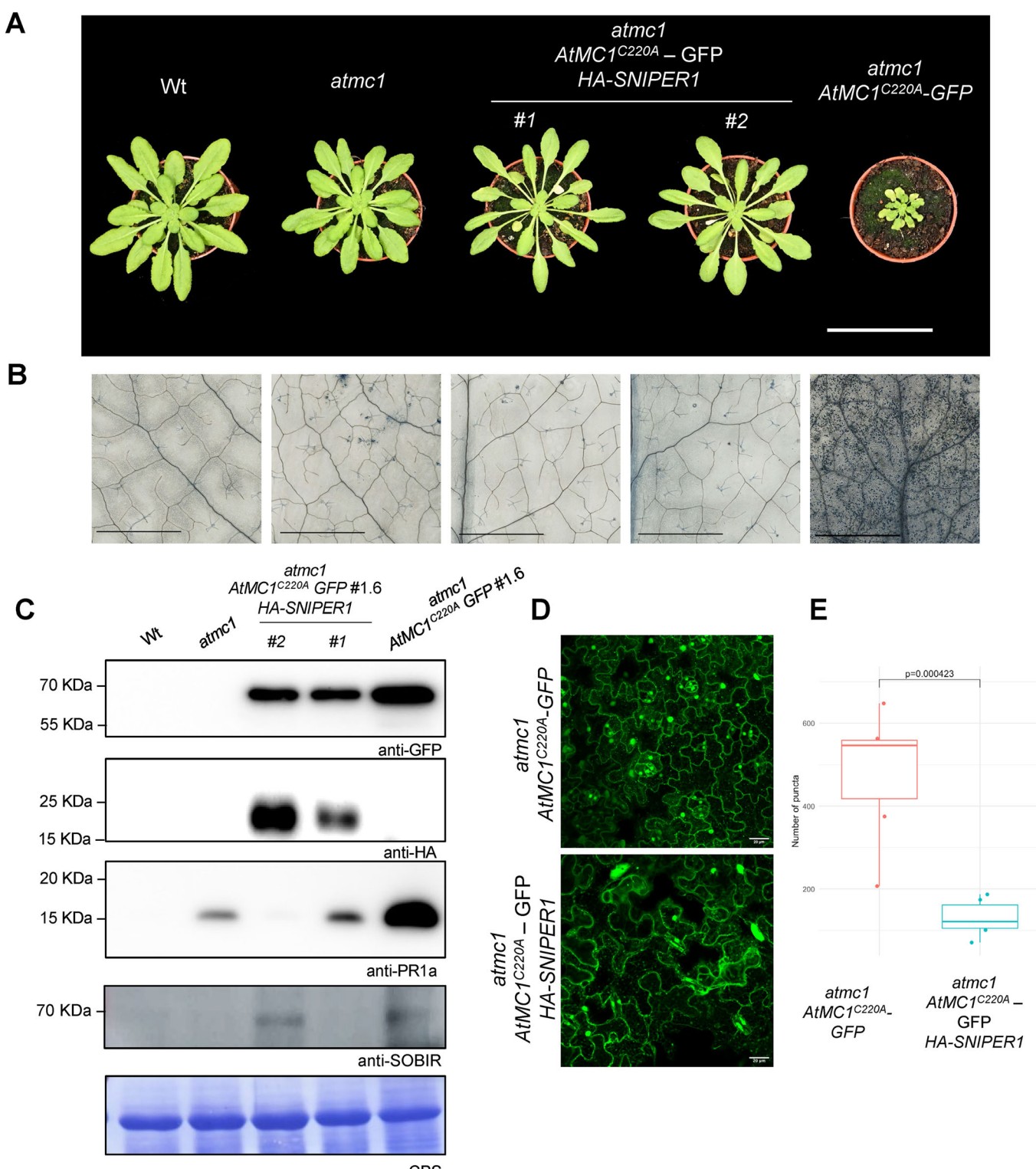

Arabidopsis DN-NLRs (139 DN-NLRs out of 166 NLRs present in Arabidopsis) into the autoimmune background (*atmc1 AtMC1<sup>C220A</sup>*-GFP) did not rescue the autoimmune phenotype

Arabidopsis DN-NLRs (139 DN-NLRs out of 166 NLRs present in Arabidopsis) into the autoimmune background ($atmc1$ $At$MC1$^{C220A}$-GFP) did not rescue the autoimmune phenotype (Appendix Fig. S7 and Appendix Table S1). Therefore, our data argues against the hypothesis that $At$MC1 or perhaps its catalytic activity could be guarded by a single NLR. Interestingly, introducing mutations in EDS1, PAD4 or the ADR1 gene family partially rescued the severe autoimmune phenotype, whereas second-site mutations in SAG101 and the NRG1 gene family did not result in phenotypic differences compared to the autoimmune plant (Fig. 3; Appendix Fig. S5; Fig. EV4). Suppression of SA synthesis, however, caused an almost complete rescue, abolishing

**Figure 9. Overexpression of the E3 ubiquitin ligase SNIPER1 rescues the autoimmune phenotype caused by catalytically inactive AtMC1.**

(A) Representative images of 40-day-old plants with the indicated phenotypes grown under short day conditions. Two independent stable transgenics in the T2 generation expressing HA-SNIPER1 (#1 and #2) under the control of a 35S constitutive promoter in the atmc1 AtMC1$^{C220A}$-GFP background are shown. Scale bar = 5.5 cm. (B) Trypan blue staining of an area belonging to the 6th true leaf of the plants shown in (A). Scale bar = 1.25 mm. (C) Total protein extracts from the plants shown in (A) were run on an SDS-PAGE gel and immuno-blotted against the indicated antisera. Comassie Blue staining (CBS) of the immunoblotted membranes shows protein levels of Rubisco as a loading control. (D) Representative confocal microscopy images from the leaf epidermis of 40-day-old plants grown under short day conditions with the indicated genotypes. Images represent a Z-stack of 10 images taken every 1 μm. Scale bar = 20 μm. (E) Quantification of the number of puncta from six different plants of the indicated genotypes. The same number of cells were counted for each genotype and the puncta was counted using the SiCE spot detector Macro for ImageJ (n = 6). P-value was calculated using a two-tailed unpaired Student's t test. In box plot, the centre line indicates the median, the bounds of the box show the 25th and 75th percentiles, the whiskers indicate minimum to maximum values. Source data are available online for this figure.

PR1a protein accumulation and rescuing the growth defects of autoimmune plants to the levels of atmc1 mutants (Fig. 3). In Arabidopsis, ADR1s are required for full ETI triggered by TNLs and contribute, but are not strictly required, for ETI mediated by certain CNLs (Saile et al, 2020). NRG1s, on the other hand, are required for HR triggered by certain TNLs but do not have obvious functions during CNL-mediated HR and disease resistance (Castel et al, 2019; Saile et al, 2020). Given that all autoimmune genotypes that are TNL-mediated are fully dependent on EDS1 (Rodriguez et al, 2016), our genetic data suggest that CNLs, which can be either fully or partially EDS1-independent, might also contribute to the phenotype of atmc1 AtMC1$^{C220A}$-GFP plants (Fig. 5). We argue that the partial rescues observed when second-site mutations in EDS1, PAD4 and ADR1 are introduced might occur due to the interference with the SA-mediated feedback loop that goes into EDS1-PAD4-ADR1 to bolster ETI responses (Cui et al, 2017), therefore preventing amplification of the constitutive immune response taking place in autoimmune plants. Preventing SA synthesis by introducing mutations in ICS1 (sid2-1) almost completely rescued the phenotype but did not completely abolish cell death (Fig. 3).

Given that certain CNLs can act independently of SA synthesis and are Ca$^{2+}$ permeable channels on their own (Bi et al, 2021; Lewis et al, 2010) and that mixed-lineage kinase domain-like (MLKL) contributes to downstream steps in TNL by mediating Ca$^{2+}$ influx (Shen et al, 2024; Mahdi et al, 2020), it is tempting to speculate that autoimmunity in atmc1 AtMC1$^{C220A}$-GFP plants could be due to hyperactivation of a combination of SA-independent and SA-dependent NLRs that require the feedback loop through EDS1-PAD4-ADR1 to amplify the immune response (Cui et al, 2017; Lewis et al, 2010; Saile et al, 2020). In agreement with this, overexpressing the E3 ubiquitin ligase SNIPER1, which is a master regulator of sNLRs (both CNLs and TNLs) but not hNLRs (Wu et al, 2020b), in the autoimmune genetic background (atmc1 AtMC1$^{C220A}$-GFP x HA-SNIPER1) rescues the autoimmune phenotype and avoids puncta accumulation (Fig. 9).

## Stabilization of AtMC1$^{C220A}$ in aberrant condensates may reveal a new proteostatic role for AtMC1 in NLR and other immune regulators clearance mediated by autophagy

Whilst Wt AtMC1-GFP displays a diffuse nucleocytoplasmic localization in leaf epidermal cells, catalytically inactive AtMC1-GFP localizes to the nucleus, cytoplasm, and puncta structures (Fig. 7). We previously showed that upon proteotoxic stress AtMC1 is recruited into similar puncta structures that we could identify as

stress granules (Ruiz-Solaní et al, 2023). Here, we could not match the observed structures with hallmark stress granule markers, since we could not obtain any double transgenic plant bearing AtMC1$^{C220A}$-GFP and a stress granule marker despite numerous attempts. Still, taking into account the enhanced condensate-prone behaviour of AtMC1$^{C220A}$-GFP compared to AtMC1-GFP it is tempting to speculate that the observed puncta correspond to aberrant stress granules that have lost their dynamicity and remain in the nucleo-cytoplasmic space, perturbing proteostasis and generating chronic stress, a situation that has extensively been linked to disease in mammals (Cao et al, 2020; Mateju et al, 2017).

Additional evidence in support of this hypothesis is partial colocalization of part of the AtMC1$^{C220A}$-GFP condensate population with autophagosomes and their vacuolar targeting, indicating their potential autophagy-mediated degradation (Fig. 8) since that is the major pathway of stress granule disassembly (Buchan et al, 2013; Field et al, 2021; Hofmann et al, 2021; Jung et al, 2020; Mahboubi and Stochaj, 2017). The autophagosomal localization of AtMC1$^{C220A}$-GFP puncta is in line with the fractionation assays in which catalytically inactive AtMC1 localizes mainly to the microsomal fractions (Fig. 5). Introgression of atg2-1 and atg5-1 mutant alleles into the autoimmune background (atmc1 AtMC1$^{C220A}$-GFP) further exacerbates the observed autoimmune phenotype, implying that the inability to degrade AtMC1$^{C220A}$-GFP puncta through autophagy has detrimental effects for the plant (Fig. 8). These results are in line with the observation that atmc1 atg18 double mutant plants also display an exacerbated early senescence phenotype compared to the one observed in atmc1 mutant plants (Coll et al, 2014). Interestingly, the maize homologue of AtMC1, ZmMC1, was shown to recruit the CNL Rp1-D21 to nucleocytoplasmatic puncta, which also co-localized with autophagosome markers (Luan et al, 2021), pointing towards a potentially conserved mechanism of NLR homeostasis that would involve NLRs recruitment into condensates and disposal via autophagy. Stabilization of NLRs and other immune regulators in atmc1 (Roberts et al, 2013) or even more so in atmc1 AtMC1$^{C220A}$-GFP plants (Fig. 7) would lead to autoimmunity.

Interestingly, grasses (Poaceae) feature some Type I metacaspases where the catalytic cysteine has been substituted for a glutamic acid. The fact that non-catalytic metacaspases have been conserved through evolution reinforces the idea that these inactive versions serve other functions in the cell. In fact, the ortholog of AtMC1 in yeast, Mca1, has been recently shown to display dual biochemical activity, acting as a protease to cause cell death and as a co-chaperone, contributing to proteostasis and delaying aging (Eisele-Bürger et al, 2023). Higher plants, with multiple

metacaspases, may have preserved this functional duality either maintaining it in single proteins or they may have diversified these two functions into separate catalytic and non-catalytic versions of metacaspases, as observed in grasses.

The master regulator of plant immunity NPR1, which act as a E3 ligase adaptor, promotes cell survival by targeting substrates for ubiquitination and degradation through formation of SA-induced NPR1 condensates (SINCs) (Zavaliev et al, 2020). SINCs are enriched with NLRs and ETI signalling components and have been proposed to act as a hub in promoting cell survival upon stress (high SA concentration). Although *At*MC1 is not present in SINCs based on proteomics data, it is tempting to speculate that *At*MC1 serves a pro-life function in the context of fine-tuning plant immunity in parallel to SINCs and a broad NLR homeostatic function in parallel to SNIPER1. Still, direct recruitment of NLRs or other immune regulators by *At*MC1 into condensates upon pathogen infection to regulate proteostasis remains to be fully determined.

### Final remarks and working model

Based on the genetics of the autoimmune phenotype caused by catalytically inactive *At*MC1 and previous findings placing *At*MC1 as a negative regulator of an auto active hNLR variant (Roberts et al, 2013), we propose a model whereby *At*MC1 might directly or indirectly participate in the turnover of NLRs and perhaps other immune regulators via condensate formation contributing to their clearance through major degradative pathways such as autophagy and the proteasome (Fig. 10). In *atmc1* mutants condensates may lose dynamicity/clearance or simply not form, leading to slightly defective proteostasis and the observed mild autoimmune phenotype, which indicate that compensation by the many systems in place that exist in plants to ensure protein quality control may partly take over (Llamas et al, 2023), potentially including redundant functions played by other metacaspases (Wu et al, 2024). These condensates have been shown to be extremely important for returning the cell to a resting state after a stress that involves a certain degree of proteotoxicity caused by increased protein synthesis (Ruiz-Solaní et al, 2023), which may be the case of an immune response. Overexpression of a catalytically inactive *At*MC1 variant might exemplify a case in which immune components are trapped in otherwise very dynamic stress condensates. In *atmc1 AtMC1^{C220A}-GFP* plants aberrant stable condensates remain in the cytoplasm and nucleus and as a result defective turnover of NLRs and other immune components occurs. In this situation autophagy may be strongly activated to degrade excessive aberrant condensates through granulophagy, although it may not be sufficient and condensates remain and cause proteotoxicity, which may be one of the underlying causes of the strong autoimmunity observed in these plants. As a result, in the absence of autophagy, *atmc1 AtMC1^{C220A}-GFP* plants display even stronger autoimmunity, accompanied by extreme dwarfism and infertility. On the contrary, enhancing proteostasis through the upregulation of degradative pathways, exemplified by the expression of SNIPER1, could mitigate the observed autoimmunity. Investigating the molecular mechanisms underlying plant immune phenotypes can provide valuable knowledge about the systems in place to maintain NLR homeostasis. Since the field of plant immunity is gaining momentum with great advances in NLR bioengineering (Marchal et al, 2022), it is more than worth considering to also use NLR regulators as tools for engineering resistance.

## Methods

**Reagents and tools table**

| Reagent/Resource | Reference or Source | Identifier or Catalog Number |
|---|---|---|
| **Experimental models** | | |
| Arabidopsis thaliana seeds | | Table S4 |
| **Recombinant DNA** | | |
| Plasmids | This study | Table S5 |
| **Antibodies** | | |
| α-GFP-HRP | Miltenyi Biotec | Cat# 130-091-833, RRID:AB_247003 |
| α-HA-HRP | Roche | Cat# 12013819001, RRID:AB_390917 |
| α-cMyc | Sigma-Aldrich | Cat# M4439, RRID:AB_439694 |
| α-FLAG | Sigma-Aldrich | Cat# F7425, RRID:AB_439687 |
| α-PR1a | Agrisera | Cat# AS10 687, RRID:AB_10751750 |
| α-cAPXa | Agrisera | Cat# AS23 4940 |
| α-H+ATPase | Agrisera | Cat# AS07 260, RRID:AB_1031584 |
| α-SOBIR1 | Agrisera | Cat# AS16 3204 |
| α-BIK1 | Agrisera | Cat# AS16 4030 |
| **Oligonucleotides and other sequence-based reagents** | | |
| Oligonucleotides | This study | Table S6 |
| **Chemicals, Enzymes and other reagents** | | |
| Bsal-HF v2 | New England Biolabs | R3733S |
| T4 DNA ligase | New England Biolabs | M0202S |
| NZYTaq II 2x Green Master Mix polymerase | NZYTech | MB35801 |
| Q5 High Fidelity DNA polymerase | New England Biolabs | M0491S |
| Amersham ECL prime reagent | Cytiva | RPN2232 |
| SuperSignal West Pico PLUS | Thermo Fisher Scientific | 34580 |
| cOmplete, EDTA free protease inhibitor cocktail | Roche | 11873580001 |
| Bio-Rad Protein Assay Dye Reagent | Bio-Rad | #5000001 |
| µMACS Anti-GFP Isolation kit | Miltenyi Biotec | 130-091-125 |
| **Software** | | |
| RStudio | Posit | "Cranberry Hibiscus" 2024.09.1 Build 394 |
| SnapGene | Dotmatics | Version 7.0.3 |

| Reagent/Resource | Reference or Source | Identifier or Catalog Number |
|---|---|---|
| Perseus | (Tyanova et al, 2016) | v1.6.10.0 |
| MaxQuant | (Cox and Mann, 2008) | version 1.6.10.43 |
| gProfiler | (Kolberg et al, 2023) | version_e111_eg58_p18_f463989d |
| Panther classification system | Phoenix bioinformatics | v19.0 |
| Design & Analysis (DA2) | Thermo Fisher Scientific | |
| ImageJ | (Bayle et al, 2017) | |
| **Other** | | |
| Amersham Image-Quant 800 luminescent imager | GE Healthcare Life Sciences | |
| Z50 Digital Camera | Nikon | |
| FV1000 inverted confocal microscope | Olympus | |
| DM6 epifluorescent microscope | Leica | |
| QuantStudio 6 Pro RT-PCR | Thermo Fisher Scientific | |
| LightCycler 480 II | Roche | |
| Mastercycler X50 | Eppendorf | |
| Trans-Blot Turbo Transfer System | BioRad | |
| MACS MultiStand | Miltenyi Biotec | |
| Precision Scale | Mettler Toledo | |

## Plant materials and plant growth conditions

*Arabidopsis thaliana* Columbia-0 (Col-0) ecotype was used for all experiments performed in this study. Arabidopsis mutants and transgenic lines are listed in Appendix Table S4. All seeds were sown directly in soil. To explore visual phenotypes and quantify fresh weight of mutants and transgenic lines, plants were grown in a controlled chamber with a short-day photoperiod of 8 h light and 16 h dark for 40 days under 65% relative humidity and 22 °C. *N. benthamiana* plants were grown at a temperature ranging from 22 to 25 °C and a relative humidity of 65% under a long-day photoperiod of 16 h light 8 h dark. Sample randomization was applied to all the experiments, but no blinding was performed.

## Plasmid construction and generation of Arabidopsis transgenics

All constructs and primers used in this study are listed in Appendix Tables S5, S6, respectively. All plasmids were assembled using GreenGate cloning (Lampropoulos et al, 2013), except for *pro35S::SSI4 (AT5G41750)-3xHA*, *proSSI4::SSI4-mCitrine*,

*proSSI4::SSI4-mCherry-FLAG*, and the CRISPR destination vectors containing the RNA guides for *AtMC1* deletion, which were generated using Gateway cloning technology (Thermo Fisher Scientific). In the case of *pro35S::SSI4 (AT5G41750)-3xHA*, the genomic DNA sequence of *AtSSI4 (AT5G41750)* was introduced firstly into a pDONR207 by a BP reaction and subsequently introduced into a pGWB514 (Addgene #74856) binary destination vector by an LR reaction. For the *proSSI4::SSI4-mCitrine* and *proSSI4::SSI4-mCherry-FLAG*, the promoter region was first cloned following the TOPO® TA Cloning® Kit instructions (Thermo Fisher Scientific). All fragments were introduced by LR into a pB7m34GW destination vector (Karimi et al, 2007) using multisite gateway recombination system. For the *AtMC1* deletion (Fig. EV1), 20 bp of the targeted sequences of *AtMC1* (5′UTR, intron 3 and 3′UTR) neighbouring a PAM sequence, tracrRNA sequence, U6 promoter, restriction enzyme sequence sites (BamHI/PstI/SalI) for cloning and attB overhangs were ordered as gBlocks® from IDT. The 3 gBlock sequences were introduced individually into different pDONR207 vectors by BP reactions. For the combination of the three gRNAs, pDONR207 vectors containing the gRNAs were digested with restriction enzymes BamHI/PstI/SalI and ligated into a new pDONR207. Finally, the assembled gRNAs were transferred to the binary vector pDe-CAS9-DsRED (Morineau et al, 2017) by an LR reaction. For generation of *atmc1 #CR2 helperless/ AtMC1C220A-GFP #4.10*, we firstly introduced the transgene (*AtMC1C220A-GFP #4.10*) into the *helperless* background (*adr1, adr1-l1, adr1-l2, nrg1.1 nrg1.2*) and subsequently we caused a CRISPR deletion in the *AtMC1* endogenous Wt alleles (*atmc1-CR #2*, Fig. EV1). For generation of Arabidopsis transgenics, the *Agrobacterium tumefaciens* (ASE + pSOUP strain) floral dipping method was followed as previously described (Clough and Bent, 1998).

## Protein extraction and western blotting

Five hundred milligrams of leaf material were mixed with extraction buffer (50 mM HEPES pH 7.5, 150 mM NaCl, 0.5% Nonidet P-40, 10% [v/v] glycerol, 1 mM EDTA pH 8, 5 mM DTT and 1× cOmplete™ EDTA-free Protease Inhibitor Cocktail (Roche) or 50 mM HEPES pH 7.5, 50 mM NaCl, 10 mM EDTA pH 8.0, 0.5% [v/v] Triton X-100, 5 mM DTT, 1x Halt Protease Inhibitor Cocktail (Thermo Fisher)) in a 5/1 volume/weight ratio and centrifuged for 10 min at $10,000 \times g$ at 4 °C. Supernatants were supplemented with 1X SDS-loading dye and boiled at 95 °C before loading into an SDS-PAGE gel. Proteins were transferred to PVDF membranes (Roche) or nitrocellulose membranes (GE Healthcare Life Sciences) using the Trans-Blot Turbo Transfer System (Bio-Rad) following the manufacturer's instructions and immunoblotted against α-GFP-HRP (1:5000 Miltenyi Biotec Cat# 130-091-833, RRID:AB_247003), α-HA-HRP (1:5000 Roche Cat# 12013819001, RRID:AB_390917), mouse α-cMyc (1:10,000 Sigma-Aldrich Cat# M4439, RRID:AB_439694), rabbit α-FLAG (1:10,000 Sigma-Aldrich Cat# F7425, RRID:AB_439687), α-PR1a (dilution 1:10,000 Agrisera Cat# AS10 687, RRID:AB_10751750), α-cAPXa (dilution 1:5000, Agrisera Cat# AS23 4940), α-H+ATPase (dilution 1:5000 Agrisera Cat# AS07 260, RRID:AB_1031584), α-SOBIR1 (dilution 1:1000, Agrisera Cat# AS16 3204), α-BIK1 (1:3000 Agrisera Cat# AS16 4030). The ECL Prime Western Blotting Detection Reagent (Cytiva) was used for detection. Image

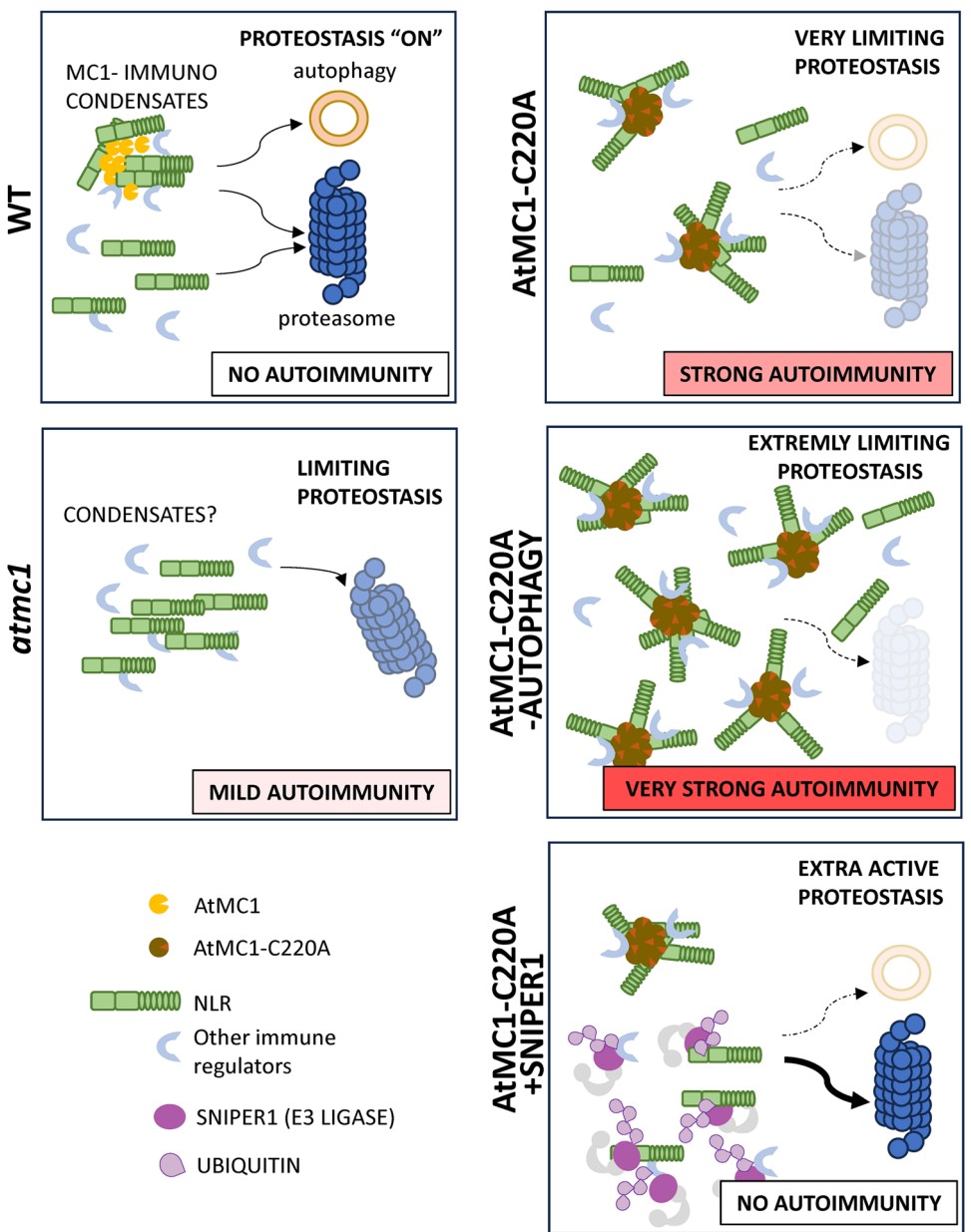

**Figure 10. Model of the role of *At*MC1 in plant autoimmunity.**

In adult Wt plants (left, upper panel), active *At*MC1 contributes to the maintenance of steady-state levels of NLRs likely by promoting the formation of transient condensates that are degraded by autophagy. In addition, NLRs are also degraded by the proteasome. In *atmc1* mutants (left, bottom panel), MC1-immuno condensates are no longer formed and NLRs are mostly degraded via the proteasome. The amount of NLRs present in the cell slightly increases and produces a mild autoimmune phenotype. Overexpression of inactive AtMC1^C220A in *atmc1* mutant background (right, upper panel), produces the stabilization of aberrant MC1-immuno condensates that can only partially be removed by autophagy, thus producing and accumulation of NLRs and a strong autoimmune phenotype. When, autophagosome formation is in addition depleted in these plants (right, middle panel), high amounts of aberrant condensates containing NLRs are accumulated which produces a very strong autoimmune response. By contrast, when exacerbating the degradation of NLRs via proteasome by overexpression of the E3 ligase SNIPER1 (right, bottom panel), no autoimmune response is observed. In this situation, some aberrant condensates might still be formed but can be successfully degraded by autophagy.

acquisition was carried out with an Amersham™ Image-Quant 800 luminescent imager (GE Healthcare Life Sciences).

## Fractionation assays

Differential centrifugations were done to obtain total, soluble, and microsomal fractions from extracts of different plant genotypes.

Briefly, 2 g of aerial plant tissue from 40-day-old plants were homogenized in liquid nitrogen with mortar and pestle. Homogenization buffer (50 mM HEPES pH 7.5, 250 mM sucrose, 5 mM EDTA pH 8, cOmplete™ EDTA-free Protease Inhibitor Cocktail (Roche), 0.5% PVP-10 (Sigma) and 5 mM DTT) was added to the previously ground powder in a 5/1 volume/weight ratio. Subsequently, samples were left rotating in a rotator disc to reach

complete homogenization for 15 min at 4 °C. Extracts were filtered through two layers of miracloth (Merck Millipore) and subjected to a 15-min centrifugation at $8000 \times g$. The resulting supernatant (Total fraction, cytosolic and membrane proteins) was normalized by a Bradford Assay (BioRad) to ensure equal amount of protein was used before further fractionation. Adjusted extracts were centrifuged at $100,000 \times g$ for 1 h at 4 °C. The supernatant was designated at Soluble fraction (cytosolic proteins) and the resulting pellet dissolved in homogenization buffer without PVP-10 and supplemented with 1% Nonidet™ P40 (Sigma), was designated as Microsomal fraction (Total membranes). Total, soluble and microsomal fractions were supplemented with 1X SDS-loading dye and boiled at 65 °C before loading into an SDS-PAGE gel.

### Immunoprecipitation and mass spectrometry coupled to liquid chromatography (IP-MS)

Protein extraction to obtain Total and Microsomal protein fractions from aerial plant tissue of 40-day-old *atmc1 AtMC1*$^{C220A}$-*GFP*, Wt *AtMC1*$^{C220A}$-*GFP* and Wt *35S::GFP* plants was done as described in the fractionation assays section. Once fractions were obtained, extracts were incubated with anti-GFP magnetic beads (Miltenyi Biotec) for 2 h at 4 °C under constant rotation. Magnetic beads were immobilized on a magnetic separator (Miltenyi Biotec), washed 4 times with homogenization buffer and eluted with 1X elution buffer (4% SDS, 40 mM TCEP (Sigma), 160 mM CAM (Sigma) and 200 mM HEPES pH 7.5) previously heated to 90 °C.

For mass spectrometry analysis, samples were processed on an Orbitrap Fusin Lumos instrument (Thermo Fisher Scientific) coupled to an Easy-nLC 1200 liquid chromatography (LC) system. A fused silica capillary (75 μm × 46 cm) was used as analytical column with an integrated PicoFrit emitter (CoAnn Tech). The analytical column was encased by a Sonation column oven (PRSO-V2) and attached to nanospray flex ion source (Thermo Fisher Scientific) at 50 °C. The LC was equipped with two mobile phases: solvent A (0.1% (v/v) formic acid, FA, in water) and solvent B (0.1% FA in acetonitrile, ACN). All solvents were of UPLC grade (Sigma). Peptides were directly loaded onto the analytical column with a flow rate around 0.5–0.8 μL/min. Peptides were subsequently separated on the analytical column by running a 105 min gradient of solvent A and solvent B (start with 9% (v/v) B; gradient 9% to 35% B for 70 min; gradient 35% to 44% B for 15 min and 100% B for 20 min) at a flow rate of 250 nl/min. The mass spectrometer was set in the positive ion mode and operated using Xcalibur software (version 2.2 SP1.48). Precursor ion scanning was performed in the Orbitrap analyzer (FTMS; Fourier Transform Mass Spectrometry) in the scan range of $m/z$ 300–1500 and at a resolution of 240,000 with the internal lock mass option turned on (lock mass was 445.120025 $m/z$, polysiloxane) (Olsen et al, 2005).

RAW spectra were submitted to an Andromeda (Cox et al, 2011) search using MaxQuant (version 1.6.10.43) using the default settings label-free quantification (Cox et al, 2014). MS/MS spectra data were searched against the Uniprot reference proteome of Arabidopsis (UP000006548_3702.fasta; 39,350 entries) and a project specific database containing 2 sequences of interest (ACE_0662_SOI_v01.fasta; 2 entries). Further analysis and annotation of identified peptides was done in Perseus v1.6.10.0 (Tyanova et al, 2016). Only protein groups with at least two identified unique

peptides were considered for further analysis. For quantification, we combined related biological replicates to categorical groups and investigated only those proteins that were found in a minimum of one categorical group at least in 3 out of 4 biological replicas. Subsequently, peptides were visualized in Volcano plots comparing different categorical groups.

### Transient expression in *Nicotiana benthamiana*

Proteins of interest were transiently expressed by leaf infiltration of 4-week-old Wt *N. benthamiana* leaves together with the anti-silencing vector p19. *Agrobacterium tumefaciens* GV3101 strain harbouring the desired constructs was used. The final OD$_{600}$ of all bacterial suspension were adjusted to a final OD$_{600}$ of 0.3 in agroinfiltration buffer (10 mM MES, 10 mM MgCl$_2$ and 150 μM acetosyringone at pH 5.6). Tissue was harvested for sample processing 3 days post-infiltration.

### Co-immunoprecipitations (co-IPs)

For co-IPs, 400 mg of ground tissue were homogenized in IP homogenization buffer (50 mM HEPES pH 7.5, 150 mM NaCl, 1 mM EDTA pH 8, cOmplete™ EDTA-free Protease Inhibitor Cocktail (Roche), 0.5% PVP-10 (Sigma), 5 mM DTT and 0.5% Nonidet™ P40 (Sigma). Samples were left rotating in a rotator disc to reach complete homogenization for 15 min at 4 °C. Extracts were filtered through two layers of Miracloth (Merck Millipore) and subjected to a 15-min centrifugation at $10,000 \times g$. The resulting supernatant was normalized by a Bradford Assay (BioRad) and incubated with anti-GFP magnetic beads (Miltenyi Biotec) for 2 h at 4 °C under constant rotation. Magnetic beads were immobilized on a magnetic separator (Miltenyi Biotec), washed 4 times with IP homogenization buffer without PVP-10 and eluted with 1X SDS loading buffer (20 Mm Tris-HCl pH 7, 10% glycerol, 2% SDS, 0.1% Bromophenol blue and 100 mM DTT). Inputs (extracts before IP) diluted in 1X SDS loading buffer and IP samples were run on an SDS-PAGE gel to visualize proteins of interest through immuno-blotting.

### Pseudomonas syringae infection and growth assays

Whole leaves from 5-week-old *Arabidopsis* plants grown under short-day conditions (8 h light and 16 h dark) were infiltrated with *Pto DC3000* at OD$_{600}$ of 0.0005 or *Pro AvrRpt2* at OD$_{600}$ 0.002 using a 1-ml needleless syringe. Two leaf discs from two different leaves (7th and 8th leaf) were collected using a 6 mm-diameter cork borer (disc area, 0.282 cm²). Samples on day 0 and day 3 after infection were grounded in 10 mM MgCl$_2$ and serially diluted 5, 50, 500, 5000 and 50,000 times on a 96-well plate. Subsequently, dilutions were spotted (10 ml per spot) on King's B medium with antibiotics. The number of colony-forming units (CFUs) per drop was calculated and bacterial growth represented as log10 CFU per cm² of tissue.

### Fresh weight experiments

For quantification of fresh weight, the aerial part of Arabidopsis plants grown for 40 days under short day conditions were cut through the stem and weigh in a precision scale (Mettler Toledo).

## Confocal microscopy

Confocal imaging of proteins of interest was done using an Olympus FV1000 inverted confocal microscope with a x60/water objective. GFP signal was excited at 488 nm, whereas mRFP signal was excited at 543 nm. To visualize the vacuolar lumen, 1 μM Concanamycin A (Sigma) was syringe infiltrated with a needleless syringe. Control treatment was performed by infiltrated the corresponding volume of DMSO. Imaging was performed 24 h post-treatment. For *atmc1 35S::AtMC1$^{C220A}$-GFP 35S::HA-SNIPER1* and *atmc1 35S::AtMC1$^{C220A}$-GFP* puncta quantification, the SiCE spot detector Macro for ImageJ (Bayle et al, 2017) was used. Information on whether images are single-plane or Z-stacks is indicated in figure legends.

## Trypan blue staining

Ten Arabidopsis leaves per genotype were harvested in 50-ml Falcon tubes and incubated in 10 mL of a 1/3 dilution (trypan blue solution/ethanol) of trypan blue solution (100 mg Phenol solid, 100 mL lactic acid, 100 mL Glycerol and 100 mL water). Falcon tubes were submerged in boiling water for 10 min until leaves become completely blue. Subsequently, trypan blue solution was removed, and leaves were incubated with 10 mL of distaining solution (1 kg Chloral hydrate in 400 mL water) overnight on an orbital shaker. After removal of distaining solution, leaves were covered in 50% glycerol and photographed using a Leica DM6 epifluorescent microscope.

## Label-free quantification proteomics

Forty-day-old Arabidopsis leaf tissue was ground and homogenized in homogenization buffer (50 mM HEPES pH 7.5, 150 mM NaCl, 1 mM EDTA pH 8, 10% Glycerol, 5 mM DTT, 0.1% Nonidet™ P40 (Sigma), and cOmplete™ EDTA-free Protease Inhibitor Cocktail (Roche). The samples containing 100 μg of protein were analysed by LC-MS/MS using Ultimate 3000 RSLCnano system in-line connected to a Q Exactive HF mass spectrometer (Thermo Fisher Scientific). For full capture of MS and MS/MS events, resolutions of 60,000 and 15,000 were used, respectively. Full-scan MS data were acquired using a mass range of 375–1500 *m/z*. The peptides were separated on a 250 mm Aurora Ultimate, 1.7 μm C18, 75 μm inner diameter (Ionopticks) kept at a constant temperature of 45 °C. Peptides were eluted by a non-linear gradient starting at 1% MS solvent B reaching 33% MS solvent B (0.1% formic acid in water/acetonitrile (2:8, v/v)) in 75 min, 55% MS solvent B (0.1% formic acid in water/acetonitrile (2:8, v/v)) in 95 min, 70% MS solvent B in 100 min followed by a 5-min wash at 70% MS solvent B and re-equilibration with MS solvent A (0.1% formic acid in water).

Data-independent acquisition spectra were searched with the DIA-NN software (v1.8.1) in library-free mode against the reviewed Araport11_TAIR11 database (May 2018). The mass accuracy was set to 20 ppm and the MS1 accuracy to 10 ppm, with a precursor FDR of 0.01. Enzyme specificity was set to trypsin/P with a maximum of two missed cleavages. Variable modifications were set to oxidation of methionine residues (to sulfoxides) and acetylation of protein N-termini. Carbamidomethylation of cysteines was set as a fixed modification. The peptide length range was set to 7–30 residues with a precursor charge state between 1 and 4. The *m/z*

range was set between 400–900 and 200–1800 for the precursor and fragment ions, respectively. Cross-run normalization was performed with RT-dependent settings, where peptide quantification was adjusted based on retention time. The quantification strategy was set to high accuracy to ensure precise measurements. The neural network classifier was set to double-pass mode, enabling improved peptide identification accuracy. In subsequent data processing using R, non-proteotypic peptides lacking characteristic properties were removed to focus on relevant results. In addition, peptide identifications with a Lib.Q-value below 0.01, indicating low confidence, were filtered out to ensure reliable outcomes. The remaining peptide quantifications were then aggregated into protein group quantifications using the median of their corresponding normalized LFQ values.

## Data availability

The mass spectrometry proteomics data for the IP-MS/MS on-bead digestions and label-free quantification (LFQ) have been deposited to the ProteomeXchange Consortium via the PRIDE (Vizcaíno et al, 2014) partner repository (https://www.ebi.ac.uk/pride/archive/) with the following dataset identifiers: PXD049206 and PXD048924.

The source data of this paper are collected in the following database record: biostudies:S-SCDT-10_1038-S44319-025-00426-4.

## Peer review information

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

## Acknowledgements

We would like to thank T. Nürnberger for pro35S::10xMyc-AtRLP42 and pro35S::10xMyc-AtSOBIR1 constructs, C. Zipfel for the pro35S::FLAG-AtRBOHF construct, Y. Dagdas for the proUBQ::mCherry-ATG8a construct and D. Hofius for sharing atg mutant seeds. We would like to thank Ignacio Rubio-Somoza for helpful comments and fruitful discussions on the project and Xin Li and Yuelin Zhang for critically reading the manuscript. We also acknowledge the core facilities that have been instrumental for this work: CRAG Plant Growth facility, CRAG Microscopy and Imaging facility and the Analytics Core Facility Essen. Research at CRAG was supported by grants PID2022-136922NB-I00 and PID2019-108595RB-I00 funded by MCIN/AEI/10.13039/501100011033 and TED2021-131457B-I00 funded by MCIN/AEI/10.13039/501100011033 and by the "European Union NextGenerationEU/PRTR" (to NSC and MV); also was supported by fellowships: Predoctoral fellowship BES-2017-080210 funded by MCIN/AEI/10.13039/501100011033 and by "ESF Investing in your future" (to JS-L); predoctoral fellowship FPU19/03778 funded by MU (o Ministerio de Universidades) (to NR-S), predoctoral fellowship PREP2022-000557 funded by MCIN/AEI/10.13039/501100011033 and by the FSE+" (to JA), postdoctoral fellowship FJC2021-046667-I funded by MCIN/AEI/10.13039/501100011033 and by the "European Union NextGenerationEU/PRTR" (to FNG). This project has received funding from the European Union's Horizon 2020 research and innovation programme under the Marie Skłodowska-Curie grant agreement No 945043 through the postdoctoral fellowship awarded to FNG by the AGenT H2020-MSCA-COFUND-2019 programme. MS-G is a recipient of the predoctoral fellowship HORIZON-MSCA-2021-COFUND rePLANT-GA101081581 Funded by the European Union. LA is a recipient of the postdoctoral fellowship HORIZON-MSCA-2021-PF-ImmunoZoneHubs-GA#101068121 funded by the European Union. Views and opinions expressed are, however, those of the author(s) only and do not necessarily reflect those of the European Union or the EUROPEAN RESEARCH EXECUTIVE AGENCY (REA). Neither the European Union nor the granting authority can be held responsible for them. AGenT and rePLANT MSCA-COFUND programmes are cofunded by the Severo Ochoa Programme for Centres of Excellence in R&D CEX2019-000902-S funded by MCIN/AEI/10.13039/501100011033; and by the CERCA Programme/Generalitat de Catalunya. We acknowledge support by the German Research Foundation (DFG) CRC1101 D09 and Reinhard-Frank Stiftung (project: helperless plant) (to FEK).

## Author contributions

**Jose Salguero-Linares**: Conceptualization; Resources; Funding acquisition; Validation; Investigation; Visualization; Methodology; Writing—original draft; Writing—review and editing. **Laia Armengot**: Conceptualization; Formal analysis; Funding acquisition; Investigation; Methodology; Writing—review and editing. **Joel Ayet**: Resources; Data curation; Formal analysis; Funding acquisition; Validation; Investigation; Visualization; Methodology; Writing—original draft; Writing—review and editing. **Nerea Ruiz-Solaní**: Investigation; Methodology; Writing—review and editing. **Svenja C Saile**: Investigation; Methodology; Writing—review and editing. **Marta Salas-Gómez**: Investigation; Writing—review and editing. **Esperanza Fernandez**: Investigation; Methodology; Writing—review and editing. **Lode Denolf**: Investigation; Methodology; Writing—review and editing. **Fernando Navarrete**: Investigation; Writing—review and editing. **Jenna Krumbach**: Investigation; Writing—review and editing. **Markus Kaiser**: Supervision; Funding acquisition; Methodology; Writing—review and editing. **Simon Stael**: Investigation; Methodology; Writing—review and editing. **Frank Van Breusegem**: Funding acquisition; Investigation; Methodology; Writing—review and editing. **Kris Gevaert**: Funding acquisition; Methodology; Writing—review and editing. **Farnusch Kaschani**: Investigation; Methodology; Writing—review and editing. **Morten Petersen**: Conceptualization; Investigation; Methodology; Writing—review and editing. **Farid El Kasmi**: Conceptualization; Investigation; Methodology; Writing—review and editing. **Marc Valls**: Conceptualization; Supervision; Funding acquisition; Writing—review and editing. **Núria S Coll**: Conceptualization; Supervision; Funding acquisition; Writing—original draft; Project administration; Writing—review and editing.

Source data underlying figure panels in this paper may have individual authorship assigned. Where available, figure panel/source data authorship is listed in the following database record: biostudies:S-SCDT-10_1038-S44319-025-00426-4.

## Disclosure and competing interests statement

The authors declare no competing interests.

# Expanded View Figures

**Figure EV1.  Absence of AtMC1 results in mild autoimmunity.**

(**A**) Schematic representation of *atmc1* CRISPR mutants, depicting the locations of the two guide RNAs (gRNAs) targeting the AtMC1 gene. The location of the T-DNA insertion of the T-DNA *atmc1* mutant is indicated with an orange triangle. Below, Sanger sequencing chromatograms showing the site of deletion of *atmc1-CR#1* and *atmc1-CR#2*. (**B**) Representative image of 40-day-old Wt, *atmc1* and *atmc1-CR#1* plants grown under short day conditions. Scale bar = 5.5 cm. (**C**) Trypan blue staining of an area belonging to the 6th true leaf of the plants shown in (**A**). Scale bar = 0.5 mm. (**D**) Plant fresh weight of genotypes shown in (**A**) ($n = 12$). Different letters indicate statistical difference in fresh weight between genotypes (one-way ANOVA followed by post hoc Tukey, $p$ value < 0.05). In box plot, the centre line indicates the median, the bounds of the box show the 25th and 75th percentiles, the whiskers indicate minimum to maximum values. (**E**) Total protein extracts from the plant genotypes shown in (**A**) were run on an SDS-PAGE gel and immuno-blotted against anti-PR1a. Coomassie Blue Staining (CBS) of the immunoblotted membranes shows protein levels of Rubisco as a loading control. (**F**) Bacterial growth on the indicated genotypes 3 days post-infection with virulent *Pseudomonas syringae* DC3000 strain. Different letters indicate statistical difference (one-way ANOVA followed by post hoc Tukey, $p$ value < 0.05) ($n = 14$). In box plot, the centre line indicates the median, the bounds of the box show the 25th and 75th percentiles, the whiskers indicate minimum to maximum values. (**G**) Bacterial growth on the indicated genotypes 3 days post-infection with avirulent *Pseudomonas syringae* AvrRpt2 strain ($n = 11$ biological replicates). *P*-value was calculated using a two-tailed unpaired Student's t test. In box plot, the centre line indicates the median, the bounds of the box show the 25th and 75th percentiles, the whiskers indicate minimum to maximum values.

▶

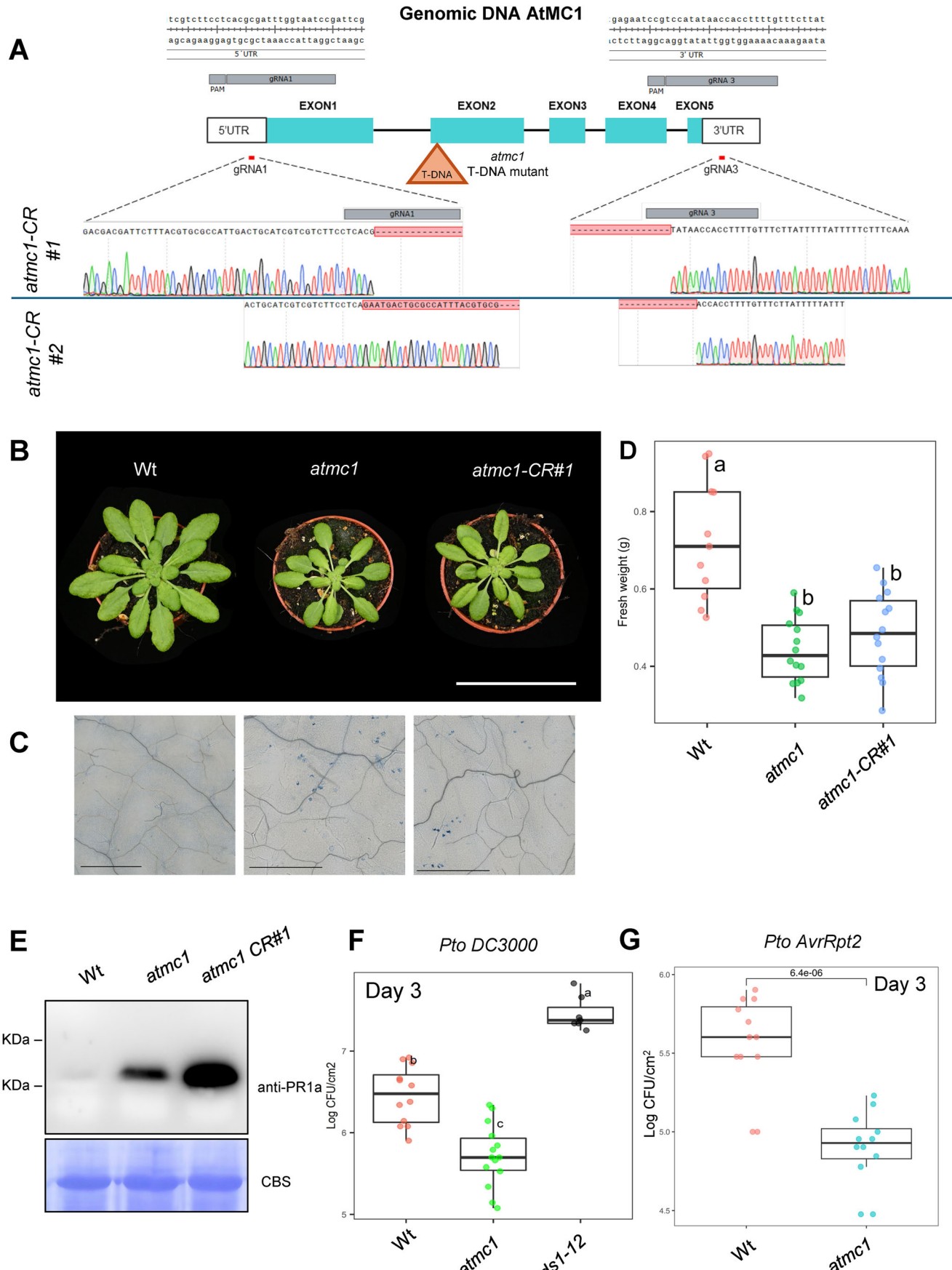

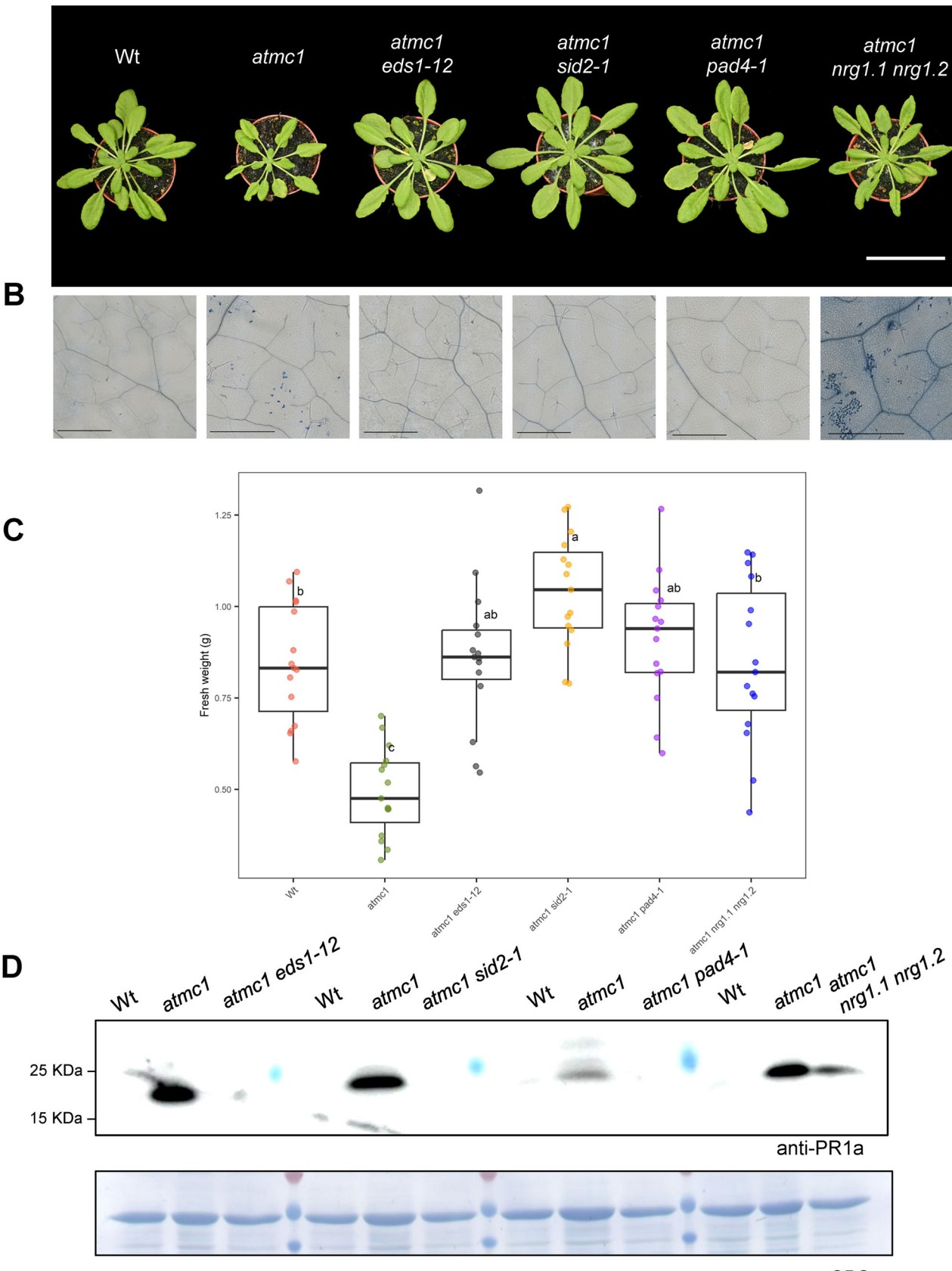

**Figure EV2. Constitutive immune activation in *atmc1* plants is dependent on SA synthesis and immune signalling through EDS1-PAD4.**

(A) Representative image of 40-day-old Wt, *atmc1*, *atmc1 eds1-12*, *atmc1 sid2-1*, *atmc1 pad4-1* and *atmc1 nrg1.1 nrg1.2* grown under short day conditions. Scale bar = 5.5 cm. (B) Trypan blue staining of an area belonging to the 6th true leaf of the plants shown in (A). Scale bar = 0.5 mm. (C) Plant fresh weight of genotypes shown in (A) ($n = 15$). Different letters indicate statistical difference in fresh weight between genotypes (one-way ANOVA followed by post hoc Tukey, $p$ value < 0.05). In box plot, the centre line indicates the median, the bounds of the box show the 25th and 75th percentiles, the whiskers indicate minimum to maximum values. (D) Total protein extracts from the plant genotypes shown in (A) were run on an SDS-PAGE gel and immuno-blotted against anti-PR1a. Comassie Blue Staining (CBS) of the immunoblotted membranes shows protein levels of Rubisco as loading control.

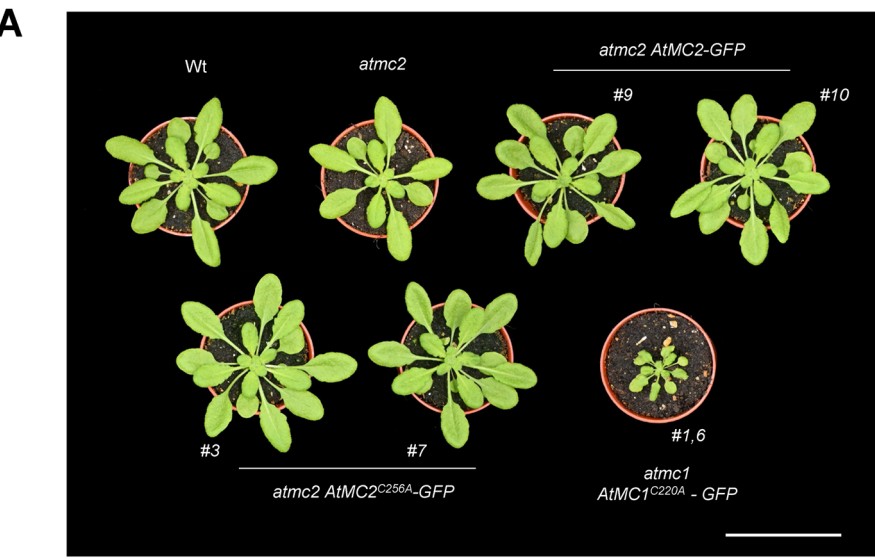

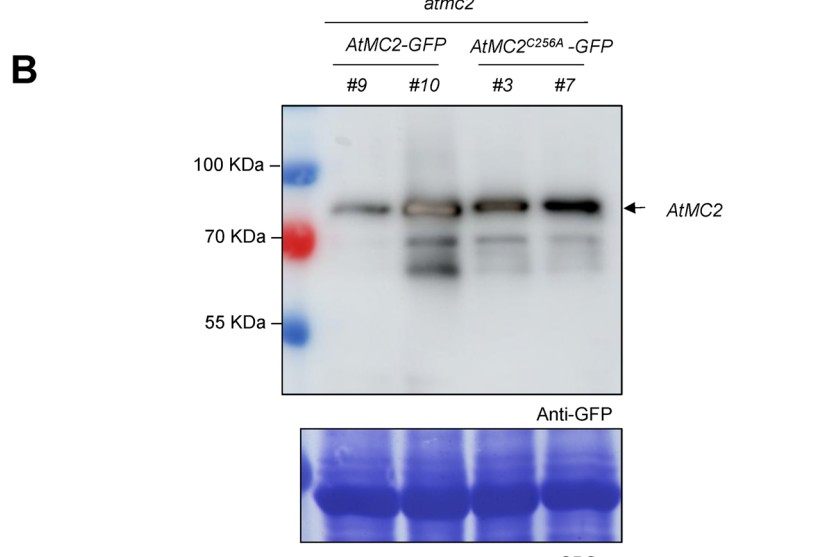

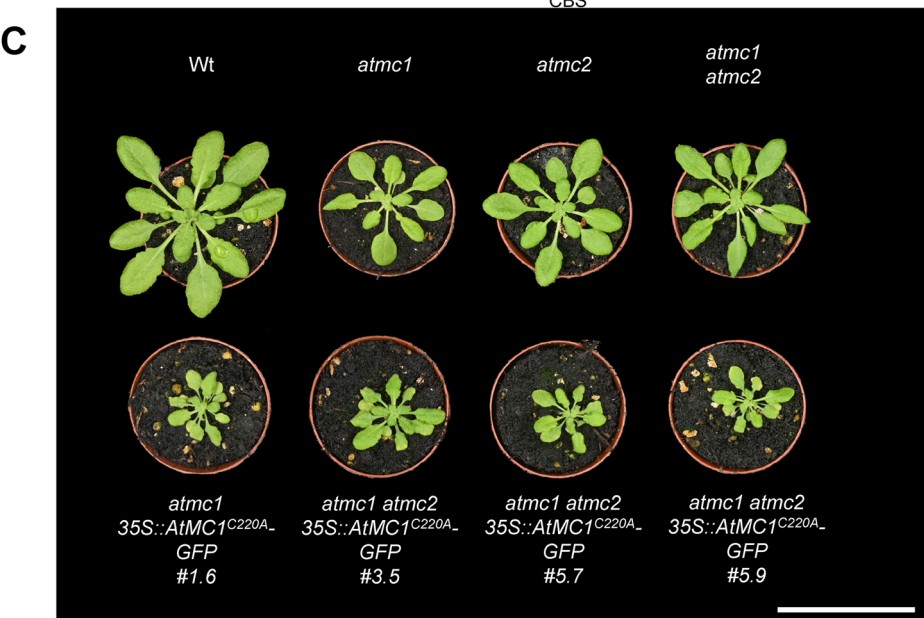

**Figure EV3. Overexpression of a catalytically inactive *AtMC2* in an *atmc2* mutant background does not cause autoimme phenotype.**

(**A**) Representative images of 40-day-old plants with the indicated genotypes grown under short day conditions. Two independent stable transgenic plants of the $T_2$ generation expressing either *AtMC2-GFP* (#9 and #10) or *AtMC2$^{C256A}$-GFP* (#3 and #7) under the control of a 35S constitutive promoter in the *atmc2* mutant background are shown. Scale bar = 5.5 cm. (**B**) Total protein extracts from the plant genotypes shown in (**A**) were run on an SDS-PAGE gel and immuno-blotted against with the indicated antisera. Comassie Blue Staining (CBS) of the immunoblotted membranes shows protein levels of Rubisco as a loading control. (**C**) Representative images of 40-day-old plants with the indicated genotypes grown under short day conditions. Three independent stable transgenic lines expressing *AtMC1$^{C220A}$-GFP* under the control of a 35S constitutive promoter in the *atmc1 atmc2* mutant backgrounds are shown. Scale bar = 5.5 cm.

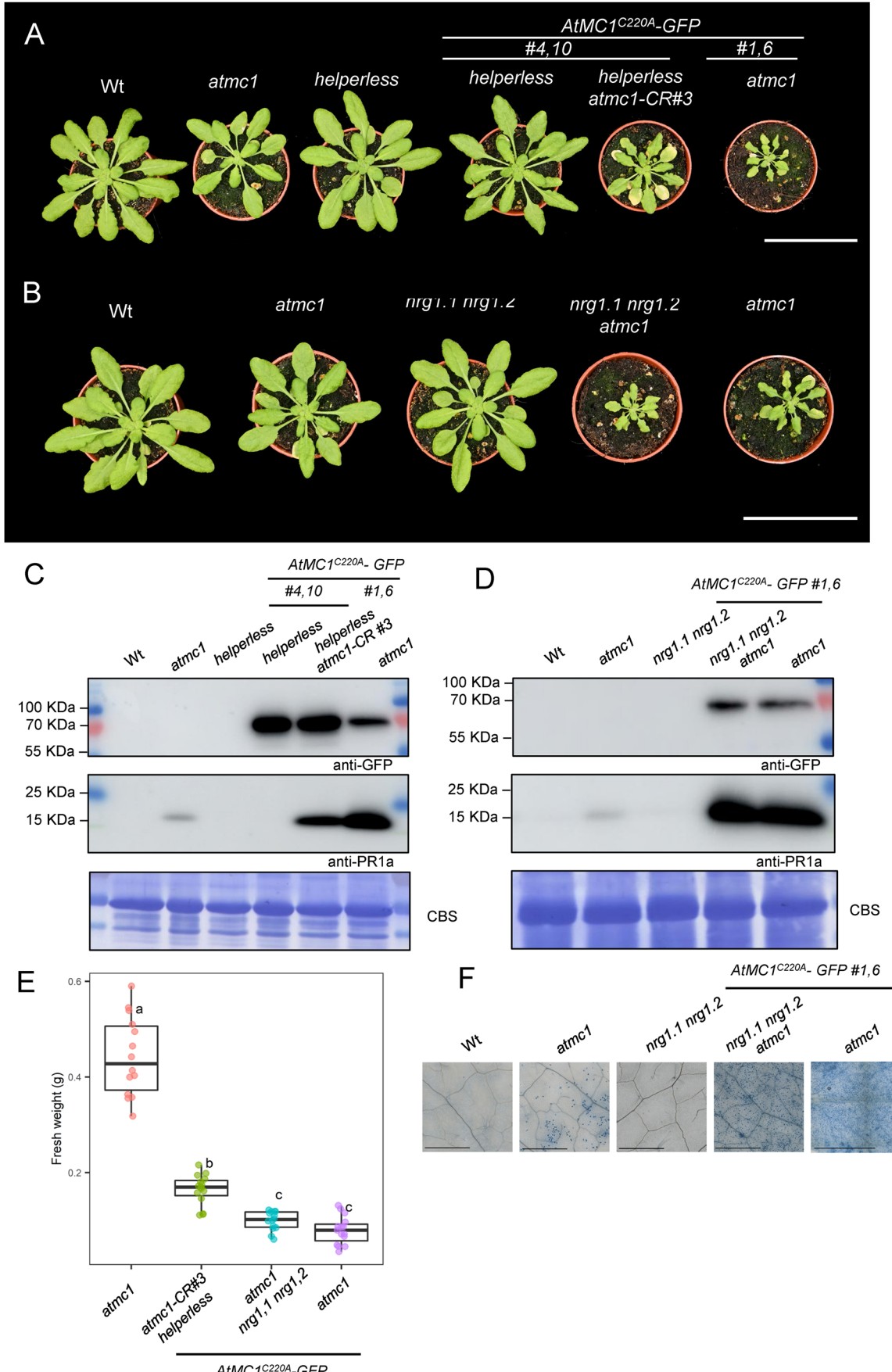

**Figure EV4. Autoimmunity caused by catalytically inactive *At*MC1 is partially dependent on the hNLR family ADR1 but not NRG1.**

(A, B) Representative images of 40-day-old plants with the indicated genotypes grown under short day conditions. Scale bar = 5.5 cm. (C, D) Total protein extracts from the plant genotypes shown in (A, B) were run on an SDS-PAGE gel and immuno-blotted against the indicated antisera. Comassie Blue Staining (CBS) of the immunoblotted membranes shows protein levels of Rubisco as a loading control. (E) Plant fresh weight of the indicated genotypes ($n = 12$). Different letters indicate statistical difference in fresh weight between genotypes (one-way ANOVA followed by post hoc Tukey, $p$ value < 0.05). In box plot, the centre line indicates the median, the bounds of the box show the 25th and 75th percentiles, the whiskers indicate minimum to maximum values. Quantification of fresh weight from Wt, *helperless, helperless/ AtMC1*$^{C220A}$-*GFP* and *nrg1.1 nrg1.2* were excluded from the fresh weight graph to better appreciate statistical differences between genotypes of interest. (F) Trypan blue staining of an area belonging to the 6th true leaf of the plants shown in (B). Scale bar = 0.5 mm.

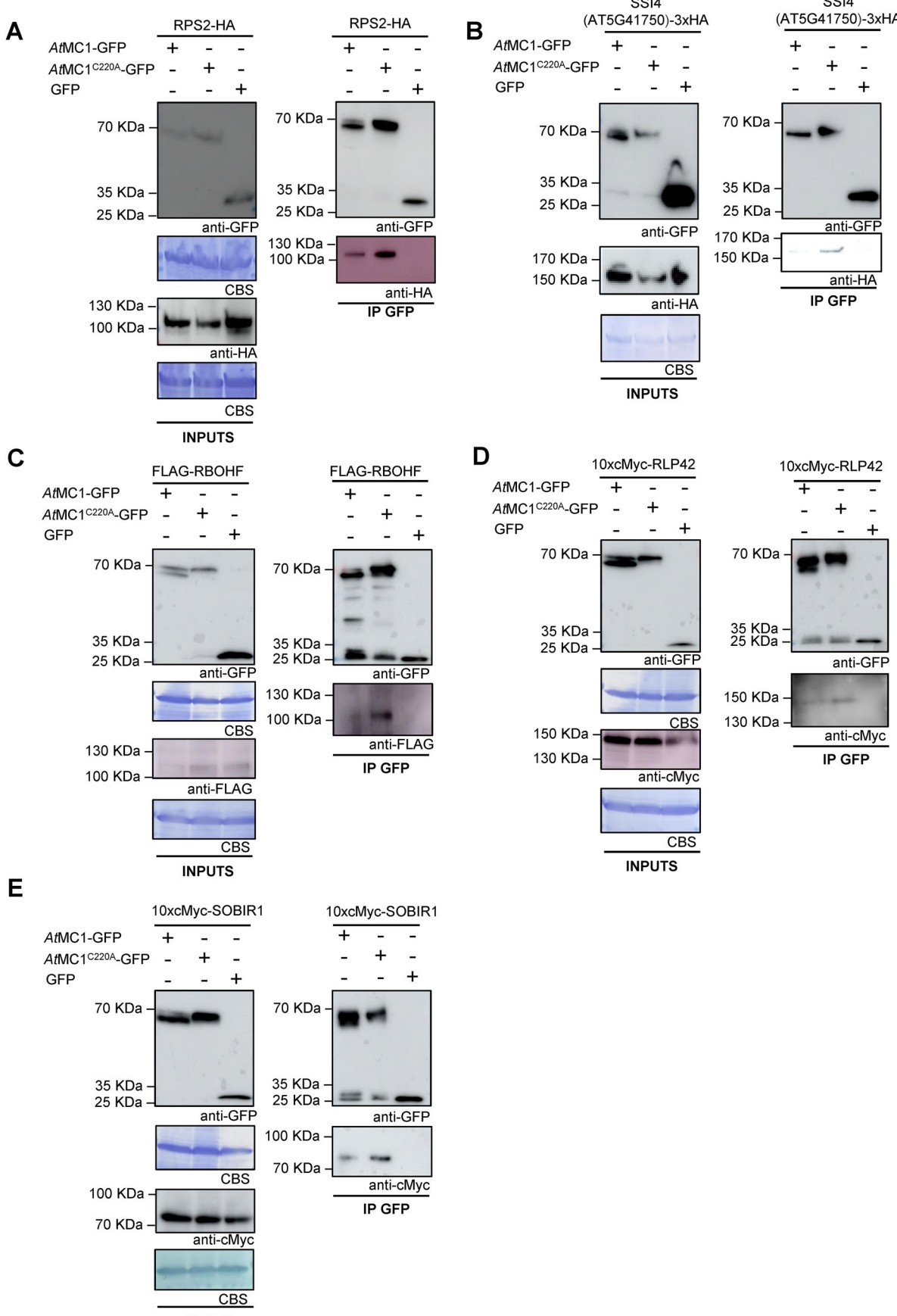

◀   **Figure EV5. Catalytically inactive *At*MC1 interacts in planta with NLRs, and immune components involved in PTI.**

(A–E) *AtMC1-GFP*, *AtMC1^{C220A}-GFP* or free GFP were transiently co-expressed with either RPS2-HA (**A**), SSI4-3xHA (**B**), FLAG-RBOHF (**C**), 10xcMyc-RLP42 (**D**) or 10xcMyc SOBIR1 (**E**) in *N. benthamiana*. 3 days post-infiltration (dpi) plant extracts co-expressing the indicated constructs were immunoprecipitated with anti-GFP magnetic beads (IP GFP). Protein inputs from protein extracts before IP (INPUTS) and eluates from IPs were run on an SDS-PAGE and immunoblotted against the indicated antisera. Coomassie Blue Staining (CBS) of the immunoblotted membranes shows protein levels of Rubisco as a loading control in the inputs.

                                                                              