## [Peer Review File · EMBO Reports]

Lack of AtMC1 catalytic activity triggers autoimmunity dependent on NLR stability

Jose Salguero-Linares, Laia Armengot, Joel Ayet, Nerea Ruiz-Solaní, Svenja Saile, Marta Salas-Gómez, Esperanza Fernandez, Lode Denolf, Fernando Navarrete, Jenna Krumbach, Markus Kaiser, Simon Stael, Frank Van Breusegem, Kris Gevaert, Farnusch Kaschani, Morten Petersen, Farid El Kasmi, Marc Valls, and Nuria Coll

Corresponding author(s): Nuria Coll (nuria.sanchez-coll@cragenomica.es)

Review Timeline:

Submission Date:	2nd Sep 24
Editorial Decision:	25th Oct 24
Revision Received:	31st Jan 25
Editorial Decision:	28th Feb 25
Revision Received:	6th Mar 25
Accepted:	10th Mar 25

Editor: Achim Breiling

Transaction Report:

Dear Dr. Coll,

Thank you for the transfer of your manuscript to EMBO reports. I have now received the reports from the three referees that were asked to evaluate your study, which can be found at the end of this email.

As you will see, the referees find the study interesting. Nevertheless, they have several comments, concerns, and suggestions, indicating that a revision of the manuscript is necessary to allow publication of the study in EMBO reports. As the reports are below, and all the concerns need to be addressed, I will not detail them further here.

Given the constructive referee comments, I would like to invite you to revise your manuscript with the understanding that the concerns of the referees must be addressed in the revised manuscript and in a detailed point-by-point response. Acceptance of your manuscript will depend on a positive outcome of a second round of review. It is EMBO reports policy to allow a single round of revision only and acceptance of the manuscript will therefore depend on the completeness of your responses included in the next, final version of the manuscript.

- 1) a .docx formatted version of the final manuscript text (including legends for main figures, EV figures and tables), but without the figures included. Figure legends should be compiled at the end of the manuscript text.
- 2) individual production quality figure files as .eps, .tif, .jpg (one file per figure), of main figures and EV figures. Please upload these as separate, individual files upon re-submission.

- 4) a complete author checklist, which you can download from our author guidelines (<https://www.embopress.org/page/journal/14693178/authorguide>). Please insert page numbers in the checklist to indicate where the requested information can be found in the manuscript. The completed author checklist will also be part of the RPF.

- 5) that primary datasets produced in this study (e.g. RNA-seq, ChIP-seq, structural and array data) are deposited in an

appropriate public database. If no primary datasets have been deposited, please also state this in a dedicated section (e.g. 'No primary datasets have been generated and deposited'), see below.

The accession numbers and database should be listed in a formal "Data Availability" section that follows the model below. This is now mandatory (like the COI statement). Please note that the Data Availability Section is restricted to new primary data that are part of this study. This section is mandatory. As indicated above, if no primary datasets have been deposited, please state this in this section

Data availability

8) Regarding data quantification and statistics, please make sure that the number "n" for how many independent experiments were performed, their nature (biological versus technical replicates), the bars and error bars (e.g. SEM, SD) and the test used to calculate p-values is indicated in the respective figure legends (also for EV and Appendix figures). Please also check that all the p-values are explained in the legend, and that these fit to those shown in the figure. Please provide statistical testing where applicable. Please avoid the phrase 'independent experiment', but clearly state if these were biological or technical replicates. Please also indicate (e.g. with n.s.) if testing was performed, but the differences are not significant. In case n=2, please show the data as separate datapoints without error bars and statistics. See also: <http://www.embopress.org/page/journal/14693178/authorguide#statisticalanalysis>

9) Please add scale bars of similar style and thickness to all microscopic images, using clearly visible black or white bars (depending on the background). Please place these in the lower right corner of the images themselves. Please do not write on or near the bars in the image but define the size in the respective figure legend.

10) Please also note our reference format:

12) We now use CRedit to specify the contributions of each author in the journal submission system. CRedit replaces the author contribution section. Please use the free text box to provide more detailed descriptions and do NOT provide your final manuscript text file with an author contributions section. See also our guide to authors: <https://www.embopress.org/page/journal/14693178/authorguide#authorshipguidelines>

13) All Materials and Methods need to be described in the main text using our 'Structured Methods' format, which is required for

all research articles. According to this format, the Methods section should include a Reagents and Tools Table (listing key reagents, experimental models, software, and relevant equipment and including their sources and relevant identifiers), uploaded as separate file, followed by a Methods section in which we encourage the authors to describe their methods using a step-by-step protocol format with bullet points, to facilitate the adoption of the methodologies across labs. More information on how to adhere to this format as well as downloadable templates (.doc) for the Reagents and Tools Table can be found in our author guidelines (section 'Structured Methods'):

14) Please order the manuscript sections like this, using these names:

Title page - Abstract - Keywords - Introduction - Results - Discussion - Methods - Data availability section - Acknowledgements (including funding information) - Disclosure and Competing Interests Statement - References - Figure legends - Expanded View Figure legends

15) Please make sure that all the funding information is also entered into the online submission system and that it is complete and similar to the one in the acknowledgement section of the manuscript text file.

I look forward to seeing a revised form of your manuscript when it is ready.

Yours sincerely,

Referee #1:

Overall, this manuscript offers novel insights into how metacaspases safeguard the plant immune system. The authors propose a model in which AtMC1 regulates NLR protein levels and prevents autoimmunity. This model is well supported by well-executed experiments and high-quality data. While the data strongly support the proposed model, I have some suggestions that could further improve the manuscript and strengthen the conclusions.

The data presented in figure 1 showing that absence of AtMC1 results in mild autoimmunity are convincing but do mirror, while also partially contradict, previously published data (Wang 2021, MPMI). I would suggest moving some of these data (A-E) to the supplementary figures while at the same time the authors should also measure autoimmunity of the AtMC1 mutant and AtMC1C220A using electrolyte leakage, MAPK activation assays and potentially qPCRs of early PTI and ETI marker genes, ROS production and Bik1 activation assays. These experiments will support already published data while will also clarify the autoimmunity phenotypes. Given that hNLRs only regulates certain aspects of ETI, these experiments will also clarify the confusing phenotypes of *atmc1/ nrg1.1/ nrg1.2*, and the inability of the helperless genetic background to fully rescue the *atmc1 AtMC1C220A236 -GFP* strong autoimmunity phenotype.

In contrast to Wang 2021, the authors show that PR1 levels are already high prior to infection. To clarify this point will be useful to evaluate the PR1 levels of the *atmc1* mutant before and after infection with Pto DC3000. Further confusing this point, in fig 4 the PR1a signal of the *atmc1* is very weak.

I found the argument put forward by the authors that the gene dosage of Wt AtMC1 alleles in suppressing the phenotype caused by catalytically inactive AtMC1 compelling but to support this statement the authors should provide some indication of the levels of Wt AtMC1 compare with the levels of the catalytically inactive AtMC1. This can be easily achieved by measuring the expression levels of the 2 genes by qPCRs using specific primers.

I found the argument put forward by the authors explaining why they use Wt AtMC1C220A -GFP plants vs atmc1 AtMC1C220A -GFP plant for the IP-MS experiments described in Fig 4 confusing. I am assuming that the authors choose the catalytically inactive AtMC1 construct in order to enhance their chances of finding interactors but that needs to be explained in more details. Nevertheless, the authors did find and afterwards validated (fig 6) meaningful interacting proteins.

I agree with the authors interpretations that the lack of Individual mutations in NLRs and PTI-related components able to rescue the autoimmune phenotype of catalytically inactive AtMC1 suggests that the autoimmune phenotype is mediated by multiple immune receptors. To verify this hypothesis, the authors need to evaluate the immunity phenotype of the of the AtMC1 adult mutant plants against multiple pathogens including DC3000 strains carrying effectors recognised by Arabidopsis (avrRpt2, avrRps4) and necrotrophic pathogens.

Further to this point, will be important to clarify if MC1 only protects adult plants from autoimmunity or can finetune the activation of immunity? The results from seedlings (Coll et al, 2010) and adult plants (Wang 2021) suggests that latter is the case. The authors can easily investigate this possibility by evaluating the immunity phenotypes (MAPK activation, electrolyte leakage etc) of the AtMC1mutant and catalytically inactive AtMC1 following pathogen infections.

Overall I really enjoyed the discussion but it needs to be expanded to further discuss the presented data with the data in Coll et al, 2010 and Wang 2021.

Referee #2:

- The manuscript describes a phenomenon of misregulated immuno condensates that are formed in a mutant plant. Although the experimental strategy is plausible, I am missing at least a conceptual explanation for the function of these condensates under normal (non-mutant) conditions. Under what native conditions would they form? Under what circumstances would their generation be adaptive for the plant? Please discuss.
- The dominant negative effect of AtMC1 C220A may also be explained by a negative effect on AtMC2 (e.g., competitive inhibition). Can the authors discuss or exclude this possibility?
- The IP data is presented as sole evidence for AtMC1 interaction with the respective NLRs/PRR. Can the authors please present additional interaction data for the selected immune receptors (e.g., Y2H or FRET-FLIM)? Can they please also discuss if the interaction may be direct?
- What mediates the condensation? Is it NLR-driven or MC1-driven, or both? Are there any surfaces or motifs on MC1 that contribute to a potentially direct interaction? (Also here, Y2H with truncated versions of NLRs and AtMC1 would be a useful method). Please at least discuss.
- What is the effect of MC1 activity on NLRs? Are the NLRs a catalytic target of MC1? Please discuss.

Line 49f: Please use a different reference here. Huang et al 2022, and Jia et al. 2022 are predominantly focused on protein structure and are not the original articles to prove genetical dependence of EDS1 and TNLs.

Line 52: Recently, Ibrahim et al. 2024 (BioRxiv, <https://doi.org/10.1101/2024.09.19.613839>) report localization of NRG1 helper NLR to target organellar membranes. I suggest to include the reference and to choose a more careful phrasing. I am not aware of any published evidence for 'perturbation of PM integrity' by NLRs, please revise.

In my opinion, the evidence in Jacob et al. 2021 for calcium channel activity could be stronger and rests on other, previous studies. I suggest to also include the reference Bi et al. 2022 (Cell) that first shows a more comprehensive study of NLR channel activity for Zar1. The authors may also want to stick to the more conservative phrasing of 'calcium-permeable cation-selective channel activity' that was used in the Bi et al. original publication.

Line 92: Grammar 'catalytic dependent manner'. catalysis dependent manner?

Line 96: Grammar 'plant have display reduced'

Line 235: Typo. Should be NRG1, not 'NGR1'

Line 305: Grammar. Should be 'interaction of AtMC1 variants with RPS2 and SS14 was confirmed...'

Line 312f: Convolutd phrasing and grammar. '1) AtMC1 or its catalytic activity or its targets are guarded by (an) NLR(s). 2)

AtMC1 participates in the proteostasis of immune components and overexpressed catalytically inactive AtMC1 binds and traps immune components (NLRs and components involved in PTI), thus preventing turnover.

Please revise the sentence avoiding brackets.

Line 328: Define 'camta3'. Give some background to camta3

Line 329: Should be 'present in the Arabidopsis Col-0 accession'.

Line 330f: Doubling. 'autoimmune plant atmc1 AtMC1C220A-GFP plants'

Line 555: '(139 DN-NLRs or out 166 NLRs present in Arabidopsis)'. Should be '139 out of 166'?

Figure 1, D: Could the authors please discuss in the results part why PR1 shows different expression between the CRISPR and

T-DNA line? This is a potential weakness of the study and at least an explanation is required.

Figure 6, A: Please adjust the alignment of the IP anti-HA blot the blot below and above.

Figure 10: I do not agree with the illustration of pentameric condensates. This is not supported by evidence. From the molecular size of the light microscopic puncta, a 'resistosome'-like structure is not supported. Rather have a look at Ma et al. 2024 (Nature), <https://doi.org/10.1038/s41586-024-07668-7>

Although, it is unclear how widespread this filament formation of SINRC2 is across other NLRs. However, these alternative NLR high molecular weight assemblies underline that pentameric assemblies are not the only possible assemblies. The authors ought to be more neutral in their illustration.

Can the authors please illustrate how EDS1 integrates into their presented working model?

Referee #3:

The manuscript by Salguero-Linares and colleagues, entitled "Lack of AtMC1 catalytic activity triggers autoimmunity dependent on NLR stability", presents impressive lines evidence supporting the role of atMC1 in regulating NLR stability by and large. This work is in line with understanding in the field of immunology, in particular, on the mode of action of Caspase8, of which function is distinct when assayed with loss-of-function mutants vs. loss-of-activity. In this sense, this work conveys an important idea of addressing loss-of-activity of an immune molecule, while leaving the molecule in the cell, to better characterization its function in the context of immunity. Note that diverse function of an immune molecule can be best addressed when different alleles are subjected under investigation in addition to the "absence" of the molecules, which is also known for well-studied NLR genes (MacQueen et al., <https://www.nature.com/articles/nplants2016110>). In this sense, this approach using catalytically inactive AtMC1 is a timely and important achievement with huge academic merits.

Their genetics establishment of creating a sensitized genetic background, by introducing the catalytically inactive AtMC1C220A to atmc1 mutant background shall be regarded as a fine genetic trick to reveal the function of AtMC1, as this transgene's effect on triggering autoimmunity is masked by the presence of WT AtMC1 in the genetic background. In other words, mild autoimmunity symptoms in the absence of AtMC1, seen in atmc1 mutants, became aggravated when catalytic inactive version of AtMC1 is introduced in large quantity, indicating that NLR activation-as usual culprits of triggering autoimmunity in general-is sensitized by the presence of "inactive version of a host immune molecule" in a dose-dependent manner. Involvement of NLRs were supported by series of genetics tests using signaling components of NLR downstream, which is quite up to date, as well as IP-MS experiments, identifying defence components including NLRs. Their proposition of AtMC1C220A possibly tethering immune components (such as NLRs) into an aberrant proteostasis status was nailed down with the cell biological probing of such structures as well as again with strong genetic complementation of the autoimmune phenotype with the introduction of SNIPER, a general NLR-targeting E3 ligase.

The genetic experiments employed in this work and thinking frames are extremely robust, which cannot be achieved in a short duration of research agenda. I could see the collective efforts from the dedicated manpower, coming from post-grad education and massive collaboration, achieving this level of science and scholarship under the continuous funding from the supporting grant agencies. Thus, this work shall be rather published sooner than later without a delay with too much of additional experiments. With massive genetic data, one can ask more to do to corroborate the proposed models in theory: however, I do not see any other route of improving this manuscript with additional genetic work. The final experiment of SNIPER suppressing the phenotype, to the geneticists' eyes, is the real proof that collective regulation of NLRs are the culprit. It is known that many NLRs degrade upon activation, and thus it is quite expected to see only well-known stable NLRs are fished out from IP-MS-MS data. The DN-experiment attempt was wonderful, while the constructs were designed to target specific ones. Further experiments targeting multiple NLRs with CRISPR-constructs might do the job to corroborate the findings, but I am not aware of any published resources. Thus, I think that the current manuscript is an adequate piece ready for publication with minor points of improvement.

Here are several minor comments that may help the authors improve parts of the manuscript.

1. It would guide readers much better if the CRISPR/Cas9-generated mutant allele information is shown as diagram in the main figure. In Figure S1, it is not clear which alleles matches with which line of the depicted chromatograms. I think there is a misalignment of the allele labeling in vertical lines with the actual chromatograms, which can be confusing. It is advisable to indicate which gRNAs were used to generate each allele, such that CR#1, supposedly a locus deletion allele, is made from double targeting gRNA1 and gRNA3. If gRNA2 is not used, no use of indicating in the figure. Or was gRNA2 included in the same vector but just came out non-functional gRNA? PAM sites should be indicated with clear indication of the gRNA sequences in the figure. Was CR#2 also generated from the same gRNA combination (gRNA1/gRNA3)? It would be also adequate to indicate the position of T-DNA insertion for the SALK line used in this study in the same figure.

2. L254: There appears to be a discrepancy between the CRISPR mutant line numbers. The text refers to atmc1-CR#3, while Figure 4 shows atmc1-CR#2. L255-256 shall be clarified with Figure S1 (not main figure 1A). It is hard to see which "single gRNA" targeting both 5' and 3' UTRs. Additionally, the authors shall fix the method information as well. L721 mentioned atmc1-CR #3, but it must be CR#2 according to S1. Please ensure consistency between the figure and the text to avoid confusion.

And, most importantly, what is the nature of this allele (#2 or #3)? Is it a locus deletion allele with non-residual expression of the gene? Otherwise, qRT shall be provided as this does not target the coding region.

3. The position of atMC2 catalytic mutant shall be clarified: L186 as AtMC2C258A vs. in Figure S5, C256A.

4. L183-184: This statement, suggesting that a threshold level of AtMC1C220A protein accumulation, is quite interesting indeed, as the native promoter driven vs. 35S driven plants give rise different phenotypes. It would be nice to have a justification on this notion, if the authors have western blot comparing AtMC1C220A protein expression driven by the native promoter versus the constitutive promoter or any other equivalent data (addressing ectopic location etc.). Accumulation of AtMC1C220A protein expression in OE lines were well appreciated in multiple figures, and this experiment might not be a challenging one to perform.

5. Below technical comments are focused on improving coIP experiments, which might authors could consider for revision.

Figure 6A: In the anti-HA blot detecting RPS2-HA in the input fractions, it is unclear why bands are present when RPS2-HA is not co-expressed. Additionally, a proper negative control should be included to demonstrate that RPS2-HA does not bind non-specifically to the anti-GFP beads. This would help confirm the specificity of the observed interactions and strengthen the validity of the results.

Figure 6B: Similarly, a proper negative control should be included to demonstrate that SSI4 does not bind non-specifically to the anti-GFP beads.

Figure 6C: It is difficult to justify the stronger interaction between SOBIR1 and AtMC1C220A in the atmc1 mutant background based on the current data, as there is already higher expression of SOBIR1 in the atmc1 mutant background, as shown by the western blot of the input fractions. Additionally, a proper negative control should be included by performing an anti-GFP pull-down in plants lacking the AtMC1C220A-GFP transgene to ensure that SOBIR1 does not bind non-specifically to the anti-GFP beads.

L356-358: I think that the statement, 'mutating the catalytic site of AtMC1 seems to strengthen the interaction of this protein with its potential clients/substrates,' is not adequately supported by the data in Figure 6. The figure does not clearly demonstrate that the mutation enhances these interactions, and further evidence or clarification is needed to substantiate this claim. The authors may provide alternative explanation by elaborating the concept of trapping, such that catalytically inactive AtMC1 cannot facilitate substrate loading/unloading processes.

Referee #1:

Overall, this manuscript offers novel insights into how metacaspases safeguard the plant immune system. The authors propose a model in which AtMC1 regulates NLR protein levels and prevents autoimmunity. This model is well supported by well-executed experiments and high-quality data. While the data strongly support the proposed model, I have some suggestions that could further improve the manuscript and strengthen the conclusions.

The data presented in figure 1 showing that absence of AtMC1 results in mild autoimmunity are convincing but do mirror, while also partially contradict, previously published data (Wang 2021, MPMI). I would suggest moving some of these data (A-E) to the supplementary figures while at the same time the authors should also measure autoimmunity of the AtMC1 mutant and AtMC1C220A using electrolyte leakage, MAPK activation assays and potentially qPCRs of early PTI and ETI marker genes, ROS production and Bik1 activation assays. These experiments will support already published data while will also clarify the autoimmunity phenotypes. Given that that hNLRs only regulates certain aspects of ETI, these experiments will also clarify the confusing phenotypes of *atmc1/ nrg1.1/ nrg1.2*, and the inability of the helperless genetic background to fully rescue the *atmc1 AtMC1C220A236 -GFP* strong autoimmunity phenotype.

As suggested by the reviewer, data from Figure 1-E have been moved to the supplement. Our data align with Wang *et al.*, 2021 (doi: 10.1094/MPMI-07-21-0197-R) in that adult *atmc1* mutant plants show restricted growth and enhanced resistant to *Pseudomonas syringae* pv. *tomato*. Wang *et al.* detect increased PR1 accumulation in *atmc1* mutant plants in comparison to Wt only after infection, while we can already detect this PR1 overaccumulation in *atmc1* mutants without any additional stimulus. We do not think this is a discrepancy, but simply due to the fact that in their time 0 PR1 signal in *atmc1* mutants is very low compared to infected samples, and thus, difficult to detect in such an immunoblot setting. In our case, since we only compare PR1 accumulation in unchallenged plants, we can detect more subtle differences between Wt and *atmc1* mutant plants.

To characterize the autoimmunity phenotypes in *atmc1* mutant and catalytically inactive plants we have performed the most commonly used experiments in the literature, that is: fresh weight measurement to determine the degree of growth restriction, trypan blue to determine the presence of dead cells and PR1 immunoblot to determine activation of defense responses. We acknowledge the reviewers' suggestion to perform further characterization of these lines by using alternative experiments such as qPCR of genes activated by PTI and ETI, ROS production and Bik1 activation. As shown in Figure 1 for Reviewers, ROS accumulates strongly in the severely autoimmune plant *atmc1 AtMC1^{C220A}*, while in *atmc1* mutant plants we do not detect such a strong accumulation, which would be in line with the idea of this mutant being only slightly autoimmune (Panel A of Figure 1 for Reviewers). We have also tested many different PTI and ETI marker genes by RT-qPCR. Panel B of Figure 1 for Reviewers shows the results for those genes that gave the most robust

expression: RLP23, ICS1, EDS1 and PR1a. A strong induction of these markers can be observed for *atmc1 AtMC1^{C220A}* plants and in *atmc1* induction can be detected for *RLP23*, *EDS1* and *ICS1*, although to lower levels than in the *atmc1 AtMC1^{C220A}* line, as expected. We did not obtain consistent results when testing Bik activation. We would prefer to stick to the original data in the final version of the manuscript, which we think provides already solid and widely accepted proof of autoimmunity. However, we leave it to the editor/reviewers discretion to decide whether we should include the experiments shown in Figure 1 for Reviewers.

In contrast to Wang 2021, the authors show that PR1 levels are already high prior to infection. To clarify this point will be useful to evaluate the PR1 levels of the *atmc1* mutant before and after infection with Pto DC3000. Further confusing this point, in fig 4 the PR1a signal of the *atmc1* is very weak.

Wang *et al.*, 2021 already showed that infection results in higher accumulation of PR1 in the *atmc1* mutant than in Wt plants. The authors of that work did not observe differences in PR1 accumulation between Wt and *atmc1* mutant before infection (time 0) but, as discussed in the previous comment, additional exposure would probably have revealed PR1 accumulation in *atmc1* mutants, molecularly underlining the restricted growth of these plants.

I found the argument put forward by the authors that the gene dosage of Wt AtMC1 alleles in suppressing the phenotype caused by catalytically inactive AtMC1 compelling but to support this statement the authors should provide some indication of the levels of Wt AtMC1 compare with the levels of the catalytically inactive AtMC1. This can be easily achieved by measuring the expression levels of the 2 genes by qPCRs using specific primers.

As noted in the manuscript the exact same *AtMC1^{C220A}* insertion (transgenic event) was used to generate Wt *AtMC1^{C220A}* and *atmc1 AtMC1^{C220A}* plants, as they were obtained by crossing. Therefore, Wt *AtMC1^{C220A}* plants have 4 copies of AtMC1 (2 wild-type and 2 catalytically inactive), while *atmc1 AtMC1^{C220A}* plants have only 2 copies of AtMC1 (both catalytically inactive). In our view, this is sufficient to postulate gene dosage effects. The catalytically active AtMC1 copies in Wt *AtMC1^{C220A}* plants can cleave *AtMC1^{C220A}*, which might be sufficient to render the plants non-autoimmune.

However, to visualize gene dosage, we have performed RT-qPCR to quantify *AtMC1* gene expression in plants expressing *35S::AtMC1^{C220A}-GFP* in a Wt or *atmc1* mutant background. As shown in Figure 2 for Reviewers, Wt *35S::AtMC1^{C220A}-GFP* plants have double dosage of AtMC1 than *35S::AtMC1^{C220A}-GFP* plants. This double gene dosage corresponds to the wild type + mutated AtMC1 versions existing in these plants. We could not compare expression levels of *AtMC1* vs *AtMC1^{C220A}* directly as suggested by the reviewer because C220A is a single-nucleotide mutation, making it difficult to differentiate the two genes by PCR.

I found the argument put forward by the authors explaining why they use Wt AtMC1C220A -GFP plants vs *atmc1* AtMC1C220A -GFP plant for the IP-MS experiments described in Fig 4 confusing. I am assuming that the authors choose the catalytically inactive AtMC1 construct in order to enhance their chances of finding interactors but that needs to be explained in more details. Nevertheless, the authors did find and afterwards validated (fig 6) meaningful interacting proteins.

We chose AtMC1^{C220A} in both backgrounds to be able to run a better controlled experiment. Also, we used the same transgenic event (AtMC1^{C220A} #1,6) in different backgrounds (autoimmunity and no autoimmunity) to rule out the possibility of different protein levels affecting the nature of interactors captured in both genotypes. This explanation has been included in the text for clarity as suggested.

I agree with the authors interpretations that the lack of Individual mutations in NLRs and PTI-related components able to rescue the autoimmune phenotype of catalytically inactive AtMC1 suggests that the autoimmune phenotype is mediated by multiple immune receptors. To verify this hypothesis, the authors need to evaluate the immunity phenotype of the of the AtMC1 adult mutant plants against multiple pathogens including DC3000 strains carrying effectors recognised by Arabidopsis (*avrRpt2*, *avrRps4*) and necrotrophic pathogens.

It has been already shown that *atmc1* mutant plants are more susceptible to necrotrophic pathogens (see Figure S6 from the article Coll *et al.*, 2014 doi: 10.1038/cdd.2014.50) and more resistant to *Pto* DC3000(*avrRpt2*) (see Wang *et al.*, 2021, Fig. 1B). We have also tested resistance of *atmc1* mutant plants to *Pto* DC3000(*avrRpt2*) as suggested and consistently, they are also more resistant to this strain (Figure EV1G). These data indicate that AtMC1-mediated autoimmunity is mediated by multiple immune receptors. These new data and interpretation have now been included in the text.

Further to this point, will be important to clarify if MC1 only protects adult plants from autoimmunity or can finetune the activation of immunity? The results from seedlings (Coll *et al.*, 2010) and adult plants (Wang 2021) suggests that latter is the case. The authors can easily investigate this possibility by evaluating the immunity phenotypes (MAPK activation, electrolyte leakage etc) of the AtMC1 mutant and catalytically inactive AtMC1 following pathogen infections.

We fully agree with the reviewer's interpretation that AtMC1 may contribute to fine-tuning immunity. The data presented in this paper and in previous papers (Coll *et al.*, 2010 doi: 10.1126/science.1194980; Coll *et al.*, 2014 doi: 10.1038/cdd.2014.50; Wang *et al.*, 2021 doi: 10.1094/MPMI-07-21-0197-R) support this hypothesis. As we have shown recently and it is included as part of the discussion of this manuscript, AtMC1 is a protein that can get recruited into condensates and that has the capacity to dissolve aggregated proteins, with all evidence pointing towards a role in protein quality control during stress responses (Ruiz-Solaní *et al.*, 2023 doi: 10.1093/plcell/koad172; Coll *et al.*,

2014). In young plants and under basal conditions, *atmc1* mutant plants do not show any obvious growth phenotype and it is only with age that these plants start displaying autoimmune phenotypes: spontaneous cell death, PR1 accumulation and growth restriction. However, young *atmc1* mutants show reduced cell death levels upon infection (Coll *et al.*, 2010 and Coll *et al.*, 2014), pointing towards a potential role in cell death execution. This pro-death / pro-life role of metacaspases has already been reported in yeast (Eisele-Bürger *et al.*, 2023 doi: 10.1016/j.celrep.2023.113372) although the mechanisms regulating this dual behavior is still far from being elucidated and is something we are actively working on. We measure cell death using trypan blue staining rather than electrolyte leakage, because it gives a better idea of the zonation of this phenomenon and it is more sensitive than ion leakage, revealing cell death associated *atmc1* phenotypes, that we could not observe otherwise. We tested MAPK activation but unfortunately, we did not obtain very clear results, perhaps as a consequence that the autoimmunity of the *atmc1* mutant is mild, compared to previously reported autoimmune mutants. In any case, we have modified the text to clarify that MC1 is involved in fine-tuning of immunity.

Overall I really enjoyed the discussion but it needs to be expanded to further discuss the presented data with the data in Coll et al, 2010 and Wang 2021. Following the reviewer's advice, we have deepened our discussion of the data in relation to the aforementioned papers.

Referee #2:

- The manuscript describes a phenomenon of misregulated immune condensates that are formed in a mutant plant. Although the experimental strategy is plausible, I am missing at least a conceptual explanation for the function of these condensates under normal (non-mutant) conditions. Under what native conditions would they form? Under what circumstances would their generation be adaptive for the plant? Please discuss.

We have shown that similar to other condensate-prone proteins, under normal conditions, AtMC1 is soluble in the cytoplasm. AtMC1 condensation occurs as a result of certain stresses and it may be derived from intracellularly surpassing a certain threshold of proteotoxicity that triggers this change in physico-chemical properties of the protein, as part of the stress induced activation of protein quality control mechanisms. Formation of these condensates would be adaptive for the plant by contributing to alleviate stress-induced proteotoxicity, thereby enhancing the plant's fitness and responses to the environment. For example, we have shown that heat and aging trigger AtMC1 condensation and increasing AtMC1 levels reduces the formation of protein aggregates, delays senescence and improves thermotolerance (Ruiz-Solaní *et al.*, 2023 doi: 10.1093/plcell/koad172). In the context of immunity, AtMC1 would contribute to the turnover of NLRs and other immune regulators via condensate formation, contributing to their subsequent clearance through major degradative pathways such as autophagy and proteasomal degradation. Lack of AtMC1 or the stabilization of immune components in the *atmc1 AtMC1-C220A* line would

result in autoimmunity derived from defects in turning off or over immune components in these plants.

- The dominant negative effect of AtMC1 C220A may also be explained by a negative effect on AtMC2 (e.g., competitive inhibition). Can the authors discuss or exclude this possibility?

To test this hypothesis, we generated *AtMC1^{C220A}-GFP* in an *atmc1 atmc2* double mutant background. As shown in new Figure EV3C, these plants show an autoimmune phenotype indistinguishable from *atmc1 AtMC1^{C220A}-GFP*, thus ruling out the possibility of competitive inhibition.

- The IP data is presented as sole evidence for AtMC1 interaction with the respective NLRs/PRR. Can the authors please present additional interaction data for the selected immune receptors (e.g., Y2H or FRET-FLIM)? Can they please also discuss if the interaction may be direct?

The reviewer is right that our evidence for interaction is based on co-immunoprecipitation data. However, we have not simply confirmed the mass-spectrometry results transiently in *Nicotiana benthamiana* (Figure EV5), but in the case of the NLRs RPS2 and SSI4 we have also generated stable double transgenics in an *atmc1 AtMC1-C220* or *AtMC1* background to test the association, which was challenging due to the strong autoimmunity of AtMC1-CA (Fig. 6). Y2H does not seem the most optimal method to test interaction with NLRs, since several of them are membrane localized and in many cases, they seem to be toxic when expressed in yeast (Farid el Kasmi, personal communication). We do not have evidence of the interaction being direct, but it seems plausible that AtMC1 is part of larger complexes that may participate in cytoplasm clearance. To be more precise, when referring to co-immunoprecipitation experiments, we have used “co-immunoprecipitation” instead of “interaction” and we also consider in the text the possibility of an indirect interaction based on the presented evidence.

- What mediates the condensation? Is it NLR-driven or MC1-driven, or both? Are there any surfaces or motifs on MC1 that contribute to a potentially direct interaction? (Also here, Y2H with truncated versions of NLRs and AtMC1 would be a useful method). Please at least discuss.

We have previously shown that AtMC1 condensation is partly mediated by one of its intrinsically disordered regions (IDR) located towards the C-terminus of the protein and known as the 360 loop (Ruiz-Solaní *et al.*, 2024 doi: 10.1093/plcell/koad172; Van Midden *et al.*, 2021 doi: 10.1002/1873-3468.14165). The N-terminal prodomain of AtMC1 also alters its condensate dynamics (Ruiz-Solaní *et al.*, 2024 doi: 10.1093/plcell/koad172) and genetically, we know that the AtMC1 prodomain is required for the autoimmune phenotype. This could potentially suggest that both the 360 loop and the N-terminus can participate in the (direct or indirect) interaction of AtMC1 with NLRs.

- What is the effect of MC1 activity on NLRs? Are the NLRs a catalytic target of MC1? Please discuss.

So far, we have not detected cleavage of the tested NLRs by AtMC1 when co-expressed (Figure 6 and Figure EV5). However, biochemical evidence strongly suggests that AtMC1 may not only act as a canonical metacaspase, cleaving substrates after arginine or lysine, but also as a disaggregase, clearing toxic amorphous protein aggregates (Ruiz-Solani *et al.*, 2023). In this regard, AtMC1 is quite unspecific in its disaggregase activity, being able to dissolve a broad range of aggregated proteins, including pathological forms of human proteins such as aggregated huntingtin, tau or transthyretin, among others. It is thus plausible that AtMC1 clears misfolded/aggregated NLRs as part of the protein quality control machinery after an immune reaction to prevent proteotoxicity and to help returning the cell to a normal state. Given the stress-dependent condensation of AtMC1, it is tempting to speculate that this clearing activity or even the canonical protease activity of AtMC1 occurs within condensates/stress granules. We are currently testing these hypotheses, although the spatio-temporal dimension of the phenomenon and the difficulty of working with condensation-prone proteins makes it technically very challenging. This is particularly the case in the context of immunity, considering the fact that NLRs are also challenging proteins to do microscopy with, in their full-length forms.

Line 49f: Please use a different reference here. Huang *et al.* 2022, and Jia *et al.* 2022 are predominantly focused on protein structure and are not the original articles to prove genetical dependence of EDS1 and TNLs.

The references have been changed as suggested.

Line 52: Recently, Ibrahim *et al.* 2024 (BioRxiv, <https://doi.org/10.1101/2024.09.19.613839>) report localization of NRG1 helper NLR to target organellar membranes. I suggest to include the reference and to choose a more careful phrasing. I am not aware of any published evidence for 'perturbation of PM integrity' by NLRs, please revise.

As suggested, the phrasing has been changed to better reflect available data and the new reference has been added.

In my opinion, the evidence in Jacob *et al.* 2021 for calcium channel activity could be stronger and rests on other, previous studies. I suggest to also include the reference Bi *et al.* 2022 (Cell) that first shows a more comprehensive study of NLR channel activity for Zar1. The authors may also want to stick to the more conservative phrasing of 'calcium-permeable cation-selective channel activity' that was used in the Bi *et al.* original publication.

We have added the suggested reference and modified the text.

Line 92: Grammar 'catalytic dependent manner'. catalysis dependent manner?
Line 96: Grammar 'plant have display reduced'
Line 235: Typo. Should be NRG1, not 'NGR1'
Line 305: Grammar. Should be 'interaction of AtMC1 variants with RPS2 and SS14 was confirmed...'
Line 312f: Convolved phrasing and grammar. '1) AtMC1 or its catalytic activity or its targets are guarded by (an) NLR(s). 2) AtMC1 participates in the proteostasis of immune components and overexpressed catalytically inactive AtMC1 binds and traps immune components (NLRs and components involved in PTI), thus preventing turnover. Please revise the sentence avoiding brackets.
Line 328: Define 'camta3'. Give some background to camta3
Line 329: Should be 'present in the Arabidopsis Col-0 accession'.
Line 330f: Doubling. 'autoimmune plant atmc1 AtMC1C220A-GFP plants'
Line 555: '(139 DN-NLRs or out 166 NLRs present in Arabidopsis)'. Should be '139 out of 166'?

All typos and convolved phrasing have been corrected.

Figure 1, D: Could the authors please discuss in the results part why PR1 shows different expression between the CRISPR and T-DNA line? This is a potential weakness of the study and at least an explanation is required.

The difference in PR1a levels could be explained by the different nature of the two mutants. The CRISPR mutant is a clean mutant lacking the entire gene (Appendix Figure S1), whereas the T-DNA mutant contains the insertion after the first exon, corresponding to approximately 74 amino acids. Regardless of this difference, the results of *Pst* DC3000 growth and plant fresh weight measurements suggest that both genotypes result in autoimmunity. However, we have now included a description of the difference between the two mutants and their potential effect in terms of PR1a accumulation in the results section.

Figure 6, A: Please adjust the alignment of the IP anti-HA blot the blot below and above.

The alignment has been adjusted.

Figure 10: I do not agree with the illustration of pentameric condensates. This is not supported by evidence. From the molecular size of the light microscopic puncta, a 'resistosome'-like structure is not supported. Rather have a look at Ma *et al.* 2024 (Nature), <https://doi.org/10.1038/s41586-024-07668-7>

Although, it is unclear how widespread this filament formation of SINRC2 is across other NLRs. However, these alternative NLR high molecular weight assemblies underline that pentameric assemblies are not the only possible assemblies. The authors ought to be more neutral in their illustration.

The figure was a bit misleading, as it did not intend to depict structured resistosomes, but rather unstructured accumulations of misfolded NLRs and

other immune regulators. We have included a new figure that hopefully reflects better this idea.

Can the authors please illustrate how EDS1 integrates into their presented working model?

Our interpretation is that the observed autoimmunity in *atmc1 AtMC1^{C220A}-GFP* plants may be due to hyperactivation of a SA-dependent NLRs that require the feedback loop through EDS1-PAD4-ADR1 node, which normally amplifies immune responses, together with SA-independent pathways. This interpretation is included in the discussion. We prefer not to include EDS1 in the scheme provided in Figure 10 to keep it simple, otherwise we feel we should include other components of the pathway that have been tested and it would become overly complicated.

Referee #3:

The manuscript by Salguero-Linares and colleagues, entitled "Lack of AtMC1 catalytic activity triggers autoimmunity dependent on NLR stability", presents impressive lines evidence supporting the role of atMC1 in regulating NLR stability by and large. This work is in line with understanding in the field of immunology, in particular, on the mode of action of Caspase8, of which function is distinct when assayed with loss-of-function mutants vs. loss-of-activity. In this sense, this work conveys an important idea of addressing loss-of-a ctivity of an immune molecule, while leaving the molecule in the cell, to better characterization its function in the context of immunity. Note that diverse function of an immune molecule can be best addressed when different alleles are subjected under investigation in addition to the "absence" of the molecules, which is also known for well-studied NLR genes (MacQueen *et al.*, <https://www.nature.com/articles/nplants2016110>). In this sense, this approach using catalytically inactive AtMC1 is a timely and important achievement with huge academic merits.

Their genetics establishment of creating a sensitized genetic background, by introducing the catalytically inactive AtMC1C220A to *atmc1* mutant background shall be regarded as a fine genetic trick to reveal the function of AtMC1, as this transgene's effect on triggering autoimmunity is masked by the presence of WT AtMC1 in the genetic background. In other words, mild autoimmunity symptoms in the absence of AtMC1, seen in *atmc1* mutants, became aggravated when catalytic inactive version of AtMC1 is introduced in large quantity, indicating that NLR activation-as usual culprits of triggering autoimmunity in general-is sensitized by the presence of "inactive version of a host immune molecule" in a dose-dependent manner. Involvement of NLRs were supported by series of genetics tests using signaling components of NLR downstream, which is quite up to date, as well as IP-MS experiments, identifying defence components including NLRs. Their proposition of AtMC1C220A possibly tethering immune components (such as NLRs) into an aberrant proteostasis status was nailed

down with the cell biological probing of such structures as well as again with strong genetic complementation of the autoimmune phenotype with the introduction of SNIPER, a general NLR-targeting E3 ligase.

The genetic experiments employed in this work and thinking frames are extremely robust, which cannot be achieved in a short duration of research agenda. I could see the collective efforts from the dedicated manpower, coming from post-grad education and massive collaboration, achieving this level of science and scholarship under the continuous funding from the supporting grant agencies. Thus, this work shall be rather published sooner than later without a delay with too much of additional experiments. With massive genetic data, one can ask more to do to corroborate the proposed models in theory: however, I do not see any other route of improving this manuscript with additional genetic work. The final experiment of SNIPER suppressing the phenotype, to the geneticists' eyes, is the real proof that collective regulation of NLRs are the culprit. It is known that many NLRs degrade upon activation, and thus it is quite expected to see only well-known stable NLRs are fished out from IP-MS-MS data. The DN-experiment attempt was wonderful, while the constructs were designed to target specific ones. Further experiments targeting multiple NLRs with CRISPR-constructs might do the job to corroborate the findings, but I am not aware of any published resources. Thus, I think that the current manuscript is an adequate piece ready for publication with minor points of improvement.

Here are several minor comments that may help the authors improve parts of the manuscript.

1. It would guide readers much better if the CRISPR/Cas9-generated mutant allele information is shown as diagram in the main figure. In Figure S1, it is not clear which alleles matches with which line of the depicted chromatograms. I think there is a misalignment of the allele labeling in vertical lines with the actual chromatograms, which can be confusing. It is advisable to indicate which gRNAs were used to generate each allele, such that CR#1, supposedly a locus deletion allele, is made from double targeting gRNA1 and gRNA3. If gRNA2 is not used, no use of indicating in the figure. Or was gRNA2 included in the same vector but just came out non-functional gRNA? PAM sites should be indicated with clear indication of the gRNA sequences in the figure. Was CR#2 also generated from the same gRNA combination (gRNA1/gRNA3)? It would be also adequate to indicate the position of T-DNA insertion for the SALK line used in this study in the same figure.

We have corrected the figure as suggested for better clarity regarding all the mutant alleles. Since Reviewer #1 asked to move some of the *atmc1* mutant data into the Supplementary Data, we have therefore considered more appropriate to also keep this diagram as part of Figure EV1.

2. L254: There appears to be a discrepancy between the CRISPR mutant line numbers. The text refers to *atmc1*-CR#3, while Figure 4 shows *atmc1*-CR#2. L255-256 shall be clarified with Figure S1 (not main figure 1A). It is hard to see which "single gRNA" targeting both 5' and 3' UTRs. Additionally, the authors

shall fix the method information as well. L721 mentioned *atmc1*-CR #3, but it must be CR#2 according to S1. Please ensure consistency between the figure and the text to avoid confusion.

The *atmc1* CRISPR mutant line numbers have been corrected throughout the text (CR #3 is now referred to CR #2).

And, most importantly, what is the nature of this allele (#2 or #3)? Is it a locus deletion allele with non-residual expression of the gene? Otherwise, qRT shall be provided as this does not target the coding region.

atmc1 CR #2 is a locus deletion allele without residual expression of the gene.

3. The position of *atMC2* catalytic mutant shall be clarified: L186 as *AtMC2C258A* vs. in Figure S5, *C256A*.

This mistake has been fixed, the correct position is *C256A*.

4. L183-184: This statement, suggesting that a threshold level of *AtMC1C220A* protein accumulation, is quite interesting indeed, as the native promoter driven vs. 35S driven plants give rise different phenotypes. It would be nice to have a justification on this notion, if the authors have western blot comparing *AtMC1C220A* protein expression driven by the native promoter versus the constitutive promoter or any other equivalent data (addressing ectopic location etc.). Accumulation of *AtMC1C220A* protein expression in OE lines were well appreciated in multiple figures, and this experiment might not be a challenging one to perform.

The stable transgenic lines expressing *AtMC1^{C220A}-GFP* under the control of their native promoter did not recapitulate the phenotype of *35::AtMC1^{C220A}-GFP* plants regardless of their level of expression (new Appendix Figure S2B and C), which indicates that other factors including localization, may also contribute to autoimmunity.

5. Below technical comments are focused on improving colP experiments, which might authors could consider for revision.

Figure 6A: In the anti-HA blot detecting RPS2-HA in the input fractions, it is unclear why bands are present when RPS2-HA is not co-expressed. Additionally, a proper negative control should be included to demonstrate that RPS2-HA does not bind non-specifically to the anti-GFP beads. This would help confirm the specificity of the observed interactions and strengthen the validity of the results.

We have now added a new immunoblot including the suggested GFP control that demonstrate that RPS2-HA does not bind non-specifically to the anti-GFP

beads. This was also demonstrated in co-immunoprecipitations performed in *Nicotiana benthamiana* shown in Figure EV5.

Figure 6B: Similarly, a proper negative control should be included to demonstrate that SSI4 does not bind non-specifically to the anti-GFP beads.

Following the reviewers' suggestion we also repeated the co-immunoprecipitation experiment of Figure 6B to include the GFP control. As shown in Figure 3 for Reviewers SSI4 does not bind non-specifically to the anti-GFP beads, which was also demonstrated in Figure EV5. However, the lines that we have used in these new experiments unfortunately express much lower levels of SSI4 than the ones used in previous experiments. As a consequence, although we still detect the stronger SSI4- AtMC1^{C220A} co-immunoprecipitation, we cannot detect it for SSI4- AtMC1, which is always weaker, and may be more transient because the protease is active. Please, note that in this particular experiment we have been using T1 lines because in further generations *SSI4* is silenced. We would therefore prefer to keep Figure 6B as it is, unless the reviewers think otherwise.

Figure 6C: It is difficult to justify the stronger interaction between SOBIR1 and AtMC1C220A in the *atmc1* mutant background based on the current data, as there is already higher expression of SOBIR1 in the *atmc1* mutant background, as shown by the western blot of the input fractions. Additionally, a proper negative control should be included by performing an anti-GFP pull-down in plants lacking the AtMC1C220A-GFP transgene to ensure that SOBIR1 does not bind non-specifically to the anti-GFP beads.

The reviewer is right, and thus we do not claim that there is stronger interaction between SOBIR1 and AtMC1-C220A in the *atmc1* mutant background. Unfortunately during this review period, we have been experiencing technical problems with the SOBIR1 antisera and our immunoblots using this antibody are not working properly, therefore we cannot provide replacement experiments for Figure 6C including the GFP negative control. We are really sorry for this inconvenience and leave it in the hands of the editor and reviewers to decide whether to accept this experiment as provided in the previous version. Otherwise, we will spend more time trying this experiment, but for this we will have to buy new anti-SOBIR1 antisera or to ask our collaborators to perform this immunoblot with their antisera and our plant material.

L356-358: I think that the statement, 'mutating the catalytic site of AtMC1 seems to strengthen the interaction of this protein with its potential clients/substrates,' is not adequately supported by the data in Figure 6. The figure does not clearly demonstrate that the mutation enhances these interactions, and further evidence or clarification is needed to substantiate this claim. The authors may provide alternative explanation by elaborating the concept of trapping, such that catalytically inactive AtMC1 cannot facilitate substrate loading/unloading processes.

The reviewer is right and we have reformulated our data interpretation and adhered to the reasoning raised as follows:

“AtMC1 has a crucial role in proteostasis during proteotoxic stress, being rapidly recruited into condensates and contributing to their timely clearance (Ruiz-Solaní *et al*, 2023). This function seems to rely on the condensation-prone physico-chemical properties of AtMC1 and on its disaggregase activity, for which an intact catalytic site is essential (Ruiz-Solaní *et al*, 2023). Considering this, one could speculate that AtMC1 might contribute to the proteostasis of NLRs and other immune regulators in situations where the levels of these proteins increase, such as during an acute immune response. Supporting this hypothesis, we observed that the levels of BIK1, SOBIR1 and RPS2 increase drastically in *atmc1 AtMC1^{C220A}-GFP* plants (Figure 7A-B, Appendix Figure S8A). Further, adding an extra copy of the AtMC1 interacting TNL SS14 to *atmc1 AtMC1^{C220A}-RFP* plants (*proSS14::SS14-mCitrine*) (Appendix Figure S8B) resulted in individuals displaying extremely strong autoimmunity (Figure 7C). However, the fact that mutating the catalytic site of AtMC1 does not particularly seem to strengthen the interaction of this protein with its potential clients/substrates (Figure 6 and EV5) could indicate that indeed AtMC1^{C220A} acts as a trap for these proteins, since this variant of the protein substrate loading/unloading processes are hindered. Together, these data indirectly support a role of AtMC1 in preventing immune receptors hyperaccumulation dependent on its catalytic activity.”

Dear Dr. Coll,

Thank you for the submission of your revised manuscript to our editorial offices. I have now received the report from the three referees that were asked to re-evaluate the study, you will find below. As you will see, the referees now fully support the publication of the study in EMBO reports.

Before I can proceed with formal acceptance, I have these editorial requests I ask you to address in a final revised manuscript:

- Please order the manuscript sections like this, using these names:

Title page - Abstract - Keywords - Introduction - Results - Discussion - Methods - Data availability section - Acknowledgements (including the funding information) - Disclosure and Competing Interests Statement - References - Figure legends - Expanded View Figure legends

- Please make sure that all the funding information is also entered into the online submission system and that it is complete and similar to the one in the acknowledgement section of the manuscript text file. Presently, grants "European Union NextGenerationEU/PRTR", AGL2016-78002-R, postdoctoral fellowship FJC2021-046667-I, European Union NextGenerationEU/PRTR, CERCA Programme/Generalitat de Catalunya and MCIN/AEI/10.13039/501100011033 are missing from the submission system. Please check.

- Please remove now the referee access information from the Data Availability section and make sure the datasets are public latest upon online publication of the paper. Moreover, please provide direct links for both datasets.

- Please provide the Appendix file with page numbers and a table of content on the first page (with page numbers). Please follow the nomenclature Appendix Figure Sx, Appendix Table Sx etc. throughout the text, and also label the figures and tables according to this nomenclature. Please move the legends below the respective figures.

- Please remove the section 'Supplementary Data' from the manuscript text file.

- Please make sure that all figure panels are called out separately and sequentially. Presently, there seems to be no individual callouts for the panels of Figure 4. Please check. Moreover, please use the callout 'Appendix Figure Sx' or 'Appendix Table Sx' for callouts of Appendix items.

- Please check again that the number "n" for how many independent experiments were performed, their nature (biological versus technical replicates), the bars and error bars (e.g. SEM, SD) and the test used to calculate p-values is indicated in the respective figure legends. Please also check that all the p-values are explained in the legend, and that these fit to those shown in the figure. Please provide statistical testing where applicable. Please avoid the phrase 'independent experiment', but clearly state if these were biological or technical replicates. Please also indicate (e.g. with n.s.) if testing was performed, but the differences are not significant. In case n=2, please show the data as separate datapoints without error bars and statistics. See also:

<http://www.embopress.org/page/journal/14693178/authorguide#statisticalanalysis>

If n<5, please show single datapoints for diagrams. Moreover:

- Please note that the exact p values are not provided in the legends of figures 2e; 3d; 4c; EV 1d, f; EV 2c; EV 4e.

- Please indicate the statistical test used for data analysis in the legends of figures 1a; 7b; 9e.

- Please note that the box plots need to be defined in terms of minima, maxima, centre, bounds of box and whiskers, and percentile in the legends of figures 2e; 3d; 4c; 7b; 9e; EV 1d, f-g; EV 2c; EV 4e.

- Please note that information related to n is missing in the legends of figures 1a; 5b; 7b; 9e; EV 1f.

- Please note that the red and black arrowheads are not defined in the legend of figure 5a. This needs to be rectified.

- Please remove the Reagents & Tools table from the manuscript text file. This should be provided only as separate file.

- There seems to be a re-use of panels or parts of panels between Figure 3B (last panel) and Figure EV4F (last panel), Figure 3B (4th panel) and Appendix Figure S5B (second panel) and Figure EV4A (second panel) and EV4B (second panel). If this is intended, please clearly explain and indicate this in the respective figure legends.

In addition, I would need from you uploaded separately:

- a short, two-sentence summary of the manuscript (not more than 35 words).

- two to four short (!) bullet points highlighting the key findings of your study (two lines each).

- a schematic summary figure as separate file that provides a sketch of the major findings (not a data image) in jpeg or tiff format (with the exact width of 550 pixels and a height of not more than 400 pixels) that can be used as a visual synopsis on our website.

Best,

Referee #1:

In this revised version of the manuscript, the authors have addressed all my concerns and questions. I also support their suggestion to retain the original data in the final version.

Referee #2:

- The manuscript describes a phenomenon of misregulated immuno condensates that are formed in a mutant plant. Although the experimental strategy is plausible, I am missing at least a conceptual explanation for the function of these condensates under normal (non-mutant) conditions. Under what native conditions would they form? Under what circumstances would their generation be adaptive for the plant? Please discuss.

We have shown that similar to other condensate-prone proteins, under normal conditions, AtMC1 is soluble in the cytoplasm. AtMC1 condensation occurs as a result of certain stresses and it may be derived from intracellularly surpassing a certain threshold of proteotoxicity that triggers this change in physico-chemical properties of the protein, as part of the stress induced activation of protein quality control mechanisms. Formation of these condensates would be adaptive for the plant by contributing to alleviate stress-induced proteotoxicity, thereby enhancing the plant's fitness and responses to the environment. For example, we have shown that heat and aging trigger AtMC1 condensation and increasing AtMC1 levels reduces the formation of protein aggregates, delays senescence and improves thermotolerance (Ruiz-Solaní et al., 2023 doi: 10.1093/plcell/koad172). In the context of immunity, AtMC1 would contribute the turnover of NLRs and other immune regulators via condensate formation, contributing to their subsequent clearance through major degradative pathways such as autophagy and proteasomal degradation. Lack of AtMC1 or the stabilization of immune components in the *atmc1 AtMC1-C220A* line would result in autoimmunity derived from defects in turning off or over immune components in these plants.

I thank the authors for directing me to their Plant Cell paper which nicely addresses additional aspects of the *in vitro* behavior of MC1. I still find the current manuscript somewhat open-ended. But this may be intentional and should not preclude from publication.

- The dominant negative effect of AtMC1 C220A may also be explained by a negative effect on AtMC2 (e.g., competitive inhibition). Can the authors discuss or exclude this possibility?

To test this hypothesis, we generated AtMC1C220A-GFP in an *atmc1 atmc2* double mutant background. As shown in new Figure EV3C, these plants show an autoimmune phenotype indistinguishable from *atmc1 AtMC1C220A-GFP*, thus ruling out the possibility of competitive inhibition.

I appreciate the additional data presented. The data is conclusive and the point sufficiently considered.

- The IP data is presented as sole evidence for AtMC1 interaction with the respective NLRs/PRR. Can the authors please present additional interaction data for the selected immune receptors (e.g., Y2H or FRET-FLIM)? Can they please also discuss if the interaction may be direct?

The reviewer is right that our evidence for interaction is based on co-immunoprecipitation data. However, we have not simply confirmed the mass-spectrometry results transiently in *Nicotiana benthamiana* (Figure EV5), but in the case of the NLRs RPS2 and SSI4 we have also generated stable double transgenics in an *atmc1 AtMC1-C220* or *AtMC1* background to test the association, which was challenging due to the strong autoimmunity of *AtMC1-CA* (Fig. 6). Y2H does not seem the most optimal method to test interaction with NLRs, since several of them are membrane localized and in many cases, they seem to be toxic when expressed in yeast (Farid el Kasmi, personal communication). We do not have evidence of the interaction being direct, but it seems plausible that AtMC1 is part of larger complexes that may participate in cytoplasm clearance. To be more precise, when

referring to co-immunoprecipitation experiments, we have used "co-immunoprecipitation" instead of "interaction" and we also consider in the text the possibility of an indirect interaction based on the presented evidence.

I agree with the authors that establishing Y2H for interaction of membrane-associated proteins can be challenging. Although soluble parts/truncations of the proteins may potentially work. The reason why I would prefer additional evidence for interaction is that MC1 appears to interact with such vastly different proteins with regards to a putative binding motif, surface signature, protein domains etc. What do PRRs and NLRs have in common to recruit direct binding of MC1?

Anyways, the presented IP data is convincing enough to argue for indirect interaction, which is now reflected in the alternative phrasing. I request no further data.

Yet, the model in figure 10 still implies a direct interaction of MC1 and (cytoplasmic?) immune receptors. Is there potentially a smart way to account for indirect/direct interaction without further confounding the model figure?

- What mediates the condensation? Is it NLR-driven or MC1-driven, or both? Are there any surfaces or motifs on MC1 that contribute to a potentially direct interaction? (Also here, Y2H with truncated versions of NLRs and AtMC1 would be a useful method). Please at least discuss.

We have previously shown that AtMC1 condensation is partly mediated by one of its intrinsically disordered regions (IDR) located towards the C-terminus of the protein and known as the 360 loop (Ruiz-Solaní et al., 2024 doi: 10.1093/plcell/koad172; Van Midden et al., 2021 doi: 10.1002/1873-3468.14165). The N-terminal prodomain of AtMC1 also alters its condensate dynamics (Ruiz-Solaní et al., 2024 doi: 10.1093/plcell/koad172) and genetically, we know that the AtMC1 prodomain is required for the autoimmune phenotype. This could potentially suggest that both the 360 loop and the N-terminus can participate in the (direct or indirect) interaction of AtMC1 with NLRs.

I thank the authors for discussing the (seemingly autonomous) condensation behavior of MC1. Since (autonomous) NLR condensation is not the focus of the current manuscript, I find this discussion will suffice.

- What is the effect of MC1 activity on NLRs? Are the NLRs a catalytic target of MC1? Please discuss.

So far, we have not detected cleavage of the tested NLRs by AtMC1 when co-expressed (Figure 6 and Figure EV5). However, biochemical evidence strongly suggests that AtMC1 may not only act as a canonical metacaspase, cleaving substrates after arginine or lysine, but also as a disaggregase, clearing toxic amorphous protein aggregates (Ruiz-Solani et al., 2023). In this regard, AtMC1 is quite unspecific in its disaggregase activity, being able to dissolve a broad range of aggregated proteins, including pathological forms of human proteins such as aggregated huntingtin, tau or transthyretin, among others. It is thus plausible that AtMC1 clears misfolded/aggregated NLRs as part of the protein quality control machinery after an immune reaction to prevent proteotoxicity and to help returning the cell to a normal state. Given the stress-dependent condensation of AtMC1, it is tempting to speculate that this clearing activity or even the canonical protease activity of AtMC1 occurs within condensates/stress granules. We are currently testing these hypotheses, although the spatio-temporal dimension of the phenomenon and the difficulty of working with condensation-prone proteins makes it technically very challenging. This is particularly the case in the context of immunity, considering the fact that NLRs are also challenging proteins to do microscopy with, in their full-length forms.

I thank the authors for elaborating on their ongoing research addressing the disaggregase properties of MC1. I will be looking forward to a follow-up manuscript.

Line 49f: Please use a different reference here. Huang et al 2022, and Jia et al. 2022 are predominantly focused on protein structure and are not the original articles to prove genetical dependence of EDS1 and TNLs.

The references have been changed as suggested.

No further comment.

Line 52: Recently, Ibrahim et al. 2024 (BioRxiv, <https://doi.org/10.1101/2024.09.19.613839>) report localization of NRG1 helper NLR to target organellar membranes. I suggest to include the reference and to choose a more careful phrasing. I am not aware of any published evidence for 'perturbation of PM integrity' by NLRs, please revise.

As suggested, the phrasing has been changed to better reflect available data and the new reference has been added.

No further comment.

In my opinion, the evidence in Jacob et al. 2021 for calcium channel activity could be stronger and rests on other, previous studies. I suggest to also include the reference Bi et al. 2022 (Cell) that first shows a more comprehensive study of NLR channel activity for Zar1. The authors may also want to stick to the more conservative phrasing of 'calcium-permeable cation-selective channel activity' that was used in the Bi et al. original publication.

We have added the suggested reference and modified the text.

No further comment.

Line 92: Grammar 'catalytic dependent manner'. catalysis dependent manner?

Line 96: Grammar 'plant have display reduced'

Line 235: Typo. Should be NRG1, not 'NGR1'

Line 305: Grammar. Should be 'interaction of AtMC1 variants with RPS2 and SS14 was confirmed...'

Line 312f: Convoluted phrasing and grammar. '1) AtMC1 or its catalytic activity or its targets are guarded by (an) NLR(s). 2) AtMC1 participates in the proteostasis of immune components and overexpressed catalytically inactive AtMC1 binds and traps immune components (NLRs and components involved in PTI), thus preventing turnover. Please revise the sentence avoiding brackets.

Line 328: Define 'camta3'. Give some background to camta3

Line 329: Should be 'present in the Arabidopsis Col-0 accession'.

Line 330f: Doubling. 'autoimmune plant atmc1 AtMC1C220A-GFP plants'

Line 555: '(139 DN-NLRs or out 166 NLRs present in Arabidopsis)'. Should be '139 out of 166'?

All typos and convoluted phrasing have been corrected.

No further comment.

Figure 1, D: Could the authors please discuss in the results part why PR1 shows different expression between the CRISPR and T-DNA line? This is a potential weakness of the study and at least an explanation is required.

The difference in PR1a levels could be explained by the different nature of the two mutants. The CRISPR mutant is a clean mutant lacking the entire gene (Appendix Figure S1), whereas the T-DNA mutant contains the insertion after the first exon, corresponding to approximately 74 amino acids. Regardless of this difference, the results of Pst DC3000 growth and plant fresh weight measurements suggest that both genotypes result in autoimmunity. However, we have now included a description of the difference between the two mutants and their potential effect in terms of PR1a accumulation in the results section.

It is indeed what I also suspected that the two mutants are not fully comparable. With the added sentence I find the confusion is resolved.

Figure 6, A: Please adjust the alignment of the IP anti-HA blot the blot below and above.

The alignment has been adjusted.

No further comment.

Figure 10: I do not agree with the illustration of pentameric condensates. This is not supported by evidence. From the molecular size of the light microscopic puncta, a 'resistosome'-like structure is not supported. Rather have a look at Ma et al. 2024 (Nature), <https://doi.org/10.1038/s41586-024-07668-7>

Although, it is unclear how widespread this filament formation of SINRC2 is across other NLRs. However, these alternative NLR high molecular weight assemblies underline that pentameric assemblies are not the only possible assemblies. The authors ought to be more neutral in their illustration.

The figure was a bit misleading, as it did not intend to depict structured resistosomes, but rather unstructured accumulations of misfolded NLRs and other immune regulators. We have included a new figure that hopefully reflects better this idea.

I thank the authors for making the adjustment. I have no further comment.

Can the authors please illustrate how EDS1 integrates into their presented working model?

Our interpretation is that the observed autoimmunity in *atmc1 AtMC1C220A-GFP* plants may be due to hyperactivation of a SA-dependent NLRs that require the feedback loop through EDS1-PAD4-ADR1 node, which normally amplifies immune responses, together with SA-independent pathways. This interpretation is included in the discussion. We prefer not to include EDS1 in the scheme provided in Figure 10 to keep it simple, otherwise we feel we should include other components of the pathway that have been tested and it would become overly complicated.

I agree with the idea to keep the figure simple.

Referee #3:

The revised manuscript has addressed minor technical issues raised in the initial assessment. As stated in the comments shared in the first round, this work exemplifies how genetics of using loss-of-activity of immune components sensitises genetic backgrounds to uncover its role in fine-tuning immunity. I believe that this work should be published without further delays. There are no major issues in the quality of work to claim the main messages.

CRAG building - Campus UAB
C. de la Vall Moronta, s/n
08193 Cerdanyola del Vallès
Barcelona
cragenomica.es

Dr. Núria S. Coll

Principal Investigator, CSIC Researcher
Centre for Research in Agricultural Genomics (CRAG)
Edifici CRAG. Campus UAB
08193 Cerdanyola del Vallès, Barcelona, Spain
Phone: +34 935636600 Fax: +34 935636601
E-mail: nuria.sanchez-coll@cragenomica.es

March 6th 2025

Dear Dr. Breiling,

On behalf of all authors, I am pleased to re-submit our work “**Lack of AtMC1 catalytic activity triggers autoimmunity dependent on NLR stability**” (EMBOR-2024-60304-T). We kindly ask you to reconsider this new version of the manuscript for publication as **Research Article** in **The EMBO Journal**

In the revised version of the manuscript, we have carefully addressed all of the editorial requests. We would like to take the opportunity to provide specific clarifications regarding two points:

- **P-values in figure legends:** We have not included exact p-values in the legends for Figures 2e, 3d, 4c, EV1d, f, EV2c, and EV4e. This is because we employed an ANOVA test with multiple comparisons in these cases, and including individual p-values would complicate the data visualization. For transparency, we have uploaded a supplementary table containing all the p-values for your review.
- **Re-use of panels/parts of panels:** Indeed, certain panels were reused between Figures 3B (last panel) and EV4F (last panel), Figure 3B (4th panel) and Appendix Figure S5B (second panel), and Figure EV4A (second panel) and EV4B (second panel). We have replaced these reused panels with images from different individuals from the same experiment to avoid raising any concern. The panels in question were originally part of a larger experiment, and the reuse was intended to present the data more clearly, as separate data units or figures. However, we believe that using images from different individuals better addresses the concern.

We trust that the manuscript is now in its final, acceptable form for publication. We sincerely appreciate your consideration and look forward to your feedback.

Best regards,

Núria S. Coll, PhD

Principal Investigator, CSIC Researcher
Centre for Research in Agricultural Genomics (CRAG)

Members of the Consortium:

Dr. Nuria Coll
CRAG
C. Vall Moronta s/n. Edifici CRAG
Campus UAB
Cerdanyola del Vallès, Barcelona 08193
Spain

Dear Dr. Coll,

I am very pleased to accept your manuscript for publication in the next available issue of EMBO reports. Thank you for your contribution to our journal.

Yours sincerely,
